# Using Sentinel-1 radar amplitude time series to constrain the timings of individual landslides: a step towards understanding the controls on monsoon-triggered landsliding

Katy Burrows[1], Odin Marc[1], and Dominique Remy[1]

[1]Géoscience Environnement Toulouse, Toulouse, France

**Correspondence:** Katy Burrows (katy.burrows@get.omp.eu)

**Abstract.** Heavy rainfall events in mountainous areas trigger destructive landslides, which pose a risk to people and infrastructure and significantly affect the landscape. Landslide locations are commonly mapped using optical satellite imagery, but in some regions their timings are often poorly constrained due to persistent cloud cover. Physical and empirical models that provide insights on the processes behind the triggered landsliding require information on both the spatial extent and timing of landslides. Here we demonstrate that Sentinel-1 synthetic aperture radar amplitude time series can be used to constrain landslide timing to within a few days and present four techniques to accomplish this based on time series of: (i) the difference in amplitude between the landslide and its surroundings, (ii) the spatial variability of amplitude between pixels within the landslide, and geometric (iii) shadows and (iv) bright spots cast within the landslide. We test these techniques on three inventories of landslides of known timing, covering various settings and triggers, and demonstrate that a method combining them allows 20-30% of landslides to be timed with an accuracy of 80%. Application of this method could provide an insight on landslide timings throughout events such as the Indian summer monsoon, which triggers large numbers of landslides every year and has until now been limited to annual-scale analysis.

## 1 Introduction

Every year, many mountainous areas in tropical zones are affected by destructive rainfall-induced landslide events that pose a major risk to people and infrastructure (Petley, 2012). With the advent of Earth observation from space, inventories of these landslides are routinely compiled from optical and multi-spectral satellite imagery (e.g., Marc et al., 2018; Emberson et al., 2022). These data are then used to inform hazard management, as inputs to physical, empirical and statistical models, and to assess the impact the event has had on the landscape, for example by estimating the volume of sediment eroded (Jones et al., 2021; Kirschbaum and Stanley, 2018; Ozturk et al., 2021; Wu et al., 2015).

Landslide early warning systems, susceptibility zonation maps, nowcasts and hazard scenarios use information on the size, location and timing of past landslides alongside information on the landscape conditions and triggering event (Guzzetti et al., 2020). While optical satellite imagery provides information on the size and location of landslides, cloud-free, daylight images are required. In unfavourable weather conditions, there may be a delay of weeks or months before cloud-free imagery over the whole area affected by triggered landslides is acquired (Robinson et al., 2019; Williams et al., 2018). This means that the timing

of the landslides is often poorly constrained by the optical satellite imagery. In practice, this strongly limits or simply prevents any attempt to relate landslide metrics and hydrometeorological metrics resulting from successive or long rainfall events, whether through empirical scalings (e.g., Marc et al., 2018, 2019b) or physical modeling (e.g., Wilson and Wieczorek, 1995; Baum et al., 2010). In many tropical settings, multiple successive typhoons are common, for example typhoons Nesat, Haitang and Talim, which made landfall within a 2-month period in 2017 in Taiwan (Janapati et al., 2019). If no cloud-free optical satellite imagery is acquired between such successive trigger events, the relationship between the hydrological impact of the storms and the triggered landslides cannot be precisely established. Similarly, the Indian summer monsoon (June-September) triggers hundreds of landslides every year in the Nepal Himalaya and cloud-free optical satellite imagery is unlikely to be available throughout this period (Robinson et al., 2019). This limits analysis of these landslides to the seasonal scale and prevents association of individual landslides or spatio-temporal clusters of landslides to specific peaks in rainfall. (e.g. Marc et al., 2019a; Jones et al., 2021). Studies based on optical satellite images affected by cloud cover that attempt to map landslides triggered by sequences of earthquakes and/or rainfall events may also be unable to distinguish between different triggers (e.g. Ferrario, 2019; Martha et al., 2017; Tanyaş et al., 2022).

Beyond remote sensing, several approaches have been used to constrain landslide timing. Landslides that occur close to inhabited areas, or that damage important pieces of infrastructure may be described in news reports or on social media (e.g., Kirschbaum et al., 2010; Franceschini et al., 2022). Information on the timing of such landslides can also be obtained from interviews with local residents (Bell et al., 2021) and through citizen science initiatives (Sekajugo et al., 2022). Rainfall intensity-duration thresholds have previously been derived for landslides dated in this way (e.g. Dahal and Hasegawa, 2008) and for landslides whose timings and properties are known through monitoring and field surveys (e.g. Guzzetti et al., 2007; Ma et al., 2015). However, such information on landslide timing is unlikely to be available for the majority of landslides in an inventory, and is usually biased towards populated areas and areas accessible by road (Sekajugo et al., 2022). Seismic records of landslides can also provide highly precise information on their timings, but will mostly record large landslides and require multiple seismic stations to allow timing of an individual, localised landslide (e.g. Yamada et al., 2012; Hibert et al., 2019) Current methods of obtaining landslide timing information in the absence of cloud-free optical satellite images are therefore not widely applicable.

Regularly acquired synthetic aperture radar (SAR) images, for example those acquired by the European Space Agency Sentinel-1 constellation, represent a new opportunity to obtain landslide timing information for many landslides at regional scale. SAR images penetrate cloud cover and the Sentinel-1 satellites acquire images every 12 days on two tracks over all land masses globally. Numerous studies have demonstrated that SAR data can be used to detect the spatial distribution of landslides in the case where their timing is already known, for example in the case of earthquake-triggered landslides where it can be assumed that the landslides occurred concurrently with ground shaking (Aimaiti et al., 2019; Burrows et al., 2019, 2020; Ge et al., 2019; Konishi and Suga, 2019; Masato et al., 2020; Mondini et al., 2019; Yun et al., 2015). SAR can be also used to monitor movements of slow-moving landslides (e.g. Ao et al., 2020; Bekaert et al., 2020; Hu et al., 2019; Kang et al., 2021; Solari et al., 2020). Mondini et al. (2019) used SAR to establish the timing of a single large landslide. However, to-date SAR has not been used to refine timing estimates of landslide inventories. Here we present landslide timing methods based on

the Sentinel-1 SAR dataset in Google Earth Engine that represent a step towards this goal of improved landslide inventory temporal resolution, and could unlock new comparisons between measured or modelled hydrological time series and landslide occurrence.

## 2  Data and Methods

In order to obtain information on event timings for landslides triggered by sequences of earthquakes or rainfall or by long rainfall events, we propose a two-step process, whereby landslide locations are mapped as polygons using optical or multi-spectral satellite imagery, and the timings of individual landslides are then obtained from SAR time series. In this paper we address the second of these steps. We use Sentinel-1 time series over inventories of landslides whose timings are already known to test potential landslide timing methods.

### 2.1  Case studies

We used three published polygon inventories of landslides whose timings are known a-priori to test and develop landslide timing methods. All three inventories are located in vegetated areas, which is generally the ideal condition for widespread landslide mapping based on multi-spectral satellite imagery.

We used two inventories of landslides triggered by short rainfall events, whose timing is therefore known to within a few days (rainfall time series are available in the supplementary material). First, landslides triggered in Hiroshima, Japan by a heavy rainfall event which took place from 28 June to 9 July 2018, which were mapped using a combination of drone and aerial imagery (inventory from The Association of Japanese Geographers, 2019). The majority of landslides triggered by this event are believed to have occurred during peaks in rainfall intensity on the 6-7 July (Hashimoto et al., 2020).

Second, we used landslides triggered by Cyclone Idai in Zimbabwe between 15-19 March 2019. This inventory was compiled as part of the study of Emberson et al. (2022) using post-event PlanetScope optical satellite images acquired on 20 and 24 March. Media reports on this event suggest that the majority of landsliding occurred between the 15-17 March (BBC News, 2019; Ministry of Information and Broadcasting, 2019; OCHA, 2019).

The third inventory used to test our methods was compiled by Roback et al. (2018) for the $M_w$ 7.8 Gorkha, Nepal earthquake, which occurred on 25 April 2015. The Nepal Himalaya is an area which experiences long periods of cloud cover and large numbers of rainfall-triggered landslides annually due to the monsoon and the country's steep topography. The steep topography of Nepal also makes it particularly challenging for SAR applications as it leads to distortion of the SAR imagery. It is thus important to test landslide timing methods in this environment, but inventories of rainfall-triggered landslides of known timing are not available. Therefore we instead used earthquake-triggered landslides. Since the inventory of Roback et al. (2018) covers a large area (28,000 km$^2$), with different areas having different Sentinel-1 coverage, we focussed on triggered landslides within three valleys: Trishuli, Bhote Kosi and Buri Gandaki. These valleys experience large numbers of rainfall-triggered landslides every year (Marc et al., 2019a).

All inventories were filtered to remove landslides smaller than 2000 m$^2$. Since the Sentinel-1 GRD data set has a pixel size of $10 \times 10$ m, this should result in a minimum of 20 SAR pixels within each landslide. This resulted in inventories of 543 landslides for the Hiroshima event and 383 for Zimbabwe. In Nepal, an additional step was required; the M$_w$ 7.8 mainshock on 25 April was followed by other possible landslide triggers including the M$_w$ 7.3 Dolakha aftershock on 12 May as well as the annual monsoon, whose onset was around 9 June (Williams et al., 2018). Therefore, we also removed all landslides specified by Roback et al. (2018) to have been triggered by an aftershock or by rainfall and used only those triggered by the mainshock in our analysis. This left 650 landslides in Trishuli, 1554 in Bhote Kosi and 922 in Buri Gandaki. The Dolakha aftershock is known to have triggered further landsliding (see Marc et al., 2019a) and Roback et al. (2018) noted that in some areas, no cloud-free optical satellite images were available between the mainshock and this aftershock, making it difficult to differentiate between these two triggers. However of the three valleys we consider here, landslides associated with this aftershock have only been observed in Bhote Kosi, which was the closest to the epicentre (Martha et al., 2017). 97% of the co-seismic landslides in Bhote Kosi were recorded as identifiable in imagery acquired prior to the aftershock and can therefore be associated definitively with the mainshock (Roback et al., 2018). Furthermore, since the co-event pair of SAR images for Bhote Kosi (24 April - 18 May 2015) spans both the Gorkha earthquake on 25 April and the Dolakha aftershock on 12 May, these two trigger events are blended into a single time window by our methods in Bhote Kosi.

## 2.2 Theory: SAR backscatter and landslides

A SAR satellite actively illuminates the Earth's surface with microwave energy, and records the phase and amplitude of the returned signal. The difference in phase between two images acquired over the same area at different times can be used to track the movement of the Earth's surface, for example movement on a fault during an earthquake, while the amplitude describes the strength of the backscattered SAR signal. The power of the signal transmitted $P_r$ and received $P_t$ by the sensor are described by Eq. 1, where $\lambda$ is the wavelength, $G^2$ is the two-way antenna gain and $R$ is the slant range (Small et al., 2004).

$$\overline{P_r} = \frac{\lambda^2}{(4\pi)^3} \cdot \int\limits_{Area} \frac{P_t G^2 x^0}{R^4} dA \tag{1}$$

This equation is solved to obtain $x^0$, the backscatter coefficient, which can be either $\sigma^0$, $\gamma^0$ or $\beta^0$ depending on whether the integration is carried out in the ground (ellipsoid) plane, the plane perpendicular to the look direction or the slant-range plane respectively (Small et al., 2004). Different studies have demonstrated that all three of these backscatter coefficients can be applied to detect vegetation removal due to landslides and other processes such as deforestation and wildfires (e.g. Ban et al., 2020; Belenguer-Plomer et al., 2019; Bouvet et al., 2018; Esposito et al., 2020; Hernandez et al., 2021; Konishi and Suga, 2018; Mondini, 2017; Mondini et al., 2019; Motohka et al., 2014). Here we used $\gamma^0$.

SAR backscatter is dependent on a number of factors, including the polarisation and wavelength used by the SAR system, the local slope orientation relative to the SAR sensor and the roughness and dielectric properties (e.g. soil moisture, presence of vegetation) of the material that the microwave energy interacts with at the Earth's surface. Sentinel-1 acquires C-band SAR data with a wavelength around 5.5 cm in two polarisations: "VV" (vertical polarisation) and "VH" (cross polarisation). We

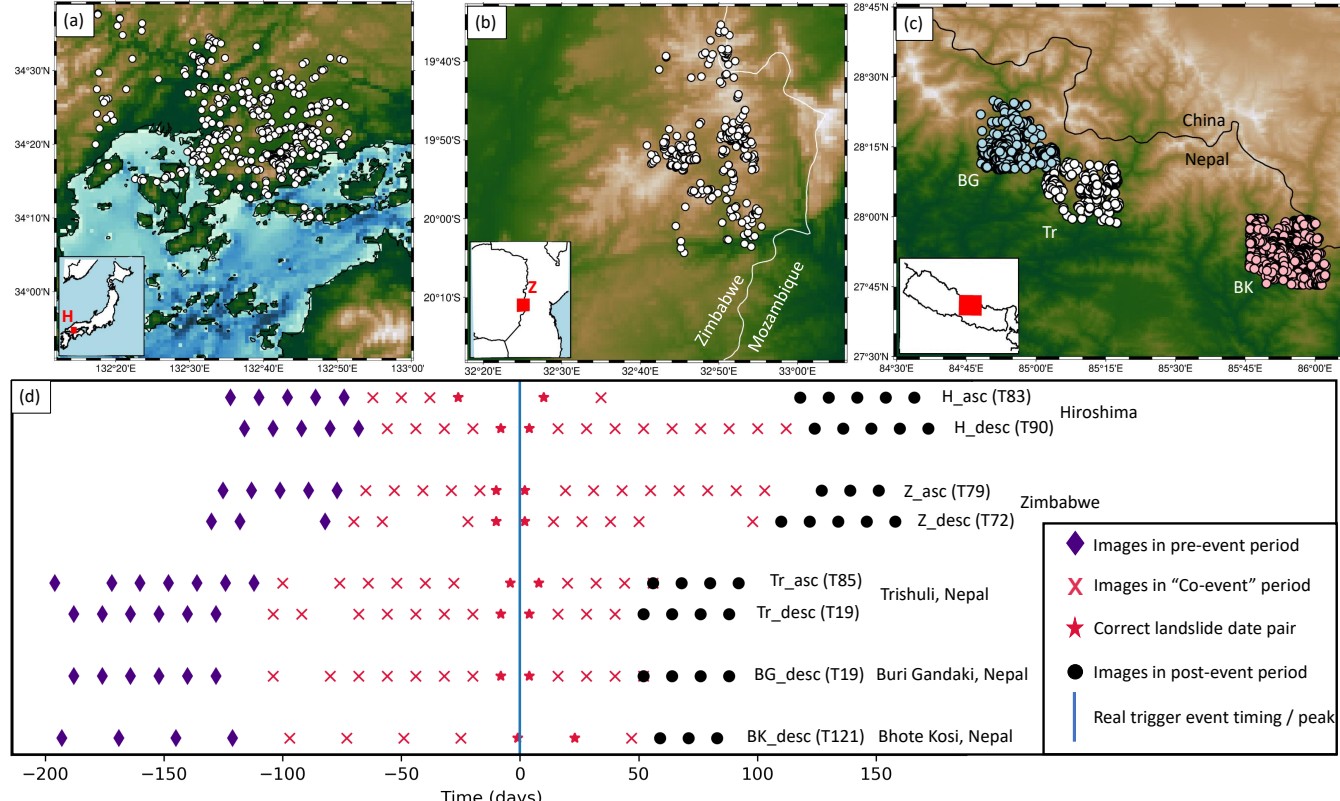

**Figure 1.** (a-c) locations of the five inventories of triggered landslides used in this study to test landslide timings. (a,b) landslide locations from The Association of Japanese Geographers (2019); Emberson et al. (2022) are shown by white circles. (c) three subsets of the inventory of Roback et al. (2018) are shown for the Buri Gandaki, Trishuli and Bhote Kosi valleys by blue, white and pink circles respectively (d) SAR image acquisition timings before, during and after a defined "co-event" window of 6 months relative to the real event timing. The orbit number of each track is given in brackets.

tested both of these polarisations, but found VV to perform better than VH so present only the results for VV (results for VH can be found in the supplementary material). VV data have also been acquired more consistently throughout the lifetime of
Sentinel-1 than VH. In general, for vertically polarised SAR images, rougher surfaces result in increased backscatter, as does increased soil moisture. However, the relationship between these properties and the SAR amplitude is not simple: roughness has a stronger effect in locations with a high incidence angle (Baghdadi et al., 2016; Dubois et al., 1995), while changes in soil moisture have a larger effect at low incidence angles (Baghdadi et al., 2016).

    Landslides alter the local topography (and therefore the local incidence angle) of the landscape through the movement of
material and remove vegetation, which alters the dielectric properties and roughness of the Earth's surface. For this reason, landslides can result in both increases and decreases in amplitude. In fact within a single landslide, the amplitude of some pixels may increase while some decrease (e.g. Mondini, 2017).

## 2.3 SAR data and preprocessing

To construct our SAR amplitude time series, we used the Google Earth Engine Sentinel-1 ground range detected (GRD) data set. Google Earth Engine is a freely accessible, cloud-based platform that allows users to access Sentinel-1 data without the technical expertise and computational facilities otherwise required to process SAR data. It also provides access to other datasets used in this study, such as Sentinel-2 and the shuttle radar topography mission (SRTM) digital elevation model (DEM). The Sentinel-1 GRD data are preprocessed following the workflow of Filipponi (2019) to obtain the backscatter coefficient $\sigma^0$ at a resolution of 20 x 22 m in radar coordinates. The data are then resampled onto a 10 m grid in projected coordinates. We then applied the module of Vollrath et al. (2020) using the 30 m SRTM DEM to carry out an angular radiometric slope correction based on the volume scattering model of Hoekman and Reiche (2015). This has the effect of converting from $\sigma^0$ (normalised in the ellipsoid plane) to $\gamma^0$ (normalised in the plane perpendicular to local satellite look direction). The aim of this step is to reduce the effects of topography on the SAR backscatter. In preliminary testing, we found that $\gamma^0$ performed better than $\sigma^0$. The module of Vollrath et al. (2020) also provides a shadow and layover mask that can be used to remove areas that are not imaged by the satellite due to the viewing angle and local topography. This masking step is important for landslide studies as they are likely to be carried out in areas of steep topography.

For each of our three events, we defined "pre-event", "co-event" and "post-event" periods (shown for each event on Fig. 1d). The length of the co-event period was defined as six months based on the intended application to the Nepal monsoon, in which landslides may occur between May and October. However, for the three Nepal inventories, this was reduced to five months in order to allow a sufficient number of pre-event images to be acquired following the satellite launch in 2014 and sufficient post-event images to be acquired before the end of July since few Sentinel-1 images are available over Nepal in August and September 2015. The lengths of the pre-event and post-event time series were selected to be long enough to calculate statistics such as the mean without requiring the processing of unnecessary images. These pre-event and post-event image stacks are required in some of the techniques outlined in Sect. 2.

Unfortunately, insufficient data were acquired on the ascending orbit over Buri Gandaki and Bhote Kosi in Nepal, so we only present results based on the descending track data for these two inventories. In Fig. 1d and throughout the manuscript, we refer to SAR data according to the event, and satellite orbit direction, for example, the ascending track over Zimbabwe will be referred to as $Z_{asc}$. Any date for which SAR imagery only covered part of the inventory was omitted from the time series.

## 2.4 Four techniques to retrieve landslide timing from SAR amplitude time series

Here, we present four potential techniques for analysing Sentinel-1 GRD time series and identifying the image pair spanning the landslide date. Figures showing these four techniques applied to three example landslides can be found in the supplementary material.

### 2.4.1 Technique 1: Landslide-background difference

We expect a landslide to result in a permanent change in an amplitude time series. However, factors other than landslides can also result in amplitude change. In particular, the rainfall that triggers the landslides will alter the soil and canopy moisture content and so may also alter the amplitude of the returned signal. To overcome this, we calculate a background amplitude signal for each landslide. First, we calculated a buffer region between 30 and 500 m around each landslide (Fig. 2a). Then we filtered this buffer to remove any pixels that lie within other landslide polygons and pixels that are dissimilar to those within the landslide. In order to assess pixel similarity we calculated three variables from pre-event satellite imagery. First, we calculated the greenest pixel composite of the normalised difference vegetation index (NDVI) from Sentinel-2 (or, where this was unavailable, Landsat-8) images acquired in the year prior to each event. Pixels with similar NDVI values are expected to have similar land-cover. Second, we used the mean amplitude $A_{mean,j}$ (Eq. 2) and third, the amplitude variability $\Delta A_{mean,j}$ (Eq. 3) for every pixel $j$ through a stack of $N$ pre-event images. Pre-event amplitude and amplitude variability have previously been used by Spaans and Hooper (2016) to identify statistically similar pixels in SAR images. This allows us to remove pixels that are unlikely to exhibit similar behaviour to those within the landslide, for example pixels located on the opposite side of a ridge, in a river or with different surface cover.

$$A_{mean,j} = \frac{1}{N} \sum_{i=1}^{N} A_{i,j} \tag{2}$$

$$\Delta A_{mean,j} = \frac{1}{N} \sum_{i=1}^{N} (A_{mean,j} - A_{i,j}) \tag{3}$$

For each landslide, we calculate the median amplitude in the landslide polygon and for these background pixels for every image in the co-event time series. A step change in the difference between the median landslide amplitude and the median background amplitude is then used as an indicator of landslide timing. As previously described, landslides can result in both increases and decreases in SAR amplitude. Thus we accept both a step increase and a step decrease in this metric as indicators of landslide timing.

### 2.4.2 Technique 2: Pixel variability

Ban et al. (2020) observed that in forested and grassland areas, the removal of vegetation due to forest fires led to an increase in the variability of vertically polarised Sentinel-1 $\gamma_0$ between neighbouring pixels. Since landslides result in a similar denudation of vegetated areas, we expect that similar effects may occur. Therefore, we calculated the standard deviation of $\gamma^0$ within each landslide polygon and used a step increase in this as a potential indicator of landslide timing (e.g. Fig.2d).

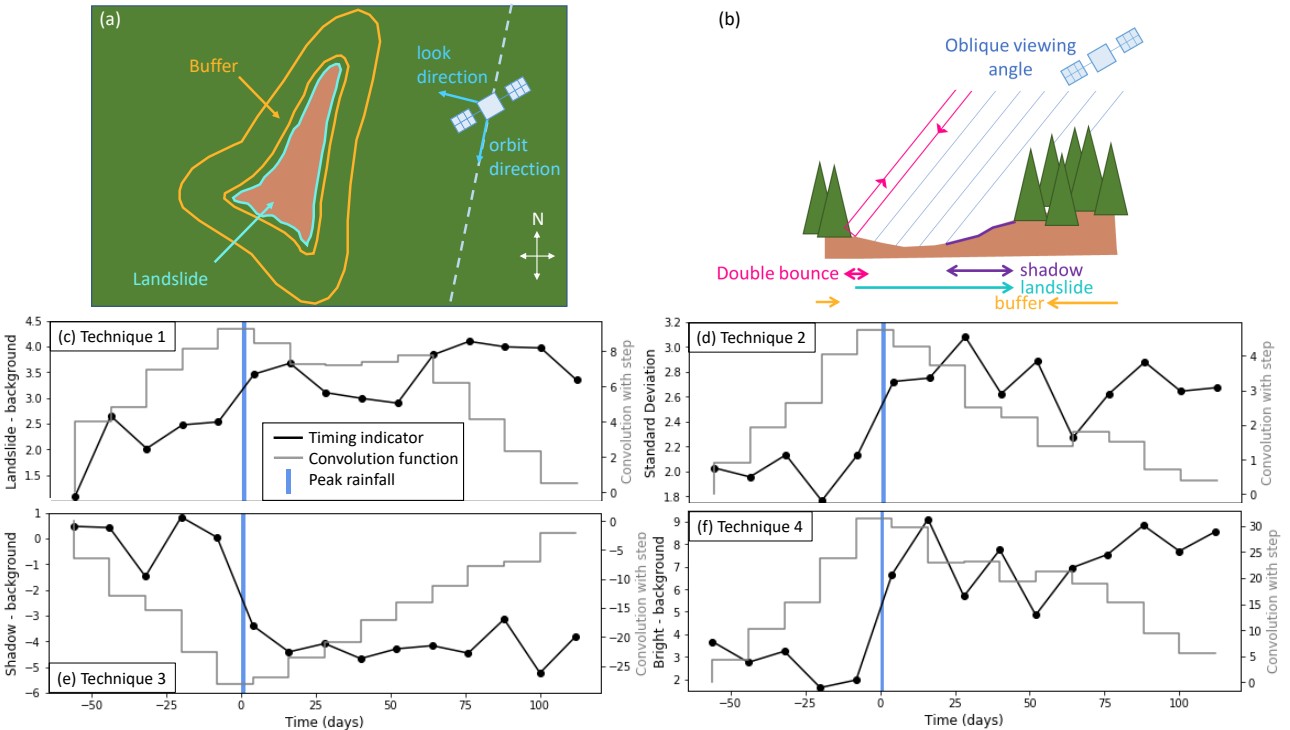

**Figure 2.** (a,b) Plan and lateral views of a landslide and satellite, showing how background and shadow regions are formed in this study. (c-f) Example time series for a single landslide from the Hiroshima dataset using SAR data from Sentinel-1 track 090D for Techniques 1-4 respectively (Sect. 2.4). Blue bar shows the duration of the rainfall event during which the landslide was triggered. Grey shows the convolution between the time series and a step function.

### 2.4.3 Technique 3: Geometric shadows

Since SAR is acquired obliquely (with an ellipsoid incidence angle of 31-44 ° for the data used here), steep changes in scatterer surface height can result in geometric shadows. The wavelength of Sentinel-1 means that is primarily scattered from the canopy in forested areas, which means that shadows can be cast at the edges of deforested areas if these edges run approximately perpendicular to the satellite look direction (Fig. 2b). Bouvet et al. (2018) developed a method for automatically detecting deforested areas based on these geometric shadows. Since landslides remove vegetation, we expect that shadows should also

be cast at the edges of landslides, and that the appearance of new shadows could be used as an indicator of landslide timing. Furthermore, the three-dimensional shape of the landslide could result in shadows cast within the landslide itself, for example if the landslide has a steep scar. This effect has previously been observed within a large landslide in Nepal by Ao et al. (2020). It is worth noting that, while Bouvet et al. (2018) applied their methods in areas of gentle slopes, the area of a shadow cast by an object of a given height is dependent on slope and aspect: trees of the same height will cast a larger shadow on slopes facing

away from the sensor than on those facing towards it. Therefore, we expect this technique to be more successful for slopes that face away from the sensor.

In comparisons of multiple inventories of the same event prepared by different people or groups, there are often small discrepancies in the exact size, shape and location of each landslide (Milledge et al., 2022; Pokharel et al., 2021). Spatial mismatches between landslide polygon locations could lead to pixels on the edges of landslides being excluded from the analysis. Since shadow pixels are most likely to lie at the edges of the landslide polygons, it is important not to exclude the edge of a landslide from the analysis. Therefore we extended the area covered by each landslide polygon by 20 m (two SAR pixels) where this did not lead to intersection with another landslide in the inventory. We then identified pixels whose amplitude decreased within this enlarged polygon as shadows. Bouvet et al. (2018) identified shadow pixels as those whose $\gamma_0$ value decreased by >= 4.5 dB during the deforestation event. We tested values between 3 and 6 dB and also found that a threshold of 4.5 dB performed best. We calculated the mean $\gamma_0$ value for every pixel from the pre-event and post-event image stacks and assigned those that decreased by >= 4.5 dB as shadow pixels. The co-event time series of these shadow pixels was then analysed and a step decrease in the median shadow $\gamma_0$ relative to the median background $\gamma_0$ (Sect. 2.4.1) was used as an indicator of landslide timing.

### 2.4.4 Technique 4: Geometric bright spots

As well as shadows, the new geometry created by a landslide scar may result in bright spots on the far side of the scar, which are due to double-bounce scattering of the microwave energy between the exposed soil and vertical objects such as tree trunks and focussing of the energy scattered from the 3D surface into a small area in the radar coordinate system (Villard and Borderies, 2007, Fig. 2b). Similarly to the Geometric shadows technique, we applied a 20 m buffer to the landslide polygon, identified pixels that had undergone a significant increase in mean $\gamma_0$ between the pre-event and post-event image stacks and assigned these as "Bright". Here we found that the optimum $\gamma_0$ increase threshold was 5 dB. The co-event time series of these bright pixels was then analysed and a step increase between median bright $\gamma_0$ compared to the median background $\gamma_0$ (Sect. 2.4.1) was used as an indicator of landslide timing.

### 2.5 Identification of landslide date pairs

Here we detail how the four techniques described above are used to retrieve landslide timings both individually and in combination. The variable associated with each technique is calculated for each landslide for every SAR image during the co-event period (Fig. 2 c-f). For each technique, we expect that the landslide should cause a step change in the time series, allowing us to identify the date pair spanning the landslide timing. In order to identify this step change, we take the co-event time series and subtract from it its mean value to obtain a co-event time series centred on zero. Then we convolve this series with a step function composed of a series of -1s and 1s that is twice its length. The output of this convolution is a series that, after truncating to the same length as the original co-event time series, should contain a peak (in the case of a step increase) or trough (in the case of a step decrease) at the location of the strongest step change in the time series.

**Table 1.** Confusion matrix for determining how convolution peak size relates to whether a landslide timing is likely to be correct

| | Peak synchronous with trigger event | Peak asynchronous with trigger event |
|---|---|---|
| Peak > threshold (timestamped) | True Positive | False Positive |
| Peak < threshold (masked) | False Negative | True Negative |

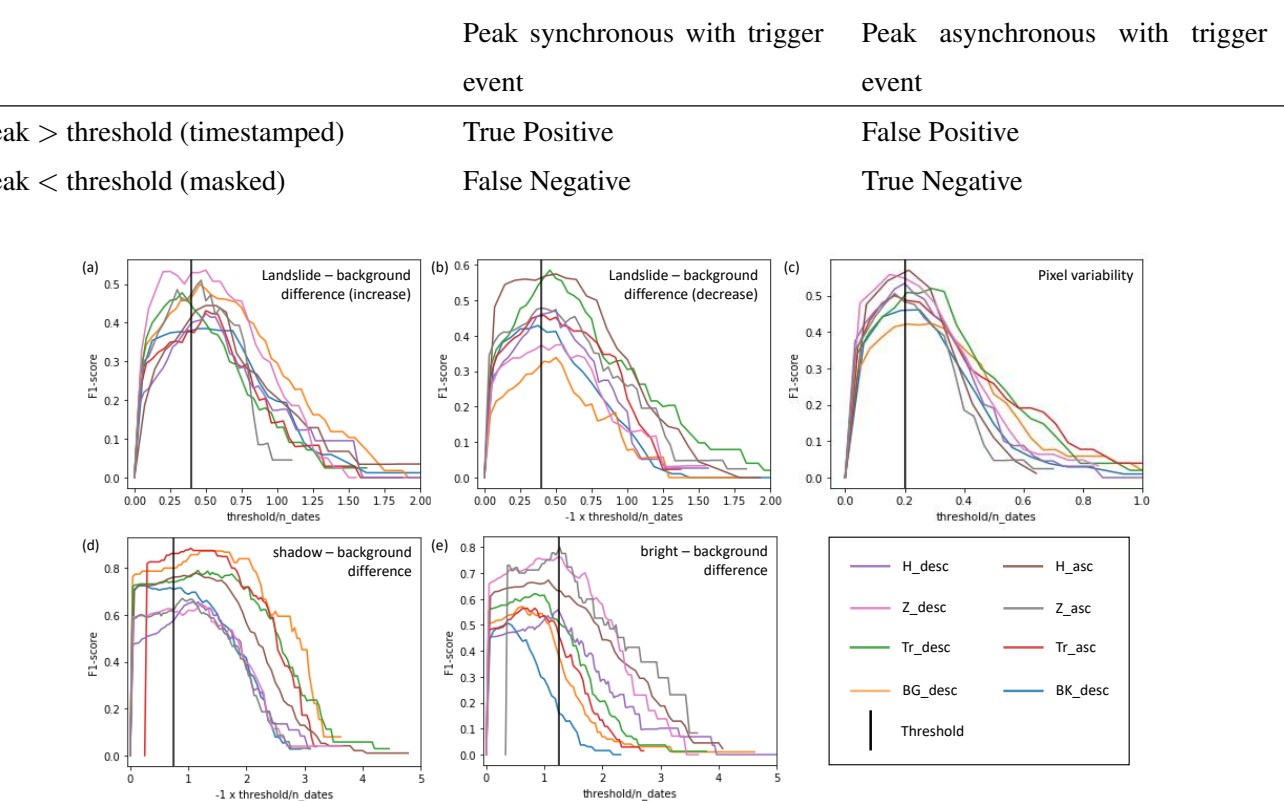

**Figure 3.** F1 scores for a range of peak thresholds for the Landslide-Background (a and b correspond to step increase and decrease respectively), Pixel Variability (c), Geometric Shadows (d) and the Geometric Bright Spots (e) Techniques. Vertical black lines show selected thresholds.

The size of the peak or trough depends on the magnitude of the increase or decrease, the level of noise elsewhere in the time series and the length of the co-event time series ($n_{dates}$). A bigger peak or trough for a time series of the same length indicates a larger step change and less noise and is therefore a more reliable indicator of landslide timing. We therefore apply a peak size threshold to remove unreliable landslide timing estimates. To select this threshold for each technique, we use the F1-measure, a statistic that combines both precision and recall. This F1-measure was calculated for a range of peak thresholds using the confusion matrix defined in Table 1 (Fig. 3). Based on this, we require a peak of $0.4 \times n_{dates}$ for the Landslide-Background Technique, $0.2 \times n_{dates}$ for the Pixel Variability Technique, $0.75 \times n_{dates}$ for the Geometric Shadows Technique and $1.25 \times n_{dates}$ for the Geometric Bright Spots Techique. We also assessed whether the level of noise in the time series for each technique (estimated from the variability in pre-event and post-event time series), could be used to indicate whether a timing estimate was likely to be correct, but found this to be less reliable than the convolution peak size.

After identifying landslide timings using each technique individually, we combined these, assigning a date pair to a landslide if it was selected by at least two of our four techniques. As previously described, a 20 m buffer was applied to each landslide polygon for Techniques 3 and 4 in order to allow for some spatial mismatch between the landslide polygons and the SAR imagery. This was not done for Techniques 1 and 2 as including non-landslide pixels unnecessarily would have the effect of muting the step change in the time series for these techniques. However, for landslides that have not been assigned a timing at this stage, we now repeat the above process using this 20 m buffer for Techniques 1 and 2 as well. This step increases the number of landslides assigned a timing by around 5%.

The final step is to combine the predictions from the ascending (satellite moving northwards and looking east) and descending (moving southwards and looking west) SAR tracks. By carrying out the process described above using both the ascending and descending track SAR time series, we can obtain two sets of timings for a given landslide inventory, which can then be combined. This has several advantages. First, landslides that are not assigned a date pair using data from one track may be better timed by the second, increasing the number of landslides that can be assigned a date pair. In particular, landslides that are masked due to foreshortening or layover may be better imaged in the other track. Second, the acquisition dates of the two tracks are slightly offset so a landslide that is assigned a date pair by both tracks is timed more precisely. For example, a correctly timed landslide in our Zimbabwe inventory should be timestamped as 7-19 March 2019 by the descending track time series and 12-24 March 2019 by the ascending track time series. From both together, the landslide would be timed as 12-19 March 2019, improving the precision from 12 days to 7. This more precise date is also more likely to be correct since it is derived from two sets of independent observations of the landslide.

## 3 Results

The number of landslides assigned the correct date pair, and the number of landslides assigned any date pair are shown for each of the techniques described in Sect. 2.1 in Table 2, followed by the combined result for each track and the combined result from both tracks for each event. Individually, none of the techniques is sufficiently accurate and consistent to provide useful information on landslide timing. However, when compared to a random baseline calculated from $\frac{1}{n_{dates}}$ (the percentage of landslides we would expect to be assigned the correct date pair by a method with no skill assigning a random date pair), all individual techniques consistently perform better than this baseline. Not all landslides are assigned a date by every technique, for example if no geometric shadows are cast within the landslide polygons.

### 3.1 Combining techniques

As previously described, we combined the four individual techniques by taking whichever date pair was predicted the most often for each landslide. Since it is not possible for both a step increase and a step decrease in the Landslide-Background Technique to predict the same date, the maximum number of times the same date can be predicted is 4. The number of landslides assigned a date pair by at least 2 techniques, at least 3 techniques and by all 4 techniques and the number of these date pairs that are correct is shown in Table 2. The strong reduction in number of timed landslides when going from an individual technique to

**Table 2.** For each case study, the total number of landslides, the number that are masked due to foreshortening or layover in the SAR images and amplitude timing results for the four techniques described in Sect. (2). For each technique and combination of techniques, we give the number of correctly assigned date pairs against the total number of assigned date pairs. Where timings were obtained from combinations of techniques (Te) or tracks (Tr), the number of these is specified in brackets.

| | Hiroshima | | Zimbabwe | | Trishuli | | Buri Gandaki | Bhote Kosi |
|---|---|---|---|---|---|---|---|---|
| Orbit direction | Desc | Asc | Desc | Asc | Desc | Asc | Desc | Desc |
| Total Landslides | 543 | | 383 | | 650 | | 922 | 1554 |
| Non-masked | 543 | 540 | 383 | 383 | 485 | 474 | 592 | 894 |
| *Individual techniques* | | | | | | | | |
| Landslide-background inc | 44/177 | 37/97 | 39/67 | 27/72 | 37/74 | 39/123 | 76/186 | 88/269 |
| Landslide-background dec | 56/182 | 121/226 | 41/172 | 55/147 | 80/160 | 54/236 | 53/152 | 100/264 |
| Pixel Variability | 101/258 | 101/167 | 79/158 | 52/112 | 84/194 | 73/169 | 100/227 | 53/152 |
| Geometric Shadows | 50/144 | 143/192 | 35/60 | 48/75 | 47/70 | 35/42 | 19/20 | 43/62 |
| Geometric Bright Spots | 35/89 | 50/68 | 28/43 | 10/11 | 59/89 | 47/90 | 45/70 | 20/42 |
| *Combined techniques, single track* | | | | | | | | |
| Combined ($\geqslant$ 2Te) | 55/71 | 91/105 | 40/52 | 39/43 | 52/66 | 37/43 | 40/54 | 45/69 |
| Combined ($\geqslant$ 3Te) | 14/16 | 31/32 | 11/11 | 2/2 | 18/18 | 7/7 | 4/6 | 7/7 |
| Combined (4Te) | 1/1 | 5/5 | 0/0 | 0/0 | 2/2 | 3/3 | 0/0 | 0/0 |
| *Combined techniques, combined tracks (final method)* | | | | | | | | |
| Asc & Desc (Total) | 135/171 | | 82/113 | | 110/130 | | - | - |
| Asc & Desc (2Te, 1Tr) | 80/111 | | 76/95 | | 80/108 | | - | - |
| Asc & Desc ($\geqslant$ 3Te) | 55/60 | | 17/18 | | 30/32 | | - | - |
| Random baseline ($1/n_{dates}$) | 7% | 17% | 10% | 7% | 8% | 8% | 8% | 14% |

2, 3 and then 4 techniques in combination underlines the fact that the nature of the change in amplitude varies widely between landslides. However, landslides dated by 2 or more techniques are correctly dated much more often. Across all 8 tracks, 503 landslides are assigned a date pair by 2 or more techniques of which 399 (79%) are correct. 99 landslides are assigned a date pair by 3 or more techniques of which 92 (93%) are correct; Fig. 4 shows the number of times each date pair in the co-event series is selected by $\geqslant$ 2, $\geqslant$ 3 and 4 techniques.

## 3.2 Combining tracks

As described in Sect. 2.5, for each event, we used the ascending and descending tracks to generate a broader and more robust set of date pairs. When requiring the same date pairs from at least 2 techniques on either of the two tracks in Hiroshima, Zimbabwe and Trishuli, we assigned date pairs to 31%, 30% and 20% of the landslides respectively. Of these assigned date

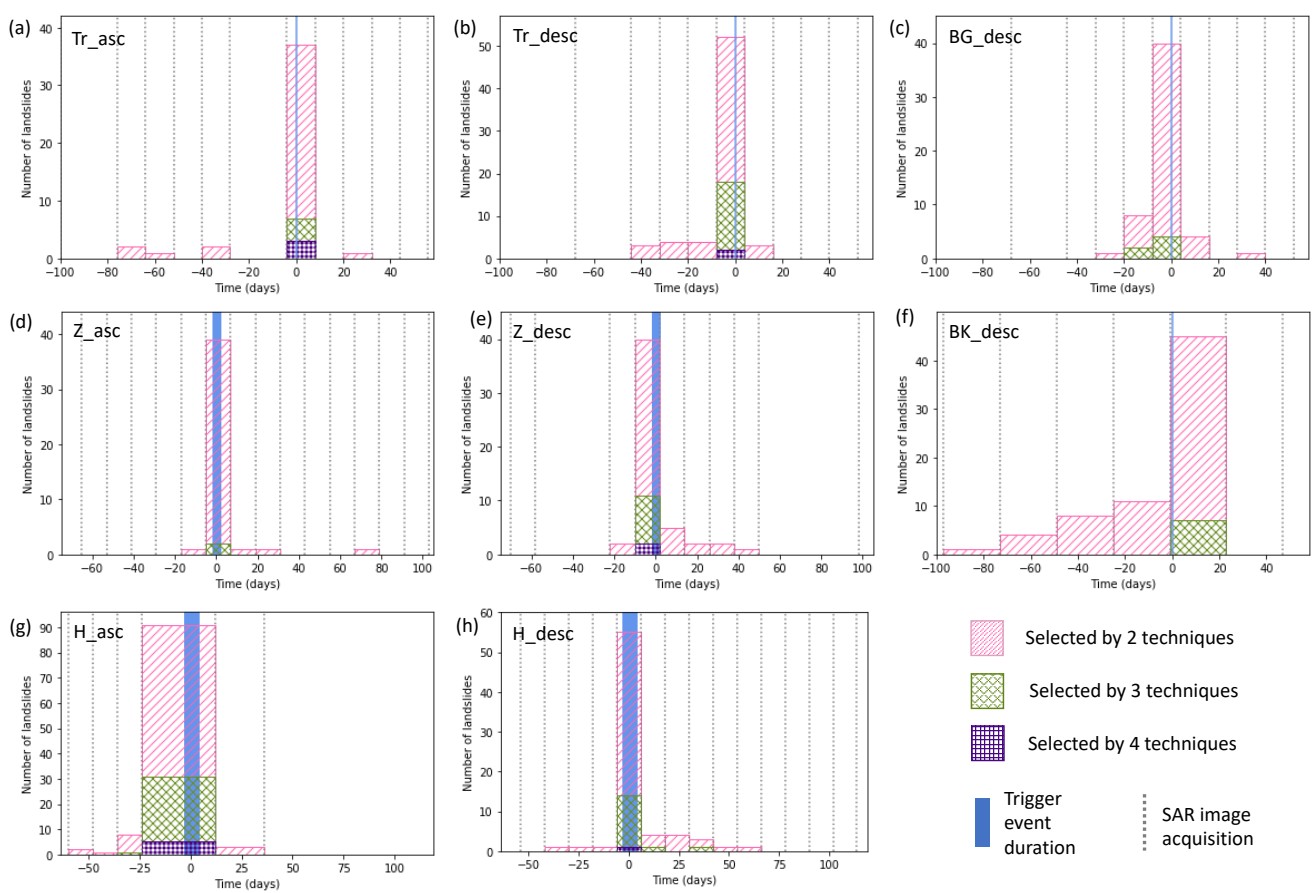

**Figure 4.** Histograms showing the predicted landslide timings for each event and SAR track when four techniques are used in combination.

pairs, 79% were correct in Hiroshima, 73% in Zimbabwe and 85% in Trishuli (Total in Table 2). These assigned dates can be divided into two subgroups: landslides timed by 2 techniques on one track only (2Te, 1Tr, Table 2) and landslides assigned timings by 3 or more techniques across both tracks ("⩾ 3Te", Table 2). Although they represent a smaller group (5-10% of the landslides from each inventory),the latter were assigned the correct date more often (89-94% of the time). We also tested the case where landslides were timestamped based on overlapping date pairs being selected by one technique from each track, but found that this yielded too many incorrect timings to be useful.

## 4 Discussion

Here, we first evaluate the success and limits of our method as a function of landslide characteristics, namely size, vegetation and slope aspect, and as a function of co-event time series length. Then we discuss reasons for landslides lacking assigned

timings, or worse, incorrect timings. Finally we consider the potential of applying InSAR coherence time series approaches to landslide timing. Note that throughout the discussion, we use "the method" to refer to our algorithm that combines assigned timings from multiple techniques and both ascending and descending track SAR (Sect. 2.5, "Asc & desc total" Table 2).

## 4.1 Factors affecting landslide timing detection ability.

We assessed the performance of our landslide timing method as a function of the landslide characteristics, in terms of pre-event vegetation and landslide area. We also analysed the effect of slope aspect on the four individual landslide timing techniques. For future applications, this helps to determine the environments where the method can be expected to work. It also provides an insight on potential biases in terms of the subset of a landslide inventory that can be assigned timings using our method. Finally, we assessed the effect that the length of the co-event period has on the performance of our method, since this may vary for future applications.

### 4.1.1 Vegetation

In order to assess the effect that vegetation cover has on the method we propose here, we compared the number of correctly timed, incorrectly timed and untimed landslides with different values of pre-event NDVI (Fig. 5 a-c). We took the maximum NDVI value for each pixel in the year preceding the event and used Sentinel-2 data for Zimbabwe and Hiroshima and Landsat 8 for Trishuli. In all three inventories, the majority of mapped landslides occurred in vegetated areas ($0.6 < NDVI < 0.8$). In all three cases, a landslide in a more vegetated area was more likely to be assigned a date and this date was more likely to be correct.

### 4.1.2 Area

Another factor that could potentially effect the applicability of our method is landslide area. Fig. 5d-f shows the distribution of landslides against landslide area. In Zimbabwe and Hiroshima, a higher proportion of larger landslides were assigned a date pair and in all three cases a higher proportion of the date pairs assigned to larger landslides were correct. We limited our testing to landslides whose area was greater than 2000 m$^2$. Since our techniques rely on landslides containing multiple SAR pixels in order to calculated the statistics such as the standard deviation, there is likely to be a lower limit on the area of landslides that can be timed that was not reached here.

### 4.1.3 Aspect

The effect of aspect on landslide timing ability is more complicated than that of vegetation and area, since it is likely to vary between the ascending and descending track SAR. Therefore, in Fig. 6, we show the ascending and descending track predictions for each individual technique for Zimbabwe (results are similar for Hiroshima and Trishuli). The different techniques we propose in Sect. 2 have different relationships with aspect. For the Landslide-Background Difference Technique, it appears that landslides on slopes facing towards the sensor are more likely to experience a step increase, while slopes facing away

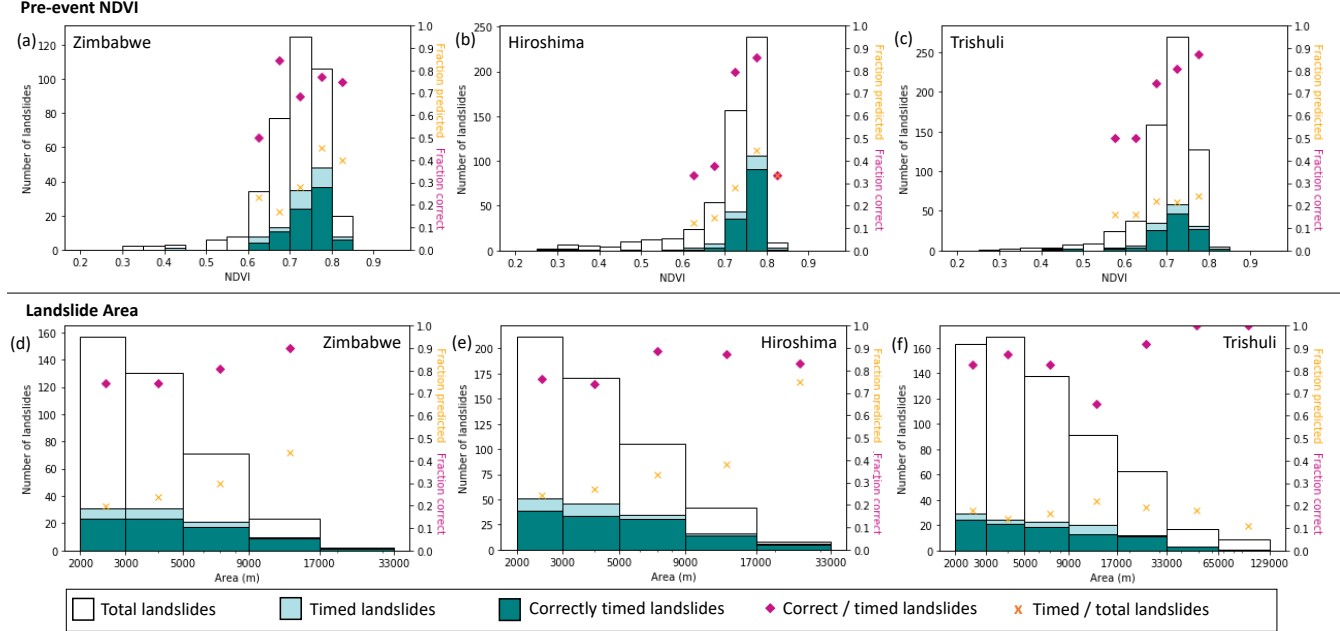

**Figure 5.** The distribution of total landslides (white), landslides assigned a time (light green) and landslides assigned the correct time (dark green) for different values of pre-event NDVI (greenest value in the year preceding the event, a-c) and landslide area (d-f). Predictions were obtained from combining ascending and descending track SAR (Sect. 3.2).

from the sensor are more likely to experience a step decrease. For the Pixel Variability and Geometric Bright Spot Techniques, aspect does not appear to have a strong effect on how likely a landslide is to be assigned the correct time. For the Geometric Shadows Technique, a higher proportion of landslides are assigned a date (and therefore exhibit a shadow) on slopes facing away from the sensor. This was expected since the same height difference will cast a larger shadow on a slope facing away from the sensor than one facing towards it (Bouvet et al., 2018). Dates assigned by the Geometric Shadows Technique also appear more likely to be correct for slopes facing away from the sensor on $Z_{asc}$, but this pattern is less clear on $Z_{desc}$. Thus a path for future improvement of our method may be to apply a variable detection threshold as a function of slope aspect, particularly for the landslide-background difference technique.

### 4.1.4 Co-event time period duration

We defined a "co-event" period of 6 months when testing the landslide timing methods in this paper. This time period was selected to be roughly the duration of the Nepal monsoon. However, some applications, for example the case of successive storms, may not require such a long window. It is therefore useful to assess how the length of this time window affects the accuracy of predictions. In order to assess this, we took the tracks with the most complete time series ($Z_{asc}$, $H_{desc}$, $Tr_{desc}$ and $BG_{desc}$) and assessed their performance over 2-8 month periods. Fig. 7 shows the percentage of assigned timings that are correct based on at least 2 techniques for each track at each time period.

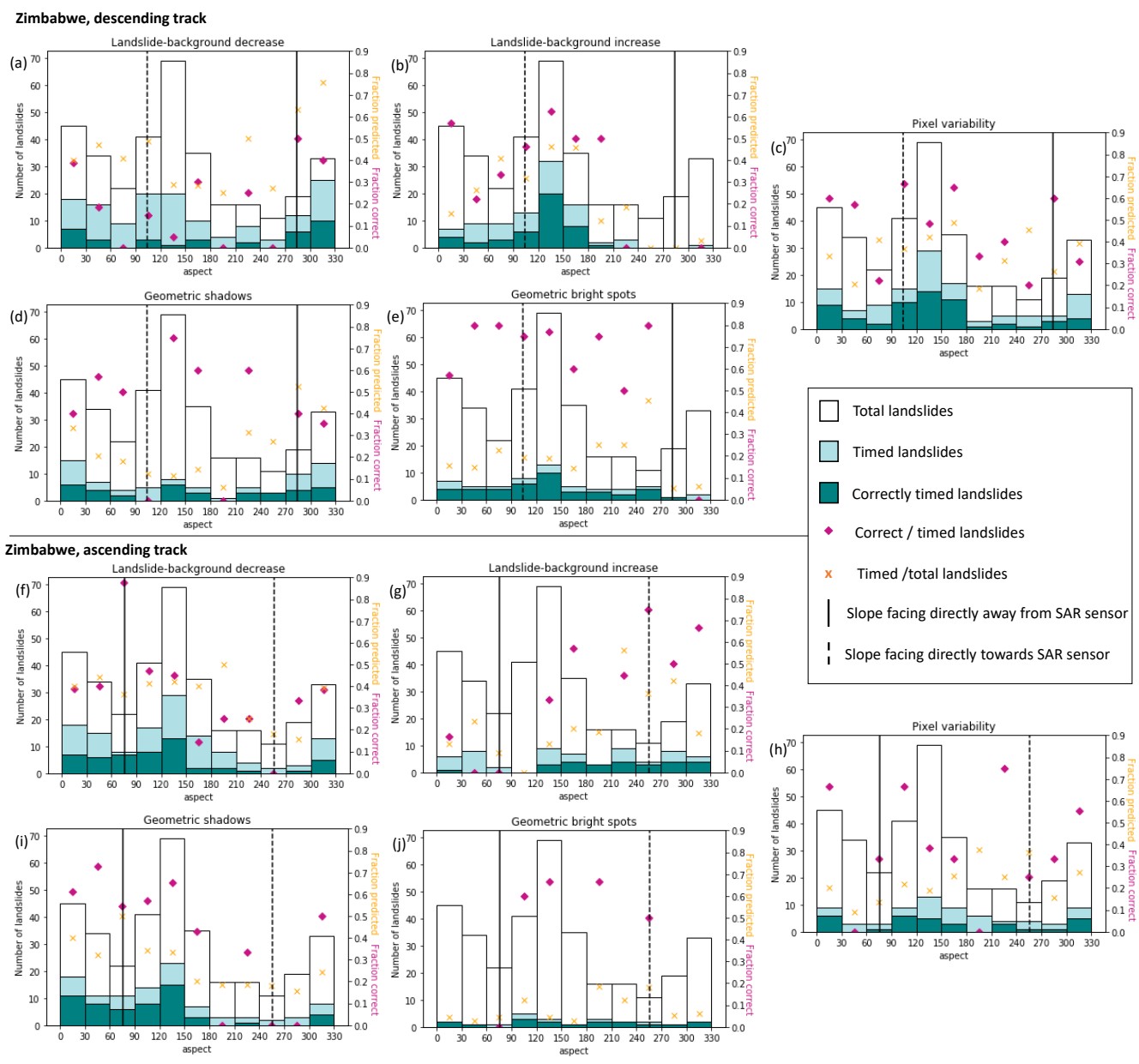

**Figure 6.** The distribution of total landslides (white), landslides assigned a time (light green) and landslides assigned the correct time (dark green) over aspect for each technique using ascending and descending track SAR over the Zimbabwe dataset.

On three of the four tracks, particularly $BG_{desc}$ and $H_{desc}$, the accuracy decreased as the co-event period was decreased. This was especially observed for periods of less than 5 months. We suggest that noise may be less attenuated in a shorter time series, resulting in increased numbers of false positives. This may explain the relatively poor performance in Bhote Kosi

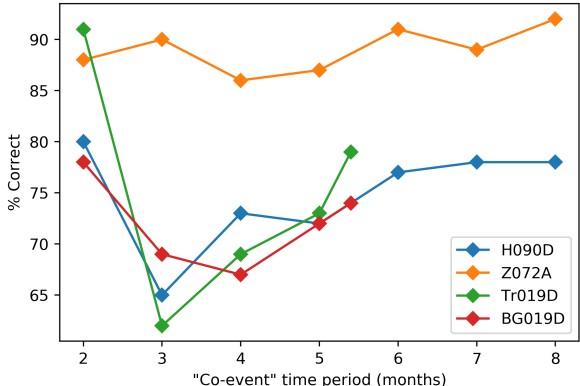

**Figure 7.** The percentage of landslide timings that are correct when assigned by $\geqslant 2$ of the techniques described in Sect. 2.4 for a range of co-event time periods.

compared to the other case study areas, since comparatively few images were available for this case study (Fig. 4). The loss of accuracy is recovered when the co-event period is further decreased to 2 months, possibly due to the comparatively small number of possible wrong date pairs available within a 2-month period. Thus for future studies that aim to constrain the timings of rainfall-triggered landslides, we recommend defining a long co-event period (6-7 months), but for studies that aim to distinguish landslides triggered by a rapid succession of triggers (e.g. the events studied by Tanyaş et al., 2022), a co-event period of 2 months or less may be better.

## 4.2 Why do some landslides have no timing estimation?

In all our case studies, a large proportion of landslides are not assigned any date pair by our method. Some of these landslides, primarily in Nepal, lie in areas of foreshortening or layover in the SAR images and so were removed from the analysis (Sect. 2.3). This represents between 25% and 43% of the landslides on each track in Nepal, so that if these masked landslides are ignored, the method sensitivity in Nepal is similar to the less steep landscapes of Zimbabwe and Japan. Beyond this, landslides that are not assigned a date pair are a direct result of the target criteria of our method: a significant step change in at least two of the techniques outlined in Sect. 2.4. We showed in Sect. 2.5 that imposing a threshold on the convolution function peak was essential to reach a usable specificity, but this will also have required some correct timings to be discarded. Thus time series with a high degree of noise or where the landslide results in only a small step change in the metric will not produce a date pair. Finally, although we attempted to account for any spatial mismatch between polygon locations and SAR imagery by expanding the boundaries of each polygon by 20 m in every direction (Sect. 2.5), any spatial disagreement beyond this scale is likely to lead to landslides not being assigned a timing.

Lack of trees (i.e. low NDVI) and unfavourable slope aspect relative to the SAR sensor are likely to suppress any shadow or bright pixels associated with a landslide, and may also reduce the change in median amplitude, hampering detection of

landslide timing. Landslides that effect sparsely vegetated areas, for example barren or agricultural lands, or areas that have previously been deforested or eroded are thus less likely to be assigned a timing by our method. Noise in the time series may be related to either natural or anthropogenic changes to the ground properties (e.g. agricultural practices, particularly on the hillslopes of Nepal).

     Future refinement of the method may increase the number of landslides assigned a timing. Possible means of accomplishing

this include finding robust and systematic links between landslide setting and optimal thresholds for the individual techniques and (as suggested by Fig 5, 6). This would allow metric thresholds to be adapted to the setting of each landslide polygon. To address the problem of noise within the time series that masks the landslide timing signal, future work may involve adding a first step to our algorithm in which pixels exhibiting high levels of temporal variability are excluded from the landslide and background areas. Finally, our method may be improved by the development of other metrics, for example based on VH

polarised SAR data or InSAR coherence time series, which have previously been used to detect landslides in forested and arid zones respectively (Cabré et al., 2020; Handwerger et al., 2022).

### 4.3    Possible causes of incorrect landslide timings

    In all of our case studies, our method assigns the wrong date pairs to small number of timed landslides. There are several possible reasons for this. There may be real changes in the time series that are not landslides, for example snowfall or melt, change

in vegetation, change in soil moisture or human activity. Activities related to the landslide, for example the removal of material from a blocked road, may also contribute to this. Random noise in the SAR signal may also result in false landslide timings. We note that for future applications, the timing confidence within a landslide population can be separated into landslides timed by 3 or more techniques and those timed by only 2 techniques (Table 2).

     Another possibility is that delayed or multi-stage failure occurred for some landslides. Our method is designed to detect only

a single failure. In the case where multi-stage failure results in more than one step change in the time series, the convolution in Sect. 2.5 will detect only the largest step change. Though it is beyond the scope of this study, in theory it would be possible to assess if the time-series contain a second peak of similar magnitude to the largest one in order to assess possible multistage failure or landslide reactivation.

     Delayed failure seems particularly likely for Zimbabwe and Hiroshima, where a large proportion of the incorrect landslide

timings are made up of the date pair immediately after the rainfall event (Fig. 4d, e, h). It is possible that some of the landslides in these inventories did not fail immediately during the rainfall, but instead failed after a delay of a few days due to rising pore pressure following rainfall infiltration within the hillslope (Iverson, 2000). This is particularly possible in the case of $Z_{desc}$, where the end of the rainfall event on 19 March 2019 coincides with the acquisition of the first post-event image, so that only a short delay would be required for the landslide to occur during the time window immediately after the rainfall (19-31 March

2019) rather than during the time window that spans the rainfall (7-19 March 2019). If these landslides are counted as correct in our analysis, the combined success rate in Zimbabwe is increased from 73% to 82%, bringing it in line with Hiroshima and Trishuli (Table 2), while for landslides timed by 3 or more techniques ("⩾3Te" in Table 2), the success rate is increased from 89% to 94%.

Although the Gorkha earthquake was followed by a large aftershock (12 May) and by the monsoon (approximate onset 9 June Williams et al., 2018), we are more confident of the true date of the landslides for this event. It is possible that some landslides could have been either triggered or reactivated by monsoon rainfall. However, none of the incorrect landslide timings in Nepal are in June, making this unlikely (Fig. 4a-c,f).

## 4.4 InSAR Coherence

Interferometric SAR (InSAR) coherence is a measure of the signal quality of an interferogram (an image used to measure ground deformation formed from two SAR images acquired over the same area at different times). InSAR coherence is sensitive to changes at the ground surface between the acquisition of the two SAR images: areas where the scatterers have changed significantly have high levels of noise in an interferogram and so a low coherence. Coherence is therefore sensitive to landslides and has previously been used to detect landslide densities or individual large landslides (Burrows et al., 2019, 2020; Goorabi, 2020; Yun et al., 2015).

The coherence of each pixel in an interferogram can be estimated from the similarity in amplitude and phase change between the two SAR images for small groups of neighbouring pixels. Coherence surfaces and the phase data required for their calculation are not available through Google Earth Engine. However, coherence for Track 19 in Nepal has previously been calculated by Burrows et al. (2019), covering the Buri Gandaki and Trishuli inventories tested here. This allows us to compare techniques of landslide timing based on SAR amplitude and InSAR coherence for these two case studies.

Burrows et al. (2019) processed the data at the same resolution used here ($20 \times 22$ m) and used a $3 \times 3$ moving window to estimate coherence, so that the coherence surface has a resolution of $60 \times 66$ m. Similarly to the Landslide-Background Technique, we obtained the median coherence of pixels within each landslide through time and the median coherence of pixels within a 60-500 m buffer of each landslide polygon to give a background coherence. We then examined the ratio between the landslide and background coherence through time. Using this ratio performed better than using the landslide coherence alone, probably because other factors, such as the length of time between the two images used to form the interferogram, can also effect coherence. Fig. 8 shows the median coherence ratio of a single landslide for different image pairs through time. This demonstrates two effects that we expect to see. First, the coherence that spans the landslide timing is low. This drop in coherence has previously been used to detect landslide locations (Burrows et al., 2019; Goorabi, 2020; Yun et al., 2015). However Sentinel-1 often has a low background coherence in vegetated areas due to its wavelength, which can make any coherence decrease due to a landslide difficult to detect. Second, the coherence of post-event image pairs is higher than pre-event image pairs due to the removal of vegetation by the landslide (previously used by Burrows et al., 2020). Based on these two observations, we propose two landslide timing detection techniques based on InSAR coherence time series.

**Technique C1:** A step increase in the coherence ratio corresponds to the first post-event image pair.

**Technique C2:** A temporary decrease in the coherence ratio corresponds to the co-event image pair. For each coherence pair, this temporary decrease is calculated from the sum of the decrease in coherence ratio from the previous image pair to this one and the increase in coherence ratio from this image to the next (adapted from the $\Delta$C_sum method of Burrows et al., 2020).

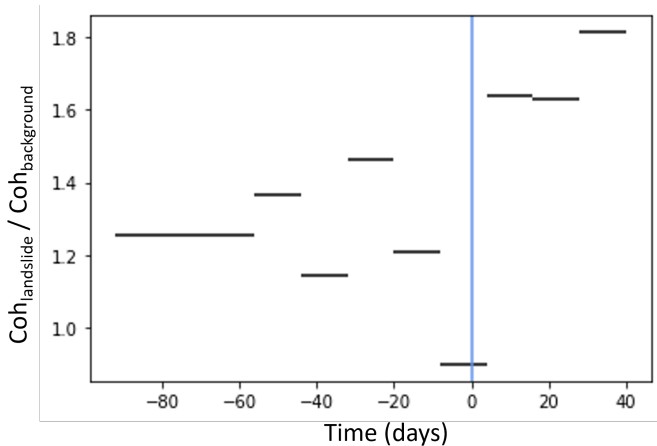

**Figure 8.** Time series of ratio between landslide coherence and background coherence for a single landslide in Trishuli (black horizontal lines). Blue vertical line shows earthquake timing.

.

**Table 3.** The ratio of correct / assigned landslide timings for the two coherence-based techniques

|  | $\text{Tr}_{\text{desc}}$ | $\text{BG}_{\text{desc}}$ |
|---|---|---|
| Technique C1 | 54/154 (36%) | 59/169 (35%) |
| Technique C2 | 82/312 (26%) | 96/396 (24%) |
| Combined | 27/57 (47%) | 20/56 (36%) |
| Non-masked landslides | 485 | 592 |

Overall, the coherence-based techniques have a lower success rate than the amplitude-based techniques (Table 2), indicating that incorporating these data would decrease the specificity of our method. However, it is worth noting that of the 47 landslides correctly timed across the two events using the C1 and C2 combined, only 3 had already been timed using the combined amplitude-based techniques in Sect. 3.1, suggesting that the incorporation of coherence techniques could increase sensitivity, if these could be made more reliable.

Currently, only the Sentinel-1 SAR constellation acquires SAR data with sufficient coverage and acquisition frequency for widespread use in landslide timing studies. These data are acquired at C-band, which usually has low coherence in vegetated areas. L-band data is better suited to InSAR-coherence-based landslide detection in vegetated areas (Burrows et al., 2020). The planned NASA-ISRO NISAR mission has a similar acquisition strategy to Sentinel and will acquire L-band SAR data. It will be worth reassessing the potential of InSAR coherence time series for landslide timing detection following the launch of this satellite.

## 5 Conclusions and Future Perspectives

In the case of long or successive rainfall events, landslide inventories compiled from optical satellite imagery are often poorly constrained in time, making it difficult to associate them with specific triggering conditions. Here we present a method of using Sentinel-1 SAR amplitude time series in Google Earth Engine to identify the timing of triggered landslides to within a few days. We find that by combining multiple techniques and ascending and descending track SAR, it is possible to assign timings to up to 30% of landslides in an inventory with an accuracy of 80%. A small number of landslides (5-10%) can be

timed with an accuracy of ⩾90%. Here we applied our method to optically-derived landslide inventories, but it could also applied to datasets from other sources, for example those based on LIDAR scans or high resolution optical images that allow landslide volumes to be estimated (Bernard et al., 2021). The precision of our method, which in most cases is 12 days, should be sufficient in the case of multiple successive storms or earthquakes to attribute landslides to a given event (Ferrario, 2019; Janapati et al., 2019; Tanyaş et al., 2022). For monsoon landslide timings, this precision is not sufficient for construction

of intensity-duration or intensity-antecedent rainfall thresholds at the hourly scale typical in the literature (e.g. Bogaard and Greco, 2018). However, thresholds based on weekly rainfall would be achievable and of interest for understanding triggering conditions in the Himalayan region. Furthermore, it should allow us to establish whether landslides occur in temporal clusters that relate to specific peaks in rainfall or are distributed throughout the monsoon. These two end-members would have very different implications in terms of hydrological and slope stability modelling and thus on hazard evaluation. Application of our

method to the Indian summer monsoon should also allow us to better constrain whether landslides systematically occur with a specific delay after the onset of the monsoon and/or simultaneously with reported flooding or bursts of intense rainfall (Gabet et al., 2004).

Our method assigns timings to only 30% of landslides in an inventory, thus timing information is not obtained for the majority of landslides. Therefore, while our method provides a valuable insight into landslide timings during long or successive rainfall

events, further work could allow us to obtain a more comprehensive view. First, our method may be refined by future studies, for example through variable metric thresholds adapted to the setting of each landslide or by incorporating both amplitude and coherence time series. Second, remote sensing approaches such as we present here could be combined, where available, with other methods of establishing landslide timing, for example reports of individual landslides or seismic data (Bell et al., 2021; Hibert et al., 2019; Yamada et al., 2012). Finally, We also expect that both the precision and the number of landslides that can

be assigned timings may increase in the future as more SAR data becomes available, for example from the planned NISAR constellation. Overall, our method represents a step towards improved temporal resolution for triggered landslide inventories. This could further our understanding of monsoon-induced landsliding in the Nepal Himalaya and elsewhere.

*Code and data availability.* Sentinel-1 GRD and Sentinel-2 data are available open-access from ESA Copernicus and were accessed through Google Earth Engine (https://developers.google.com/earth-engine/datasets/catalog/sentinel, last access 19 Jan 2022). Images from the USGS

Landsat archive were accessed through Google Earth Engine (https://developers.google.com/earth-engine/datasets/catalog/landsat, last access 19 Jan 2022). Landslide polygons were obtained from Roback et al. (2018) for Nepal (available at https://doi.org/10.5066/F7DZ06F9),

The Association of Japanese Geographers (2019) for Hiroshima and Emberson et al. (2022) for Zimbabwe. The Google Earth Engine code used in the manuscript is included in the supplementary material. Further calculations were carried out using the Python Numpy package. Images were produced using Python Matplotlib (Hunter, 2007) and PyGMT (Uieda et al., 2021) software.

*Author contributions.* KB and OM conceived the study. KB carried out data curation and analysis of the data and wrote the original draft of the manuscript. All authors were involved in reviewing and editing the manuscript and in developing the methodology.

*Competing interests.* The authors declare no conflict of interest.

*Acknowledgements.* KB is fund by a post-doctoral grant from the Centre National d'Études Spatiales (CNES): "Characterising the temporal evolution of rainfall-triggered landslides using radar and optical satellite data". We thank Robert Emberson for sharing some landslide
inventories.

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
