# Peer review of "Using Sentinel-1 radar amplitude time series to constrain the timings of individual landslides: a step towards understanding the controls on monsoon-triggered landsliding"

_Natural Hazards and Earth System Sciences, 2022_

## Author Comment (AC1)

We thank the reviewer for taking the time to read and evaluate our manuscript. We will prepare a line by line response, but here we respond to some of the major comments made on the manuscript (Quoted in bold in this document).

First in the review introduction, two major points are raised (In bold), that are also referred to later in the review

**"However, the authors are able to come up with a time estimation only for 20% of landslides with an accuracy of 80%. Therefore, I doubt if this is successful research in the end. Frankly speaking, I am not sure and just hesitating to say that the results are promising. However, what I can say is the output of this research is not fulfilling what the authors are promising in the abstract/conclusions.**

First, we will revise the abstract and conclusions to make sure that it is clear that our method will only provide landslide timings for around 20% of the landslides in an inventory and that we do *not* promise a full inventory with timings for all landslides. This was not previously clear and we thank the reviewer for identifying this, which has also been identified by Reviewer 3. Our abstract previously ended: *" our methods allow 20% of landslides to be timed with an accuracy of 80%. This will allow multi-temporal landslide inventories to be generated for long rainfall events such as the Indian summer monsoon, which triggers large numbers of landslides every year and has until now been limited to annual-scale analysis."*

We will change this to *" " our methods allow 20% of landslides to be timed with an accuracy of 80%. . Application of our methods could provide an insight on landslide timings throughout events such as the Indian summer monsoon, which triggers large numbers of landslides every year and has until now been limited to annual-scale analysis"* . This removes the words *"multi-temporal landslide inventories"*, which were misleading since we cannot provide a complete inventory where all the landslides have timings assigned.

However, while this is a drawback, we do not believe that it renders the work unsuccessful. First, landslide inventories often include thousands of landslides, meaning that by timing 20% of these, we obtain landslide timing information on a statistically useful number of landslides. Similarly to the large number of studies deriving rainfall threshold our methods would allow us to constrain the rainfall characteristic which have preceded the triggering of landslides (even with a time window is of 6-12 days, at the scale of the monsoon it would be a substantial improvement), or to determine whether a subsample of monsoon-induced landslides display spatio-temporal clusters which could be interpreted in terms of triggers.

**This being said, one could consider this paper as a step towards developing better tools along this research direction and in this regard, could be still valuable. And yet, authors do not clearly present their work. Unfortunately, the manuscript is not well written. I had to read some parts more than once to understand the authors' point. The figures are not well designed either. I have many comments that I hope the authors find useful to improve their work."**

We will take on board the individual comments throughout the manuscript on the text and the figures, and hope these will make the manuscript clearer.

**"Last but not least, I would like to test the code/tool they developed but unfortunately, it is not available. This is a preprint with DOI, so I did not really get why it was not shared already."**

The code will be provided as a supplement to the next version of the manuscript as requested. I apologise for omitting it, I was not aware it was necessary at this stage.

Then, later in the review, we identify the following major points to be addressed (Omitting smaller corrections to the text and figures, which we will address in a later, comprehensive response).

**Line 70: "heavy rainfall event which took place from 28 June to 9 July 2018" I guess these are the dates that they were able to acquire pre- and post- event images to map landslides, right? But was the study area also exposed to heavy precipitation during the entire period? It would be useful to see the amount of precipitation (as time series) that each of your study areas received during the periods under consideration.**

No, these are not the dates of the acquired images, they are the days between which there was heavy rain over Hiroshima as specified by Hashimoto et al. (2020).
We will attach rainfall time series (NASA GPM product) for Hiroshima and Zimbabwe as a supplement so that the extents of the event are visible. We agree this would help with visualising the timelines of the events.

**Line 77: I thought you focus on rainfall-triggered landslides as also you indicated in the title of your manuscript. Why do you use the co-seismic landslide inventory of Roback et al. (2018)?**
**Ok, now I understand what you have done. You removed landslides triggered by the mainshock and work with the others. However, you do not know if these landslides were triggered by the aftershocks or rainfall events. Also, it is not clear when they were triggered. The confusing thing is you mentioned that you focus on "inventories of landslides whose timings are known a-priori to test and develop landslide". However, this does not one of those inventories.**
This is not correct.
As stated at line 87 "*we also removed all landslides whose trigger was specified by Roback et al. (2018) to be something other than the mainshock*" i.e. we remove landslides triggered by aftershocks or by rainfall and keep only the landslides triggered by the mainshock. Since the sentence was not clear before, we will change to "*we also removed all landslides specified by Roback et al. (2018) to have been triggered by an aftershock or by rainfall, and use only those triggered by the mainshock in our analysis*."
For landslides triggered by the mainshock, their timing is known, which should resolve the confusion here. We detail below why we test our methods not only on well-timed rainfall triggers (Hiroshima and Zimbabwe) but also on the Nepal earthquake.

**Why do not you simply pick another rainfall-induced landslide event inventory?**
The reason we do not use a rainfall-triggered landslide inventory here is that Nepal is a country for which the methods we develop here will be especially useful due to its extensive annual monsoon-triggered landsliding and long periods of cloud cover.
It is also particularly challenging for SAR applications due to its steep topography, which results in distortion of the SAR images. It is thus particularly important to test the methods in this environment. Unfortunately, rainfall-triggered landslide inventories of known timing are not available here, so we use an inventory of earthquake-triggered landslides instead.
This is stated in lines 78-81 of the manuscript and will be emphasized more in the revisions.

**Lines 81-83: "since the inventory of Roback et al. (2018) covers a large area, with different areas having different Sentinel-1 coverage, we focused on triggered landslides within three large valleys" This does not explain why you focus on these three rectangular areas (actually one of them has a weird shape). You can cover a larger area mapped by**
**Roback et al. (2018) and one Sentinel 1 image should be covering at least an area covering two of those rectangles. So how did you identify these rectangles really?**
The valleys were selected since Marc et al. 2019a have mapped landslides in these valleys during several monsoons following the earthquake and therefore will be of interest for future studies.

In fact, the Sentinel-1 time series is different for each Valley, making it complicated to combine these (See Fig.1). Although tracks 19 and 85 cover both Trishuli and Buri Gandaki, track 85 is much more

complete over Trishuli, which is further South than Buri Gandaki, which is further north. The two valleys lie in the same track but within different scenes and several scenes are missing on track 85 for the Buri Gandaki Valley. This is probably because the satellite had not long been operational at the time of the earthquake. It is therefore not possible to combine them into a single time series. I attach a map below to illustrate this. Scenes on the descending track (19) are shown in green. In this case, BG and Tr are in the same scene (BK is outside and belongs to track 121 instead). On track 85 (shown in orange), BG and BK are in different scenes, while Tr lies in an area where the two overlap. Since data for Tr can be supplied from either the northern or southern scene, it has a more complete time series than either BG or BK. It is for this reason that data from track 85 is only used in Trishuli, while BK and BG, we are limited to the descending tracks (121 and 19 respectively).

[Figure]

Finally, the squares on Figure 1c do not show the exact extents of the inventories but show the locations instead, as in figures 1a and 1b, where the study areas are also not square. In order to be clearer, we will replace the squares, which are not particularly informative, with plots of landslide density for each event. This should make the extents of each inventory clearer.

**It is not clear how did you define pre-, co- and post- event image acquisitions. Do you explain this later in the method section? But you already refer to the term "co-event pair" in line 91, so the reader needs to know what it means. For instance, you indicate that in the Hiroshima case the heavy rainfall event occurred between 28 June and 9 July. Therefore,**
**landslides were triggered (or mapped, as I mentioned this is confusing anyway) during this 10-day period. And you also mention that landslides were most likely triggered between 6 and 7th of July. Then why do you have such a large time period for "co-event pair"**
I apologise, we have generated confusion by using co-event (no quotation marks) to refer to the true pair of images spanning the landslide timing (as in line 91) while using "co-event" (with quotation marks) to refer to a "co-event" period that we have defined for testing the methods (i.e. a series of images that contains the correct co-event pair). Evidently, these are too similar and we will change to an alternative. Possibly "true co-event pair" and "designated co-event series".

**Line 92: "these two earthquakes can be considered as a single triggering event in Bhote Kosi" You cannot consider them as a single event. However, if you cannot differentiate landslides possibly triggered by different factors for a given period of time, this needs to be indicated as a source of uncertainty in your analyses. I do not know what could be the consequences, but obviously, this needs to be discussed later on in the manuscript.**
While it is true that in terms of the spatial distribution of landsliding, we cannot consider them as a single trigger, that is not the aim of this study. Here we are trying to identify the timing of a landslide whose location we already know. Both the mainshock on 25 April and the aftershock on 12 May lie

between our SAR acquisitions on 24 April and 18 May, it does not matter if the landslide happened at the time of mainshock or the aftershock, an assigned timing of 24 April – 18 May is right in either case.

I think this confusion has again arisen from confusion between our defined "co-event" 6 month time series and the correct co-event image pair. (i.e. the same as at line 91). As previously mentioned, we will be more careful to distinguish between these two in the next version of the manuscript

**Lines 160-166: Based on what you explained here how we should interpret Figure 2c? "A step change in the difference between the median landslide amplitude and the median background amplitude is then used as an indicator of landslide timing." Based on your interpretation, could you point out the timing of the landslide in Figure 2c. Which one is a step increase or a step decrease? Other than the signal received from the shadow area, I do not see any significant change in overall fluctuations of amplitude values associated**
**with rainfall events (indicated by the blue bar in fig2c).**

The step change is observed in the difference between the landslide and background time series. It is true that it is hard to see when considering the median landslide amplitude on its own. In Figure 2c, you can see that the distance between the orange and teal lines increases after the rainfall event (when the landslide happened). However, it is true that is not the clearest way to display this. We will redesign this figure incorporating the plots below, which shows the time series for each method.

[Figure]

The time series above are for a different landslide to the example shown in the original manuscript (One from the descending track 19 SAR time series over Trishuli, Nepal). The landslide selected in the previous manuscript version demonstrated Methods 1 and 3 well, but for Method 2, the time step change was not clearly visible. This was raised in the next comment from the reviewer:

**Lines 168-171: The same comment as above, please explain how you interpret Figure 2d. I do not see a specific change in the trend associated with the blue bar other than some fluctuations.**

Therefore, we have decided to change to a different landslide polygon for which this trend is clearer.

We also add the convolution functions (in grey on the new figure), which demonstrate how the peak / trough in the function corresponds to a step change in the landslide timing detection method. This peak / trough is how we select the step change.

**Line 198: "The step function was made up of a series of -1s and 1s of twice the length of the co-event time series"** **Do twice the length of the co-event time-series means 12 months? And why?**

This does not mean 12 months, it merely means twice the number of points in the SAR times series. (e.g. for a series of 12 SAR images, we convolve with a series made up of 12 -1s followed by 12 1s i.e. a series of length 24). The output of this convolution is a series the same length as our SAR time series with a peak at the location that best agrees with the step function. In this way, we can automatically detect the location of a step change in the SAR measurements.

**Can't you make a figure to explain what you have done at this step?**

We hope that by adding the convolution functions in the above figure, it will be clearer how this process works and how the peak / trough in the convolution corresponds to the step increase / decrease in the landslide timing method.

---

## Author Comment (AC2)

**Example landslide 1:** Landslides triggered by heavy rain in Hiroshima, 2018 (Image © 2021 Google, Maxar technologies). (b) Difference between pre-event and post-event $\gamma_0$ for one landslide polygon (c) $\gamma_0$ change for shadow pixels selected from within a 10 m buffer of the landslide polygon. (d) absolute value of ($\gamma_0 - $ mean($\gamma_0$)) before and after the landslide (showing increased pixel variability) (e) $\gamma_0$ change for background pixels selected from between 30 m and 500 m from the landslide polygon. Landslide polygons (grey) from Emberson et al. (2021).

[Figure]

**Example landslide 2:** Landslides triggered by heavy rain in Zimbabwe, 2019 (Image © 2022 Google, CNES Airbus Maxar technologies). (b) Difference between pre-event and post-event $\gamma_0$ for one landslide polygon (c) $\gamma_0$ change for shadow pixels selected from within a 10 m buffer of the landslide polygon. (d) absolute value of ($\gamma_0$ – mean($\gamma_0$)) before and after the landslide (showing increased pixel variability) (e) $\gamma_0$ change for background pixels selected from between 30 m and 500 m from the landslide polygon. Landslide polygons (grey) from Emberson et al. (2021).

---

## Author Comment (AC3)

Preliminary response to reviewer 3

We thank the reviewer for taking the time to review our paper. We will prepare a full response to all comments made on the manuscript later on in the review process, but here we provide a preliminary response to some of the major points raised in this review (reviewer comments in bold)

**Generally, I suggest to report the results more as a potential contribution towards using Sentinel-1 data for estimating time-windows of event occurrence. Some parts of the manuscript read as if a well-working method is presented that works generically for identifying timings of landslides. However, this is still very much work in progress. For instance, I don't think that *"This will allow multi-temporal landslide inventories to be generated for long rainfall events such as the Indian summer monsoon"* in a comprehensive manner.**

We will revise the manuscript to make it clearer throughout that our methods cannot establish timings for all the landslides in an inventory (and in fact, will only provide timings for ~20%).
The quote here is taken from the abstract. In response to this and to similar comments made by reviewer 2, we will change the end of the abstract.
Previous text: *"our methods allow 20% of landslides to be timed with an accuracy of 80%. This will allow multi-temporal landslide inventories to be generated for long rainfall events such as the Indian summer monsoon, which triggers large numbers of landslides every year and has until now been limited to annual-scale analysis."*
New text: *" our methods allow 20% of landslides to be timed with an accuracy of 80%. Application of our methods could provide an insight on landslide timings throughout events such as the Indian summer monsoon, which triggers large numbers of landslides every year and has until now been limited to annual-scale analysis"* .
This removes the words "multi-temporal landslide inventories", which were misleading since we cannot provide a complete inventory where all the landslides have timings assigned.
We will make also make sure this is clear in the discussions / conclusions and throughout the manuscript.

We will also make sure it is clear throughout that we are not assigning specific dates, but instead time windows of (in most cases) 12 days to each landslide.

**There will definitely be a biases in terms of identified slides,**
The biases towards which slides can be assigned a timing is related to section 3.3 "Factors effecting the performance of each method." For example, it is clear that larger landslides are more likely to be assigned a timing than smaller landslides using our methods (Fig. 5 d-f).
In the manuscript, we considered how this effected where our methods could be applied (for example, they will not work well for inventories of small landslides or in arid environments), but we did not consider how well the timings of the 20% of landslides we assign a timing to using SAR methods will represent the full inventory. In fact, this 20% will be biased to contain a higher proportion of large landslides and landslides in more heavily vegetated areas than the original inventory. We will add this point to the revised version manuscript

**a vast majority of sildes will be missed or - worse - labelled incorrectly,**
Our methods should not incorrectly label a large percentage of slides. We expect that if we apply our methods to an inventory of rainfall-triggered landslides, we will obtain an inventory in which ~80% have no timing information, ~16% are correctly timed and ~4% are incorrectly timed. We will be careful to make this clearer in the abstract, results and conclusions sections of the revised manuscript.

and things might look dire when thinking beyond the scope of this study, e.g. if no polygons are availabe.

If polygons are not available for an event, it will not be possible to apply our methods in their current form – they are not designed to be applied to events for which we do not have a pre-existing landslide inventory. We will ensure this is clear in the revised version of the manuscript.

**It took me a while till I figured out the meaning of the terminology you used for the orbit IDs (e.g. "H083A"). Please specify more clearly that this is a combination of study area, orbit number and orbit direction.**

Yes, this was also raised by reviewer 2, who suggested a change of terminology from e.g. "H083A" to "Hiroshima_asc" to describe the tracks. We will make this change, which should also resolve this comment.

**"We tested both of these polarisations, but found VV to perform better than VH so present only the results for VV." This is an interesting finding. How was this evaluated?**

Yes it is interesting since other studies (e.g. Handwerger et al. 2022) use VH and find it to be the most successful option. We attach a version of table 2 in the original manuscript which contains the same results from VH in Zimbabwe and Hiroshima (VH data are not available for the Nepal case study, which occurred soon following the launch of Sentinel-1 before dual-pol data began to be consistently acquired – this would be a further disadvantage to using VH).

| | Hiroshima | | Zimbabwe | | Trishuli | | Buri Gandaki | Bhote Kosi | | Hiroshima (VH) | | Zimbabwe (VH) | |
|---|---|---|---|---|---|---|---|---|---|---|---|---|---|
| Track | H090D | H083A | Z079D | Z072A | Tr019D | Tr085A | BG019D | BK121D | | H090D | H083A | Z079D | Z072A |
| Total landslides | 543 | | 383 | | 650 | | 922 | 1554 | | 543 | | 383 | |
| non-masked | 543 | 540 | 383 | 383 | 485 | 474 | 592 | 894 | | 543 | 540 | 383 | 383 |
| ls-b inc | 137 (26%) | 92 (36%) | 90 (50%) | 66 (41%) | 106 (38%) | 107 (35%) | 152 (36%) | 313 (36%) | | 94 (12%) | 97 (29%) | 40 (20%) | 26 (12%) |
| ls-b dec | 126 (38%) | 205 (57%) | 182 (23%) | 155 (34%) | 156 (37%) | 143 (22%) | 113 (27%) | 310 (32%) | | 182 (25%) | 251 (63%) | 262 (28%) | 266 (36%) |
| pix var | 160 (45%) | 181 (59%) | 134 (50%) | 83 (47%) | 141 (42%) | 125 (44%) | 141 (30%) | 261 (43%) | | 155 (32%) | 222 (56%) | 152 (16%) | 110 (27%) |
| shadow | 79 (49%) | 122 (75%) | 43 (55%) | 58 (72%) | 45 (80%) | 50 (80%) | 17 (88%) | 52 (87%) | | 144 (43%) | 227 (74%) | 125 (39%) | 140 (47%) |
| combined >2m | 51 (75%) | 99 (88%) | 48 (65%) | 39 (85%) | 45 (76%) | 33 (88%) | 33 (64%) | 86 (63%) | | 71 (54%) | 166 (80%) | 88 (33%) | 87 (60%) |
| combined >3m | 10 (80%) | 22 (100%) | 5 (80%) | 2 (50%) | 8 (100%) | 3 (100%) | 1 (100%) | 5 (100%) | | 9 (89%) | 38 (97%) | 17 (35%) | 9 (89%) |
| asc & desc | 122 (80%) | | 81 (73%) | | 70 (81%) | | - | - | | 196 (69%) | | 142 (44%) | |
| asc & desc 2m,1t | 79 (73%) | | 64 (72%) | | 52 (77%) | | | | | 139 (65%) | | 102 (23%) | |
| asc & desc 3m | 43 (91%) | | 17 (76%) | | 18 (94%) | | - | - | | 57 (81%) | | 40 (75%) | |
| baseline (1/n) | 7% | 17% | 10% | 7% | 8% | 8% | 8% | 14% | | 7% | 17% | 10% | 7% |

As you can see, we have a generally lower accuracy for VH data, especially in Zimbabwe. Since this may be interesting for future studies, we can include it as a supplement.

**The copernicus DEM would have been a more recent DEM version, also available globally at a resolution of 30 m.**

We chose to use the SRTM DEM here since it is already available as a dataset in Google Earth Engine, making it easier to integrate into the slope correction module for future users.

**l. 161/Figure 2: "A step change in the difference between the median landslide amplitude and the median background amplitude is then used as an indicator of landslide timing." It might be beneficial to plot this difference?**

Yes, this is a good point. In response to this and to similar comments from reviewer 2, we have prepared 3 panels (below) which will be incorporated into Figure 2. This makes the step change clearer (Note we have also changed to a different landslide polygon due to changes requested by reviewer 2 – these time series are thus not comparable to those in Figure 2 of the original manuscript).

These panels also show as a grey line the convolution between the method and a step function to make it clearer how a step change results in a peak or trough in the convolution function.

[Figure]

**Overall, appropriate performance metrics and their interpretation is of key importance. In fact, when thinking about the implications of the method presented here, this is crucial. If no validation data are available (e.g. when this method is applied to a new data set), a vast majority of identified dates (more precise: time windows) will be incorrect. This needs to be discussed.**
The problem is not that a vast majority of landslides will be assigned an incorrect time window, but instead that the majority of landslides will not be assigned any time window at all. The subpart of the inventory that has timings assigned will be biased towards larger landslides and those in more vegetated areas. We will discuss this in the revised version of the manuscript. (See response to earlier comments.)

Of those landslides that are assigned a time interval, we expect that 80% of the time, this interval should be correct. Therefore if, in a new dataset, we observe a spatio-temporal cluster of landslides, we can assume the timing of this cluster is correct, since it is very unlikely that all the timed landslides in this cluster would be assigned the same incorrect date.

**Publishing the code (e.g. on GitLab/GitHub) would be welcome for the final manuscript, but also of interest from a reviewer's perspective. If there are concerns with respect to sharing code before the publication is accepted, there are surely opportunities for embargos.**

The GEE and python codes will be shared as a supplement to the next version of the manuscript.

---

## Author Comment (AC4)

**Example landslide 1:** Landslides triggered by heavy rain in Hiroshima, 2018 (Image © 2021 Google, Maxar technologies). (b) Difference between pre-event and post-event $\gamma_0$ for one landslide polygon (c) $\gamma_0$ change for shadow pixels (red) and bright pixels (blue) selected from within a 10 m buffer of the landslide polygon. (d) absolute value of $(\gamma_0 - mean(\gamma_0))$ before and after the landslide (showing increased pixel variability) (e) $\gamma_0$ change for background pixels selected from between 30 m and 500 m from the landslide polygon. Landslide polygons (grey) from the Association of Japanese Geographers (2019)

[Figure]

**Example landslide 2:** Landslides triggered by heavy rain in Zimbabwe, 2019 (Image © 2022 Google, CNES Airbus, Maxar technologies). (b) Difference between pre-event and post-event $\gamma_0$ for one landslide polygon (c) $\gamma_0$ change for shadow pixels (red) and bright pixels (blue) selected from within a 10 m buffer of the landslide polygon. (d) absolute value of ($\gamma_0 - \text{mean}(\gamma_0)$) before and after the landslide (showing increased pixel variability) (e) $\gamma_0$ change for background pixels selected from between 30 m and 500 m from the landslide polygon. Landslide polygons (grey) from Emberson et al. (2021).

[Figure]

**Example landslide 2:** Landslides triggered by heavy rain in Zimbabwe, 2019 (Image © 2022 Google, CNES Airbus). (b) Difference between pre-event and post-event $\gamma_0$ for one landslide polygon (c) $\gamma_0$ change for shadow pixels (red) and bright pixels (blue) selected from within a 10 m buffer of the landslide polygon. (d) absolute value of $(\gamma_0 - \text{mean}(\gamma_0))$ before and after the landslide (showing increased pixel variability) (e) $\gamma_0$ change for background pixels selected from between 30 m and 500 m from the landslide polygon. Landslide polygons (grey) from Emberson et al. (2021).

---

## Author Comment (AC5)

Response to reviewer 2

We thank the reviewer for taking the time to read and evaluate our manuscript. The following is a line by line response to their comments (Reviewer comments in bold type).

**In this paper, the authors propose a SAR-based technique to estimate the possible time-window of landslides mapped as a part of seasonally generated inventories. To test their methods, they use two rainfall-triggered landslide event inventories and one post-seismic inventory including landslides that might have been triggered by the aftershocks of the 2015 Gorkha earthquake and/or rainfall events that occurred following the event. In this context, I should stress that the paper focuses on an interesting research question for sure and it appears as a nice fit for the journal and, in particular, for this special issue.**

**However, the authors are able to come up with a time estimation only for 20% of landslides with an accuracy of 80%. Therefore, I doubt if this is successful research in the end. Frankly speaking, I am not sure and just hesitating to say that the results are promising. However, what I can say is the output of this research is not fulfilling what the authors are promising in the abstract/conclusions.**

First, we will revise the abstract and conclusions to make sure that it is clear that our method will only provide landslide timings for around 20% of the landslides in an inventory and that we do not promise a full inventory with timings for all landslides. This was not previously clear and we thank the reviewer for identifying this, which has also been identified by Reviewer 3.

Our abstract previously ended: " *our methods allow 20% of landslides to be timed with an accuracy of 80%. This will allow multi-temporal landslide inventories to be generated for long rainfall events such as the Indian summer monsoon, which triggers large numbers of landslides every year and has until now been limited to annual-scale analysis.*" We will change this to "*our methods allow 20-30% of landslides to be timed with an accuracy of 80%. Application of our methods could provide an insight on landslide timings throughout events such as the Indian summer monsoon, which triggers large numbers of landslides every year and has until now been limited to annual-scale analysis*" . This removes the words "multi-temporal landslide inventories", which were misleading since we cannot provide a complete inventory where all the landslides have timings assigned.

Similarly in the conclusions section, we will change "*These methods will allow us to generate multi-temporal landslide inventories for long rainfall events, unlocking comparisons between rainfall data, hydrological models and triggered landsliding.*" to "*These methods will provide information on the timings of some individual landslides and allow spatio-temporal clusters of landslides to be associated with peaks in rainfall during long rainfall events, unlocking comparisons between rainfall data, hydrological models and triggered landsliding.*"

However, while this is a drawback, we do not believe that it renders the work unsuccessful. First, landslide inventories often include thousands of landslides, meaning that by timing 20% of these, we obtain landslide timing information on a statistically useful number of landslides. Similarly to the large number of studies deriving rainfall threshold our methods would allow us to constrain the rainfall characteristics which have preceded the triggering of landslides (even with a  time window of 6-12 days, at the scale of the monsoon it would be a substantial improvement), or to determine whether a subsample of monsoon-induced landslides display spatio-temporal clusters which could be interpreted in terms of triggers.

We have also made some adjustments to the methods, which somewhat improve the number of landslides for which we can assign a timing from from 20% to 30% in favourable cases. This does not change the overall message of the manuscript, since we keep the same 80% accuracy and still do not assign a timing for the majority of the landslides in an inventory, but it represents an improvement to the method. All following replies and alterations to the text therefore incorporate these new sensitivity levels.

(i)       A new method based on bright spots observed at the edges of landslide polygons

As described in Section 2.4.3 of our original manuscript, shadows are cast by trees at the edges of landslide scars due to the imaging geometry of the SAR sensor. (See figure below, altered from Figure 2b of the original manuscript). On the opposite side of the scar, we may observe a bright patch in the SAR amplitude

images due to double bounce scattering between the exposed soil and trees on the far side of the landslide scar, and the focussing of the microwave energy into a small area (Villard and Borderies, 2007). Similarly to the shadow method, we compare the pre-event and post-event SAR time series and identify pixels which have experienced a strong increase in amplitude (we found a threshold of >5dB to perform best in this case). Incorporating this method means that, increases the final number of landslide that are assigned a timing by 3% in Hiroshima, 2% in Zimbabwe and 7% in Trishuli.

A description of this method will be added to Section 2.4 (SAR amplitude techniques for landslide timing) as " 2.4.4 Method 4: Geometric bright spots"

[Figure]

 (ii)  Increase in size of the landslide polygons

As described at lines 185-195 and 317-326 of our original manuscript, there may be a spatial mismatch between the optically-derived landslide polygons and the SAR imagery. Previously, we increased the size of landslide polygons by 10m (1 SAR pixel) for the geometric shadows method to try to account for this. However, this effect may also affect the results from other methods, and decrease the number of landslides for which we are able to assign a time window.

Therefore, for landslides that are not assigned a date by at least 2/4 of our methods, we now increase the size of the landslide polygon using a 20 m  buffer for all methods and repeat, with the aim of trying to assign landslide timings to some of the landslides that experience this spatial mismatch. This improves the number of landslides we are able to detect in each case study event by 4% in Hiroshima and 5% in Zimbabwe and Trishuli, and will be added to the next version of the manuscript.

Since this will now be described in its own subsection in the methods section, text referring to this at lines 185-195 and 317-326 of our original manuscript will be removed.

**This being said, one could consider this paper as a step towards developing better tools along this research direction and in this regard, could be still valuable. And yet, authors do not clearly present their work. Unfortunately, the manuscript is not well written. I had to read some parts more than once to understand the authors' point. The figures are not well designed either. I have many comments that I hope the authors find useful to improve their work.**

We will take on board the individual comments throughout the manuscript and the figures and hope that by addressing these, the manuscript will be made clearer.

**Last but not least, I would like to test the code/tool they developed but unfortunately, it is not available. This is a preprint with DOI, so I did not really get why it was not shared already.**

The code will be provided as a supplement to the next version of the manuscript as requested. I apologise for omitting it, I was not aware it was necessary at this stage.

**Overall, I recommend a rejection to give adequate time to the authors for a comprehensive revision for the manuscript for clarity, pulling some of the speculation and assumptions to the discussion, adding more definitions of terms, and framing the paper in hypotheses. This should help the reader**

**understand what you did and why you did it. Because in the current version, authors do not really help the reader to find their way through the manuscript.**

**Below I've included line-by-line suggestions and highlighted all these points.**

**Line 18: "to emergency response coordinators". I would say rainfall-induced landslide inventories are rarely used by emergency response coordinators as they are generated at least weeks or months after an event. But if you are referring to some kind of indirect usage of the dataset (for instance, as an input to develop a landslide early warning system or something) please be more specific.**

**Line 18: "physical and empirical" there are also statistically-based models exploiting the very same dataset**

**Lines 18-19: Could you please cite relevant literature.**

We have added references to Jones et al., 2021, (estimate of eroded sediment volumes); Ozturk et al., 2021 (landslide polygons used to train and validate regression models) ; Wu et al. 2015 (physical model of rainfall-triggered landsliding tested on optically-derived landslide inventory)

Jones, J.N., Boulton, S.J., Bennett, G.L., Stokes, M. and Whitworth, M.R., 2021. Temporal variations in landslide distributions following extreme events: Implications for landslide susceptibility modeling. *Journal of Geophysical Research: Earth Surface*, *126*(7), p.e2021JF006067.
Ozturk, U., Saito, H., Matsushi, Y., Crisologo, I. and Schwanghart, W., 2021. Can global rainfall estimates (satellite and reanalysis) aid landslide hindcasting?. *Landslides*, *18*(9), pp.3119-3133.
Wu, Y.M., Lan, H.X., Gao, X., Li, L.P. and Yang, Z.H., 2015. A simplified physically based coupled rainfall threshold model for triggering landslides. *Engineering geology*, *195*, pp.63-69.

**Line 20: "the size location" the size, location**
This typo will be corrected in the revised manuscript.

**Line 22: "the size and location" the size, location and timing too. As you said occurrence dates of landslides could not be accurate in some cases via optical images but also, as you said, if we have cloud-free images it is doable.**
Optical images *always* be used to get info. on the size and location of landslides, but the timing is not always constrained because of cloud cover, as we go on to describe in the following sentences. We feel it would be confusing to include timing in the list here because it would immediately be contradicted in the following sentence.
**Line 24: "Williams et al., 2018; Robinson et al., 2019". These examples are from earthquake-triggered landslide events. But you focus on rainfall-triggered landslides. So, please replace them with some examples of rainfall-triggered landslide events.**

It is true these papers focus on earthquake-triggered landslides, but they both focus on areas that also experience frequent rainfall-triggered landsliding. Therefore observations on the usability of optical satellite images for earthquake triggered landslides are also relevant for rainfall-triggered landslides. Furthermore, while we focus on rainfall events, the methods we propose could be applied to any event where the timing of triggered landslides is unknown due to cloud cover. This also includes sequences of earthquakes and storms, for example Lombok, Indonesia (Ferrario, 2019), Papua New Guinea, (Tanyas et al. 2022) and Gorkha, Nepal (Martha et al. 2017).

We will add a sentence on this at line 34 of the original manuscript.

New references:

Ferrario, M.F., 2019. Landslides triggered by multiple earthquakes: insights from the 2018 Lombok (Indonesia) events. *Natural Hazards*, *98*(2), pp.575-592.

Tanyaş, H., Hill, K., Mahoney, L., Fadel, I. and Lombardo, L., 2022. The world's second-largest, recorded landslide event: Lessons learnt from the landslides triggered during and after the 2018 Mw 7.5 Papua New Guinea earthquake. *Engineering Geology*, *297*, p.106504.

**Lines 28-29: This line needs to be rewritten. Also, why did you prefer the term "landfall", why not "landslide"**

A typhoon 'making landfall' refers to the time when the storm moves over land after having developed over the ocean. We are not using 'landfall' as a synonym for 'landslide'. With this, the sentence no longer needs to be rewritten.

**Lines 30-31: Is that the case? Landslides triggered by each of those typhoons were mapped separately or not? It is not clear from the line if you indicate what already happened or this is just a hypothetical remark.**

We will change *"If no cloud-free optical satellite imagery is acquired between these trigger events"* to *"If no cloud-free optical satellite imagery is acquired between such successive trigger events"* to make it clear this is hypothetical.

**Lines 33-34: "This limits analysis of these landslides to the annual scale (e.g. Marc et al., 2019a; Jones et al., 2021)." But, for instance, Marc and others generated monsoon-induced landslide inventories and to do that you just need pre- and post- monsoon images. So you do not need cloud-free optical satellite images through the monsoon. Please remove this reference and also please be more specific about the limitations of generating seasonal landslide inventories.**

Since these multi-temporal inventories are compiled from images before and after each monsoon season, we can only associate them with a given monsoon season and not with any particular peak in rainfall within that monsoon season. In this case, annual and seasonal almost mean the same thing, since almost all the landslide occurring annually in Nepal occur during the monsoon season. To make this clearer, we will change *« This limits analysis of these landslides to the annual scale »* to *« This limits analysis of these landslides to the seasonal scale and prevents association of individual landslides or spatio-temporal clusters of landslides to specific peaks in rainfall.*

**Line 35: "Current alternative methods of landslide timing are generally not widely applicable." Please rewrite this line, is not clear what you mean. What are those alternative methods? And why do you think they are not widely applicable (any reference for this?). You haven't said anything about any alternative methods yet. Please first describe them and then you can evaluate those methods based on the literature.**

The evidence for this statement comes in the following sentences. We will restructure this paragraph so that this sentence is given as a conclusion at the end. To address this and the following comments on lines 36 and 43, we will rewrite this paragraph, including more recent references :

Previous text: *Current alternative methods of landsliding are generally not widely applicable. Landslides that occur close to inhabited areas or that damage important pieces of infrastructure may be described in news reports or social media (e.g. Kirschbaum et al. 2010) Information on the timing of such landslides can also be generated through interviews with local residents (Bell et al. 2021). Rainfall intensity-duration thresholds have previously been derived for landslides dated in this way (e.g. Dahal and Hasegawa, 2008) and for landslides whose timings and properties are known through monitoring and field surveys (e.g. Guzzetti et al. 2007 ; Ma et al., 2015). However, this is unlikely to be the case for the majority of landslides in an inventory, and will be biased towards populated areas. Seismic recordings of landslides can also provide highly precise information on their timings, but will mostly record large landslides and require multiple seismic stations to allow timing of an individual, localised landslide (e.g. Yamada et al. 2012; Hibert et al. 2019)*

New text: *Landslides that occur close to inhabited areas or that damage important pieces of infrastructure may be described in news reports or social media (e.g. Kirschbaum et al. 2010; Franceschini et al. 2021) Information on the timing of such landslides can also be generated through interviews with local residents (Bell et al. 2021) and through citizen science initiatives (Sekajugo et al. 2022). Rainfall intensity-duration thresholds have previously been derived for landslides in this way (e.g. Dahal and Hasegawa, 2008) and for landslides whose timings and properties are known through monitoring and field surveys (e.g. Guzzetti et al. 2007 ; Ma et al., 2015). However, such information on landslide timing is unlikely to be available for the majority of landslides in an inventory, and is usually biased towards populated areas and areas accessible by road (Sekajugo et al. 2022). Seismic recordings of landslides can also provide highly precise information on their timings, but will mostly record large landslides and require multiple seismic stations to allow timing of an individual, localised landslide (e.g. Yamada et al. 2012; Hibert et al. 2019). Current methods of obtaining landslide timing information in the absence of cloud-free optical satellite images are therefore not widely applicable.*

**Line 36: But this is not the method Kirshbaum and others or if you take a look at more recent literature  Franceschini and others (DOI 10.1007/s10346-021-01799-y) used, this is the source information for them. Please tell us the method they used.**

See changes to text above

**Line 43: "will" Why did you switch to the future tense**

See changes to text above.  *"will be biased"* changed to "*is usually biased*"
**Line 49. Please put a full stop before giving the example.**

A full stop is not needed here as this is all one sentence. However, it was broken up by our placing a list of references halfway through. These will be move to the end of the sentence

Previous text : "*Numerous studies have shown that SAR data can be used to detect the spatial distribution of landslides in the case where their timing is already known (e.g. Aimaiti et al. 2019; Burrows et al. 2019, 2020; Ge et al. 2019 ; Konishi and Suga 2019 ; Masato et al., 2020 ; Mondini et al. 2021 ; Yun et al. 2015) , for example in the case of earthquake-triggered landslides."*

New text :  "*Numerous studies have shown that SAR data can be used to detect the spatial distribution of landslides in the case where their timing is already known, for example in the case of earthquake-triggered landslides (e.g. Aimaiti et al. 2019; Burrows et al. 2019, 2020; Ge et al. 2019 ; Konishi and Suga 2019 ; Masato et al., 2020 ; Mondini et al. 2021 ; Yun et al. 2015)."*

**Line 57: "timed landslide information" this is the first time that I have heard this term and it sounds weird, please rephrase it. And please do it not only here but through the manuscript.**

We will change "*timed landslide information"* to "*information on landslide timing"*

**Line 61: "three potential landslide timing methods" you haven't said anything about these methods yet, so it is not clear what these methods are.**
To improve clarity, we will change "*We use Sentinel-1 time series over inventories of landslides whose timing is already known to test three potential landslide timing methods individually and in combination"*  in the previous version of the manuscript to "*We present four methods to constrain landslide timing using Sentinel-1 SAR time series and test these on inventories of landslides whose timing is already known."* in the new manuscript version

**Line 63: "Case study events" does not sound right. Please revise it. e.g., Case studies or Landslide inventories**
We will change this to « Case studies »
**Line 66: Why did you take 20 pixels as your threshold? Why not 10? It could be better to do it without any filtering first. And then, you can identify the threshold for the landslide size that your method works well.**

Statistics we use, such as the standard deviation of pixels within the landslide, require the landslide to contain multiple SAR pixels. As shown in Figure 5d-f (original manuscript) larger landslides are both more likely to be assigned a time window, and more likely for this time window to be correct. Furthermore since landslide-area distributions obey a power law, decreasing the threshold to 1000m$^2$ would greatly increase the number of landslides required to be processed.

**Line 68: "inventories of landslides" landslide inventories. Btw, you do not really need to cite Emberson et al. (2021) for the Hiroshima inventory because it was already available, right?**

We will change « *inventories of landslides* » to « *landslide inventories* »

The Hiroshima inventory was not mapped by Emberson et al, but it is included (along with the Zimbabwe inventory) in the landslide inventories analysed in that study. The reference to Emberson et al. here (line 68 of the original manuscript) refers to both of these inventories. However, elsewhere in the manuscript, we will change the reference for the Hiroshima inventory from Emberson et al. (2022) to the association of japanese geographers, 2019 (e.g. at Line 414 of original manuscript)

**Line 70: "heavy rainfall event which took place from 28 June to 9 July 2018" I guess these are the dates that they were able to acquire pre- and post- event images to map landslides, right? But was the study area also exposed to heavy precipitation during the entire period? It would be useful to see the amount of precipitation (as time series) that each of your study areas received during the periods under consideration.**

No, these are not the dates of the acquired images, they are the days between which there was heavy rain over Hiroshima as specified by Hashimoto et al. (2020). We will attach rainfall time series (NASA GPM product) for Hiroshima and Zimbabwe as a supplement so that the extents of the event are visible. We agree this would help with visualising the timelines of the events.

**Line 74: "Planetdove" Planet Scope?**

Yes, Planet Dove are the satellites, PlanetScope is the name of the constellation so it is better to use this. Thank you for correcting the mistake.

**Lines 74-75: "images acquired on 20 and 24 March….the majority of landsliding occurred between the 15-17 March" You mean, they did not examine pre and post images to identify landslides solely triggered by the rainfall event, is this correct?**

Apologies, this was not clear, we should have specified that the images acquired on the 20th and 24th of March were post-event images.

"*This inventory was compiled as part of the study of Emberson et al. (2021) using Planetdove optical satellite images acquired on 20 and 24 of March*" at line 74 of the original manuscript will be changed to "*This inventory was compiled as part of the study of Emberson et al. (2021) using post-event PlanetScope optical satellite images acquired on 20 and 24 of March.*"

**Line 77: I thought you focus on rainfall-triggered landslides as also you indicated in the title of your manuscript. Why do you use the co-seismic landslide inventory of Roback et al. (2018)?**

**Ok, now I understand what you have done. You removed landslides triggered by the mainshock and work with the others. However, you do not know if these landslides were triggered by the aftershocks or rainfall events. Also, it is not clear when they were triggered. The confusing thing is you mentioned that you focus on "inventories of landslides whose timings are known a-priori to test and develop landslide". However, this does not one of those inventories.**

This is not correct. As stated at line 87 of the original manuscript "*we also removed all landslides whose trigger was specified by Roback et al. (2018) to be something other than the mainshock*" i.e. we remove landslides triggered by aftershocks or by rainfall and keep only the landslides triggered by the mainshock. Since the sentence was not clear before, we will change to "*we also removed all landslides specified by*

*Roback et al. (2018) to have been triggered by an aftershock or by rainfall, and use only those triggered by the mainshock in our analysis.*" For landslides triggered by the mainshock, their timing is known, which should resolve the confusion here.

**Why do not you simply pick another rainfall-induced landslide event inventory?**

The reason we do not use a rainfall-triggered landslide inventory here is that Nepal is a country for which the methods we develop here will be especially useful due to its extensive annual monsoon-triggered landsliding and long periods of cloud cover. It is also particularly challenging for SAR applications due to its steep topography, which results in distortion of the SAR images. It is thus particularly important to test the methods in this environment. Unfortunately, rainfall-triggered landslide inventories of known timing are not available here, so we use an inventory of earthquake-triggered landslides instead.

To improve the clarity, we will make the following change (Line 79 of original manuscript)

previous version : « *It is therefore useful to test landslide timing methods in this area, and, since well-timed landslide information is not widely available, we used earthquake triggered landslides...*

new version : *The steep topography of Nepal also makes it particularly challenging for SAR applications as it leads to distortion of the SAR imagery. It is thus important to test landslide timing methods in this environment, but inventories of rainfall-triggered landslides of known timing are not available. Therefore we instead used earthquake-triggered landslides…*
**65 timing methods.**

Apologies, but I cannot see any error on line 65

**Line 81: "we used earthquake-triggered landslides, which can be assumed to occur concurrently with the ground shaking" you already said it above, please remove this.**
We have removed the second part of this sentence, which as you point out, had already been stated at line 50 of the original manuscript.

**Lines 81-83: "since the inventory of Roback et al. (2018) covers a large area, with different areas having different Sentinel-1 coverage, we focused on triggered landslides within three large valleys" This does not explain why you focus on these three rectangular areas (actually one of them has a weird shape). You can cover a larger area mapped by Roback et al. (2018) and one Sentinel 1 image should be covering at least an area covering two of those rectangles. So how did you identify these rectangles really?**
The valleys were selected since Marc et al. 2019a have mapped landslides in these valleys during several monsoons following the earthquake and therefore will be of interest for future studies.

In fact, the Sentinel-1 time series is different for each Valley, making it complicated to combine these (See Fig.1). Although tracks 19 and 85 cover both Trishuli (Tr) and Buri Gandaki (BG), track 85 is much more complete over Trishuli, which is further South than Buri Gandaki, which is further north. The two valleys lie in the same track but within different scenes and several scenes are missing on track 85 for the Buri Gandaki Valley. This is probably because the satellite had not long been operational at the time of the earthquake. It is therefore not possible to combine them into a single time series. I attach a map below to illustrate this. Scenes on the descending track (19) are shown in green. In this case, BG and Tr are in the same scene (Bhote Kosi (BK) is outside and belongs to track 121 instead). On track 85 (shown in orange), BG and BK are in different scenes, while Tr lies in an area where the two overlap. Since data for Tr can be supplied from either the northern or southern scene, it has a more complete time series than either BG or BK. It is for this reason that data from track 85 is only used in Trishuli, while BK and BG, we are limited to the descending tracks (121 and 19 respectively).

[Figure]

Finally, the squares on Figure 1c do not show the exact extents of the inventories but show the locations instead, as in figures 1a and 1b, where the study areas are also not square. In order to be clearer, we will replace the squares, which are not particularly informative, with plots of landslide density for each event. This should make the extents of each inventory clearer.

**Lines 82-83: "large area & large valleys" please be more specific; either tell it directly or not mention it at all.**

The area covered by the inventory of Roback et al is 28,000 km², this will be given in brackets in the revised manuscript.

*« large valleys »* will be changed to *« valleys »* in the updated version of the manuscript

**Line 83: "valleys see large numbers of rainfall-triggered landslides" it does not sound correct, please fix the language.**

*"see"* will be changed to *"experience"* in the next version of the manuscript

**Line 84: "the timing of which would be one of the key applications of our method" Please remove this line. You already indicated your motivation.**

This will be removed in the revised manuscript.

**Line 89: You are using the inventory mapped by Roback and others but for some reason, you are citing Marc and others. Too much self-citation, remove Marc et al. (2019)**

Roback et al. do not draw any conclusions about landslides triggered by the 12 May aftershock, since most of the images they used to map the landslides were acquired between 2-8 May (i.e. between the mainshock and the aftershock). Therefore their inventory does not provide information landslides triggered by aftershocks and we cannot cite their paper here. However, landslides triggered by the aftershock have been observed in Marc et al. (2019a) and Martha et al. (2017).

Line 90: "close enough" not clear what this means. What would be close enough? What were the PGA or PGV values at those valleys? Did Martha and others map no landslide at those valleys and say this based on their observations? Or is this just an interpretation?
To improve clarity, we will change "*Of the three valleys we consider here, only Bhote Kosi was close enough to the epicentre to be affected by that event*" to "*Of the three valleys we consider here, landslides associated with this aftershock have only been observed in the Bhote Kosi (Martha et al. 2017), which was the closest to the epicentre.*"

**Figure 1: If you have such a plot (i.e., panel d) then please indicate ascending and descending images in panel d. This will be specifically important to see the dataset you used in the Gorkha case**

**where some of the ascending images should be missing. Please properly indicate what those abbreviations stand for (e.g., H in panel a and so on).**

Figure 1 will be redesigned to be made clearer in the next version of the manuscript. We will use the change suggested elsewhere in this review (the comment on line 141) to change e.g. H083A to H_asc. With this labelling, it should be clear which tracks are ascending and which are descending

**It is not clear how did you define pre-, co- and post- event image acquisitions. Do you explain this later in the method section? But you already refer to the term "co-event pair" in line 91, so the reader needs to know what it means. For instance, you indicate that in the Hiroshima case the heavy rainfall event occurred between 28 June and 9 July. Therefore, landslides were triggered (or mapped, as I mentioned this is confusing anyway) during this 10-day period. And you also mention that landslides were most likely triggered between 6 and 7th of July. Then why do you have such a large time period for "co-event pair"**

I apologise, we have generated confusion by using co-event (no quotation marks) to refer to the true pair of images spanning the landslide timing (as in line 91) while using "co-event" (with quotation marks) to refer to a "co-event" period that we have defined for testing the methods (i.e. a series of images that contains the correct co-event pair). Evidently, these are too similar and we will change to an alternative e.g. "trigger pair" to refer to the pair of images that span the trigger event and "co-event series" to refer to the series of images we have designated as co-event for testing purposes.

**Also, why do you represent the real event date as a single day? You mentioned above some time slots that landslides were most likely triggered. Why do not you indicate them also in panel d?**

Yes, that is correct. We have selected day 0 to correspond with the peak in rainfall, however the rainfall events were longer than 1 day, therefore the line should be thicker in places (as in e.g. Fig. 4). This will be changed in the next version of the manuscript.

**Line 92: "these two earthquakes can be considered as a single triggering event in Bhote Kosi" You can not consider them as a single event. However, if you cannot differentiate landslides possibly triggered by different factors for a given period of time, this needs to be indicated as a source of uncertainty in your analyses. I do not know what could be the consequences, but obviously, this needs to be discussed later on in the manuscript.**

While it is true that in terms of the spatial distribution of landsliding, we cannot consider them as a single trigger, that is not the aim of this study. Here we are trying to identify the timing of a landslide whose location we already know. Both the mainshock on 25 April and the aftershock on 12 May lie between our SAR acquisitions on 24 April and 18 May, it does not matter if the landslide happened at the time of mainshock or the aftershock, an assigned timing of 24 April – 18 May is right in either case.

I think this confusion has again arisen from confusion between our defined "co-event" 6 month time series and the correct co-event image pair. (i.e. the same as at line 91). As previously mentioned, we will be more careful to distinguish between these two in the next version of the manuscript.

We will replace "*these two earthquakes can be considered as a single triggering event in Bhote Kosi*" with "*these two trigger events are blended by our methods into a single time window*" to make this clearer.

**Lines 131-133: Could please explain how you defined these time windows (i.e, 6, 3 and 2 months)? What is the logic behind it?**
6 months is the approximate length of the monsoon season in Nepal (e.g. May – October), and we expect that our methods will be useful applied to monsoon-triggered landslides. Altering the length of this window is later explored in Fig 7. / Sect. 4.3 of the original manuscript.

The periods of 2 and 3 months were selected to strike a balance between having enough images to calculate reliable statistics and having too much time elapse over the course of the time series (which could lead to incorporating landslides from previous monsoon seasons etc.). Using more images also increases the computation cost.

**Line 131: "approximately six months" Later on you are saying two cases you took it as 6 months and in another one like 5 months. No need to repeat the same things. Please remove "approximately six months"**

We will rewrite lines 131-138 of the original manuscript to reduce repetition and explain the choice of 6 months, 2 months etc.

**Line 139: "In this figure" Which figure? Figure 1d? Then say it, please.**

*"this figure"* at line 139 of the previous manuscript will be changed to *"Fig 1d"* in the revised manuscript

**Line 141: "ascending track 72 over Zimbabwe will be referred to as Z072A" this is not a good idea. Why don't you refer to it, for instance, as Zimbabwe-asc or Z-acs. Or something like we can easily understand what you are referring to.**

Yes, there is no need to include the track number here since there is never more than one ascending or descending track used in any event. Therefore we will write the acronym as e.g. Z-asc. This will be changed throughout the text and figures of the revised manuscript

**Line 143: Please make a kind of introduction and tell us that you will introduce three different methods for some reason. And please indicate that reason too. It is difficult to follow the text. You are explaining your method (which is ok, I do not have any complaints) but if we do not understand why you are providing this information, we cannot follow you.**

We will add an introduction to Section 2.4 to make the section easier to follow

**Line 150: "pixels that are dissimilar to those within the landslide, for example pixels located on the opposite side of a ridge, in a river or with different surface cover" Could you be more specific? How do you define similar and dissimilar pixels? Based on what? Based on land cover? Or do you have some other criteria you take into account?**

**I see, in the next lines you are explaining those variables. But please first tell us what we are talking about (i.e., what you mean by dissimilar pixels) and then you can mention that you removed them.**

This is described in the following lines. We will restructure this paragraph to make it clearer

**Line 151: "three surfaces" three variables might be better**

*"three surfaces"* in the original manuscript will be changed to *"three variables"* in the revised manuscript

**Line 154: "amplitude variability" is this the third one? You mentioned the first and second variables but which one is the third?**
Yes amplitude variability is the third. We will change

*Second, we used a stack of N pre-event SAR images i (Fig 1) to calculate the mean amplitude $A_{mean,j}$ and amplitude variability $\Delta A_{mean,j}$ for every pixel j through time.*

In the original manuscript to

**"***Second, we used the mean amplitude $A_{mean,j}$ (Eq. 2) and third the amplitude variability $\Delta A_{mean,j}$ (Eq. 3) of each pixel j through a stack of N pre-event SAR images.***"**

in the revised manuscript

**Figure 2: Please fix the label of the panel (c) and please also indicate the label (c). Remove label (b) from panel (a).**

Sorry for the oversight, we will fix the labels on this figure. In the revised version of the manuscript

**What do you mean by "vegetation removed"? Do you mean because of landsliding? If it is the case,**

**no need to indicate this.**
We will remove this label from the figure.

**Is the blue bar not centered for some reason? Did you do this on purpose? Or is this something you need to fix? And please indicate the corresponding panels while referring to "blue bars".**
The blue bars represent the duration of the rainfall event, but the time series is centred on the peak in the rainfall, which is not necessarily halfway through the duration of the rainfall event.

We will add indications to the corresponding panels when describing the blue bars in the figure caption
**Line 160: "this"?**
We will change "*this landslide*" to *"each landslide"*
**Line 164: "When combining methods, we found" This is still your method section and you haven't said anything about other methods yet. This is to say that I do not understand what you are referring to?**
The combining methods is done in section 3.1, so this sentence refers to that. However, we agree it does not belong here and will move to Section 3.1
**Lines 160-166: Based on what you explained here how we should interpret Figure 2c?**

**"A step change in the difference between the median landslide amplitude and the median background amplitude is then used as an indicator of landslide timing." Based on your interpretation, could you point out the timing of the landslide in Figure 2c. Which one is a step increase or a step decrease? Other than the signal received from the shadow area, I do not see any significant change in overall fluctuations of amplitude values associated with rainfall events (indicated by the blue bar in fig2c).**

The step change is observed in the difference between the landslide and background time series. It is true that it is hard to see when considering the median landslide amplitude on its own. In Figure 2c, you can see that the distance between the orange and teal lines increases after the rainfall event (when the landslide happened). However, it is true that is not the clearest way to display this. We will redesign this figure incorporating the plots below, which show the time series for each method.

[Figure]

*Figure 1 Example time series for each method described in Section 2.4 for a single landslide from the Hiroshima data set using SAR data from Sentinel-1 track 019D. The blue bar shows the duration of the peak rainfall associated with this event (6-7 July 2018).*

The time series above are for a different landslide to the example shown in the original manuscript The landslide selected in the previous manuscript version demonstrated Methods 1 and 3 well, but for Method 2, the time step change was not clearly visible. This was raised in the next comment from the reviewer:

**Lines 168-171: The same comment as above, please explain how you interpret Figure 2d. I do not see a specific change in the trend associated with the blue bar other than some fluctuations.**

Therefore, we have decided to change to a different landslide polygon for which this trend is clearer. (We also include here the time series for Method 4, which was not included in the previous version of the manuscript)

We also add the convolution functions (in grey on the new figure), which demonstrate how the peak / trough in the function corresponds to a step change in the landslide timing detection method. This peak / trough is how we select the step change.

**Lines 185-188: I can not see any connections between these two lines. Could you be more clear about what you mentioned about uncertainty in landslide mapping in the first line?**

We will add a connecting sentence *"Small spatial mismatches between landslide polygon locations and SAR pixel locations could lead to pixels on the edge of landslides being excluded."*

**Line 196: "Step change identification" As usual, please help the reader to follow you. You have just mentioned three methods to identify the timing of landslides. And I guess you are going to combine these three methods to get the best result out of all, right? This is also not clear and needs to be indicated. And here you keep going with another step of your methodology. I think it would be great if you make a flow chart explaining your methodology. You can briefly describe each and every step of your method at the beginning and then we would have an idea about what is going to be in the next step. I know what I am suggesting is a super smart thing, is quite a traditional way of presenting your method but it is also a good way of doing this.**

Section 2.5 will be rewritten in response to this comment, the comment at line 198 and comments made by reviewer 3 in order to make the methods section easier to understand.

**Line 197: "Sects. 2.4.2, 2.4.1 and 2.4.3" just say above**
This will be changed to "above" in the next version of the manuscript.

**Line 198: "The step function was made up of a series of -1s and 1s of twice the length of the co-event time series" Do twice the length of the co-event time-series means 12 months? And why?**

This does not mean 12 months, it merely means twice the number of points in the SAR times series. (e.g. for a series of 12 SAR images, we convolve with a series made up of 12 -1s followed by 12 1s i.e. a series of length 24). The output of this convolution is a series having the same length as our SAR time series with a peak at the location that best agrees with the step function. In this way, we can automatically detect the location of a step change in the SAR measurements.

We are rewriting this section in response to this comment, the comment at Line 196 as well as comments made by reviewer 3 in order to make this section easier to understand.

**Can't you make a figure to explain what you have done at this step?**

We hope that by adding the convolution functions to the panels we will add to Figure 2 (see response to comments at line 160-166 , it will be clearer how this process works and how the peak / trough in the convolution corresponds to the step increase / decrease in the landslide timing method.

**Lines 216-217: "the correct date by chance for a method with no skill" is not clear!**

We will change "*the correct date by chance for a method with no skill*" in the original manuscript to "*the correct date randomly by a method with no skill (i.e. assigning a random date pair)*" in the revised mansucript

**Table 2: What do those percentages stand for? For instance, in Hiroshima (H083A), you have 540 landslides and based on Pixel Variability you correctly identified the occurrence dates of 181 landslides, right? This means you correctly identified 33% of them. Then where did 59% come from? Obviously, the percentages indicate something else but I did not get what it is. I am sorry maybe it is my fault that I could not get it but this is not clear for sure.**

This has been misunderstood. In fact, we have 540 landslides in Hiroshima, 181 of these landslides are assigned a timing by the Pixel variability method and 59% of these 181 are correct (i.e. 107 correct).

In order to make this less confusing, we will remove the percentages and instead provide correct / assigned in each column of the table. The table caption will also be updated to reflect this and to make this less confusing.

**Actually based on these numbers and what you present in Figure 5, you can make an estimation for only a small fraction of the examined landslide population, right?**

Yes this is correct.

**You mentioned about confusion matrix, then why don't you present your results based on that structure?**

The confusion matrix in Table 1 of the original manuscript was designed to assess how the size of the peak in the convolution function can be used to predict whether or not an assigned date is correct and to allow the calculation of the F1-score.

After the thresholds for each method have been selected and we move to combining predictions from multiple methods and tracks, there is no way to divide the landslides into TP, FP, FN, TN because we have three categories: landslide assigned the correct timing, landslide assigned the incorrect timing and landslide not assigned any timing.

**Lines 243-245: "Out of all the non-masked landslides in each inventory, 23% were assigned a date in Hiroshima, 21% in Zimbabwe and 14% in Trishuli and of these, 80% of the estimated dates in Hiroshima were correct, 73% in Zimbabwe and 81% in Trishuli (Table 2)." So as you also indicated in the abstract, these are the percentages of correctly predicted landslides:**

**Hiroshima ~18%**

**Zimbabwe~15%**

**Trishuli~11%**

**Then how about the rest? Then I do not think what you mentioned in your abstract is convincing:**

**"This will allow multi-temporal landslide inventories to be generated for long rainfall events such as the Indian summer monsoon, which triggers large numbers of landslides every year and has until now been limited to annual-scale analysis."**

**Landslides could occur on different dates over a monsoon season in a given area of interest. And we would miss a great majority of them if we use this technique. Therefore, I do not think we can confidently argue that multi-temporal inventories can be generated based on this method (This method does not mean to generate multi-temporal inventories anyway). I am not saying this method is useless but is also clear that this could be just a small step towards what you are arguing in your abstract.**

Yes we have changed the abstract and conclusions to better reflect this and have also slightly improved these success rates by altering the methods. See response to this comment at the beginning of this document.

**Line 250: "Factors affecting performance of each method" This does not sound like your results. You should move this section to the Discussion section.**

Yes this can be moved to the discussion section in the revised manuscript.

**Line 383: " Application to future events" Please merge this section with the conclusion section, no need to have this heading, the paper is already too long.**

We prefer to keep this section separate from the conclusions, since the removal of one heading will not really change the length of the paper.

**Line 384: As you said you are just estimating the time window that landslide might have occurred. So you are not estimating the exact occurrence date of landslides. You should clarify this also in your title.**

Yes. In the next version of the manuscript, all references to landslides being "dated" or "assigned a date" (which implies a precision of one day) will be changed to assigned a timing /date pair/ time window or similar

In line 384 of the original manuscript (and elsewhere), we will change "*dated*" to "*timed*"

We will give the revised manuscript a new title to better reflect the results in the manuscript e.g. "*Using Sentinel-1 radar amplitude time series to constrain the timing of individual landslides: a step towards understanding the controls on monsoon-triggered landsliding*"

**Line 406: "generate multi-temporal" You are not generating landslide inventories. You are just trying to label existing landslide inventory in terms of their time of occurrences.**

In the conclusions section, we will change "*These methods will allow us to generate multi-temporal landslide inventories for long rainfall events, unlocking comparisons between rainfall data, hydrological models and triggered landsliding.*" to "*These methods will provide information on the timings of some individual landslides and allow spatio-temporal clusters of landslides to be associated with peaks in rainfall during long rainfall events, unlocking comparisons between rainfall data, hydrological models and triggered landsliding.*"

**Lines 414-415: "Google Earth Engine and Python codes used in generating the time series and detecting landslide timings will be provided if the manuscript is accepted for publication" The authors should share the code so we can check how it works really.**

The code will be provided as a supplement to the next version of the manuscript as requested. I apologise for omitting it, I was not aware it was necessary at this stage.

---

## Author Comment (AC6)

We thank the reviewer for taking the time to review our paper. The following is a line by line response to all comments made on our manuscript (reviewer comments given in bold type).

**This manuscript presents a method for estimating the time-window of landslide occurrence based on Sentinel-1 GRD data. The method is tested against inventories of landslides with known timestamp. The underlying research question is definitely an interesting one and may also be of importance for e.g. finding timestamps to known polygons or for doublechecking timestamps reported in landslide inventories. I do commend the authors as clearly a lot of work was put into the analysis.**
**Having worked with Sentinel-1 data for similar endeavors, the reported goodness-of-fit of results is well in line with my previous experience and expectations. However, the fact that only 1/5 of all landslides can be detected at all with reasonable accuracy indicates one of the following two conclusions for me:**
- **the level of maturity of the analyses is still rather low (this is not limited to the study at hand, but rather a general statement);**
- **SAR data is only suitable to a limited extent for the task at hand, or rather for the detection of sudden gravitational natural hazard events in general.**
**Overall, I think that this is an interesting contribution that is worth publishing subject to major revision. Generally, I suggest to report the results more as a potential contribution towards using Sentinel-1 data for estimating time-windows of event occurrence. Some parts of the manuscript read as if a well-working method is presented that works generically for identifying timings of landslides. However, this is still very much work in progress. For instance, I don't think that** *"This will allow multi-temporal landslide inventories to be generated for long rainfall events such as the Indian summer monsoon"* **in a comprehensive manner.**

We will revise the manuscript to make it clearer throughout that our methods cannot establish timings for all the landslides in an inventory (and in fact, will only provide timings for ~20%).
The quote here is taken from the abstract. In response to this and to similar comments made by reviewer 2, we will change the end of the abstract.
Previous text: *"our methods allow 20% of landslides to be timed with an accuracy of 80%. This will allow multi-temporal landslide inventories to be generated for long rainfall events such as the Indian summer monsoon, which triggers large numbers of landslides every year and has until now been limited to annual-scale analysis."*
New text: *" our methods allow 20-30% of landslides to be timed with an accuracy of 80%. Application of our methods could provide an insight on landslide timings throughout events such as the Indian summer monsoon, which triggers large numbers of landslides every year and has until now been limited to annual-scale analysis"* .
Similarly in the conclusions section, we will change "*These methods will allow us to generate multi-temporal landslide inventories for long rainfall events, unlocking comparisons between rainfall data, hydrological models and triggered landsliding.*" to "*These methods will provide information on the timings of some individual landslides and allow spatio-temporal clusters of landslides to be associated with peaks in rainfall during long rainfall events, unlocking comparisons between rainfall data, hydrological models and triggered landsliding.*"

This removes the words "multi-temporal landslide inventories", which were misleading since we cannot provide a complete inventory where all the landslides have timings assigned.
We will make also make sure this is clear in the discussions / conclusions and throughout the manuscript.
We will also make sure it is clear throughout that we are not assigning specific dates, but instead time windows of (in most cases) 12 days to each landslide.

We have also made some adjustments to the methods, which somewhat improve the number of landslides for which we can assign a timing from 20% to 30% in favourable cases. This does not change the overall message of the manuscript, since we keep the same 80% accuracy and still do not assign a timing for the majority of the landslides in an inventory, but it represents an improvement to the method. All following replies and alterations to the text therefore incorporate these new sensitivity levels.

(i)       A new method based on bright spots observed at the edges of landslide polygons

As described in Section 2.4.3 of our original manuscript, shadows are cast by trees at the edges of landslide scars due to the imaging geometry of the SAR sensor. (See figure below, altered from Figure 2b of the original manuscript). On the opposite side of the scar, we may observe a bright patch in the SAR amplitude images due to double bounce scattering between the exposed soil and trees on the far side of the landslide scar, and the focussing of the microwave energy into a small area (Villard and Borderies, 2007). Similarly to the shadow method, we compare the pre-event and post-event SAR time series and identify pixels which have experienced a strong increase in amplitude (we found a threshold of >5dB to perform best in this case). Incorporating this method means that, increases the final number of landslide that are assigned a timing by 3% in Hiroshima, 2% in Zimbabwe and 7% in Trishuli.

A description of this method will be added to Section 2.4 (SAR amplitude techniques for landslide timing) as " 2.4.4 Method 4: Geometric bright spots"

[Figure]

(ii)       Increase in size of the landslide polygons

As described at lines 185-195 of our original manuscript, there may be a spatial mismatch between the optically-derived landslide polygons and the SAR imagery. Previously, we increased the size of landslide polygons by 10m (1 SAR pixel) for the geometric shadows method to try to account for this. However, this effect may also affect the results from other methods, and decrease the number of landslides for which we are able to assign a time window.

Therefore, for landslides that are not assigned a date by at least 2/4 of our methods, we now increase the size of the landslide polygon using a 20 m  buffer for all methods and repeat, with the aim of trying to assign landslide timings to some of the landslides that experience this spatial mismatch. This improves the number of landslides we are able to detect in each case study event by 4% in Hiroshima and 5% in Zimbabwe and Trishuli, and will be added to the next version of the manuscript.

Since this will now be described in its own subsection in the methods section, text referring to this at lines 185-195 and 317-326 of our original manuscript will be removed.

**There will definitely be a biases in terms of identified slides,**
The biases towards which slides can be assigned a timing is related to section 3.3 "Factors effecting the performance of each method." For example, it is clear that larger landslides are more likely to be assigned a timing than smaller landslides using our methods (Fig. 5 d-f).
In the manuscript, we considered how this effected where our methods could be applied (for example, they will not work well for inventories of small landslides or in arid environments), but we did not consider how well the timings of the 30% of landslides we assign a timing to using SAR methods will represent the full inventory. In fact, this 30% will be biased to contain a somewhat higher proportion of large landslides and landslides in more heavily vegetated areas than the original inventory. We will add this point to the revised version manuscript

New text at line 252 of original manuscript: *"For future applications, this helps to determine the environments where our methods can be expected to work well. It also provides an insight on potential biases in terms of the subset of a landslide inventory that is assigned timings by our methods."*

**a vast majority of sildes will be missed or - worse - labelled incorrectly,**
Our methods should not incorrectly label a large percentage of slides. We expect that if we apply our methods to an inventory of rainfall-triggered landslides, we will obtain an inventory in which ~70% have no timing information, ~24% are correctly timed and ~6% are incorrectly timed. We will be careful to make this clearer in the abstract, results and conclusions sections of the revised manuscript.

**and things might look dire when thinking beyond the scope of this study, e.g. if no polygons are availabe.**
If polygons are not available for an event, it will not be possible to apply our methods in their current form – they are not designed to be applied to events for which we do not have a pre-existing landslide inventory. We will ensure this is clear in the revised version of the manuscript.

**Rather, I suggest to present the status quo and clearly highlight the limitations and highlight needs for further research based on the findings of this study. Also, the title should be clarified to indicate that time-intervals are identified rather than exact time stamps in terms of exact dates**
The new manuscript will be given a new title that better reflects the results in the paper. For example *"Using Sentinel-1 radar amplitude time series to constrain the timing of individual landslides: a step towards understanding the controls on monsoon-triggered landsliding"*

**On a sidenote, I was slightly confused when I saw that the special issue title concerns the "Himalayan region", and study areas in this manuscript include Hiroshima and Zimbabwe. Since the Tr, BG and BK case studies are located in Nepal I think that's fine. The authors might consider adding some indication of the case study areas in the title, as I think it is actually very nice consider inventories from three different locations.**

The manuscript was submitted to the special issue since, while only 1 of our 3 landslide inventories was located in the Himalayas, the methods we develop are expected to be particularly useful in Nepal due to high levels of landsliding and cloud cover during the monsoon season in that country.

We will emphasise this in the revised introduction
previous text (lines 31-34 of original manuscript): *"the Indian Summer Monsoon (June – September) triggers hundreds of landslides every year in the Nepal Himalaya and cloud-free optical satellite imagery is unlikely to be available throughout this period (Robinson et al. 2019). This limits analysis of these landslides to the annual scale (e.g. Marc et al. 2019a; Jones et al. 2021).*

new text: *"the Indian Summer Monsoon (June – September) triggers hundreds of landslides every year in the Nepal Himalaya and cloud-free optical satellite imagery is unlikely to be available throughout this period (Robinson et al. 2019). This limits analysis of these landslides to the seasonal scale and prevents association of individual landslides or landslide clusters with specific peaks in rainfall (e.g. Marc et al. 2019a; Jones et al. 2021)."*

This is also better reflected in the proposed new manuscript title.

**Specific Comments**
**Abstract: suggest to remove "thousands of" as this is somewhat unspecific without a time unit and potentially misleading**
"thousands of" will be removed in the revised version of the abstract.
**"Landslide locations are typically mapped using optical satellite imagery". There are many more methods that are "typically" used for such purposes, including ALS and orthophotos. The authors even mention this in section 2.1 ("drone and aerial imagery"). This might not be the case for all regions around the world as this clearly depends on the country under consideration, but the regions of interest have not yet been specified up to this point. VHR optical satellite imagery is expensive, while the spatial resolution of free data (e.g. Sentinel-2) is often too coarse to detect small slides. Free VHR data might be available e.g. through**

**Google Earth, but not at the temporal resolution required to pinpoint the time windows to periods of some days.**

We do not really consider small landslides in this study as these are difficult to detect with Sentinel-1 SAR, which has a relatively coarse resolution (once multi-looked to reduce noise) and our methods are shown in Figure 5 of the original manuscript to perform better on larger landslides.

Also, optical satellite images are also subject to cloud cover so, while they allow smaller landslides to be mapped, they do not address the problem we are tackling in this manuscript.

We will change "*typically*" here to "*often*"

**Section 2.1: I think the structure of this section can be improved. For instance:**
**l.64f: "We used three published polygon inventories of landslides whose timings are known a-priori to test and develop landslide timing methods." Since the authors continue "We filtered each inventory to remove ..." I was wondering whether there was a reference for these data sets? This point is re-established two lines later with a reference to Emberson et al. (2021), leading to some interruption of the flow from**
**a reader's perspective. l.66: "10 × 10 m SAR pixels" are mentioned. So I assume at this point that S-1 GRD data was used. Yet, the data source is unclear at this point. Also, why 20? I suggest to keep methodological considerations (e.g. filtering slides < 2000m², minimum number of SAR pixels, etc) separate from the initial inventory description.**

We will restructure this section so that we first describe the inventories and then describe how they were filtered to remove small landslides. We will also change "*10 x 10 m SAR pixels*" to "*Sentinel-1 GRD pixels, of size 10 x 10 m*"

**line 74: Planetdove: Do you mean PlanetScope DOVEs?**

Yes, we should have used "PlanetScope" here, not Planetdove. Thank you for correcting this.

**Please double check figure references in the body text. Fig. 1 - specifically, only Fig.1(d) - is referenced the first time on line 135. Fig. 4 is the first figure to be referenced in the text.**

This was a mistake. At line 119 of the previous version of the manuscript, we refer to "*(e.g. Tozang landslide, Mondini et al, 2021, Fig 4)*". "*Fig 4*" there was not intended to refer to Figure 4 of our paper, but to Figure 4 in the paper of Mondini et al. 2021. This was not clear, and in the next version of the manuscript, we will reduce this reference to "*(Mondini et al. 2021)*" only.

**It took me a while till I figured out the meaning of the terminology you used for the orbit IDs (e.g. "H083A"). Please specify more clearly that this is a combination of study area, orbit number and orbit direction.**

Yes, this was also raised by reviewer 2, who suggested a change of terminology from e.g. "H083A" to "H_asc" to describe the tracks. We will make this change in the next version of the manuscript, which should also resolve this comment.

**"We tested both of these polarisations, but found VV to perform better than VH so present only the results for VV." This is an interesting finding. How was this evaluated?**

Yes it is interesting since other studies (e.g. Handwerger et al. 2022) use VH and find it to be the most successful option. We attach a version of table 2 in the original manuscript which contains the same results from VH in Zimbabwe and Hiroshima (VH data are not available for the Nepal case study, which occurred soon following the launch of Sentinel-1 before dual-pol data began to be consistently acquired – this would be a further disadvantage to using VH).

| | Hiroshima | | Zimbabwe | | Trishuli | | Buri Gandaki | Bhote Kosi | | | Hiroshima (VH) | | Zimbabwe (VH) | |
|---|---|---|---|---|---|---|---|---|---|---|---|---|---|---|
| Track | H090D | H083A | Z079D | Z072A | Tr019D | Tr085A | BG019D | BK121D | | | H090D | H083A | Z079D | Z072A |
| Total landslides | 543 | | 383 | | 650 | | 922 | 1554 | | | 543 | | 383 | |
| non-masked | 543 | 540 | 383 | 383 | 485 | 474 | 592 | 894 | | | 543 | 540 | 383 | 383 |
| ls-b inc | 137 (26%) | 92 (36%) | 90 (50%) | 66 (41%) | 106 (38%) | 107 (35%) | 152 (36%) | 313 (36%) | | | 94 (12%) | 97 (29%) | 40 (20%) | 26 (12%) |
| ls-b dec | 126 (38%) | 205 (57%) | 182 (23%) | 155 (34%) | 156 (37%) | 143 (22%) | 113 (27%) | 310 (32%) | | | 182 (25%) | 251 (63%) | 262 (28%) | 266 (36%) |
| pix var | 160 (45%) | 181 (59%) | 134 (50%) | 83 (47%) | 141 (42%) | 125 (44%) | 141 (30%) | 261 (43%) | | | 155 (32%) | 222 (56%) | 152 (16%) | 110 (27%) |
| shadow | 79 (49%) | 122 (75%) | 43 (55%) | 58 (72%) | 45 (80%) | 50 (80%) | 17 (88%) | 52 (87%) | | | 144 (43%) | 227 (74%) | 125 (39%) | 140 (47%) |
| combined >2m | 51 (75%) | 99 (88%) | 48 (65%) | 39 (85%) | 45 (76%) | 33 (88%) | 33 (64%) | 86 (63%) | | | 71 (54%) | 166 (80%) | 88 (33%) | 87 (60%) |
| combined >3m | 10 (80%) | 22 (100%) | 5 (80%) | 2 (50%) | 8 (100%) | 3 (100%) | 1 (100%) | 5 (100%) | | | 9 (89%) | 38 (97%) | 17 (35%) | 9 (89%) |
| asc & desc | 122 (80%) | | 81 (73%) | | 70 (81%) | | - | - | | | 196 (69%) | | 142 (44%) | |
| asc & desc 2m,1t | 79 (73%) | | 64 (72%) | | 52 (77%) | | - | - | | | 139 (65%) | | 102 (23%) | |
| asc & desc 3m | 43 (91%) | | 17 (76%) | | 18 (94%) | | - | - | | | 57 (81%) | | 40 (75%) | |
| baseline (1/n) | 7% | 17% | 10% | 7% | 8% | 8% | 8% | 14% | | | 7% | 17% | 10% | 7% |

As you can see, we have a generally lower accuracy for VH data, especially in Zimbabwe.

Since this may be interesting for future studies, we can include it as a supplement, which we will reference at line 110 of the original manuscript.

In general, it is expected that VH should be more sensitive to changes in volume scattering (here, scattering within the forest canopy). Therefore, since the removal of vegetation due to a landslide should reduce this component, we would expect VH to decrease for a landslide polygon. Therefore, of the methods analysed here, we might expect VH to perform better than VV for the landslide-background decrease method (ls-b dec in the above table), which is in fact the case for H083A, Z079D and Z072A (but not H090D). However, since VH performs worse than VV for the other methods, it is better to use VV.

This table was calculated prior to the addition of our new method based on bright patches at the edges of landslide polygons and so this method is not included in the table. However, the double-bounce scattering component is generally very weak in cross-polarised SAR images, therefore we do not expect that this method would have been successful using VH.

**The copernicus DEM would have been a more recent DEM version, also available globally at a resolution of 30 m.**

We chose to use the SRTM DEM here since it is already available as a dataset in Google Earth Engine, making it easier to integrate into the slope correction module for future users.

**l. 123: "... geographic coordinates at a resolution of 20 x 22 m and a pixel size of 10 x 10 m". I suggest to specify this further, this statement might be confusing to an audience from the broader field of natural hazards research not familiar with (SAR) satellite data**
Previous text: "*we used the Google Earth Engine Sentinel-1 ground range detected (GRD) data set. These data are preprocessed following the workflow of Filipponi et al. (2019) to obtain the sigma0 backscatter coefficient at a resolution of 20 x 22 m and a pixel size of 10 x 10 m.*"
New text
"*we used the Google Earth Engine Sentinel-1 ground range detected (GRD) data set. These data are preprocessed following the workflow described in Filipponi et al. (2019) to obtain the backscatter coefficient sigma0 at a resolution of 20 x 22 m in radar coordinates. The data are then resampled onto a 10 m grid in geographic coordinates.*"
**l. 161/Figure 2: "A step change in the difference between the median landslide amplitude and the median background amplitude is then used as an indicator of**

**landslide timing." It might be beneficial to plot this difference?**

[Figure]

*Figure 1 Example time series for each method described in Section 2.4 for a single landslide from the Hiroshima data set using SAR data from Sentinel-1 track 019D. The blue bar shows the duration of the peak rainfall associated with this event (6-7 July 2018).*

Yes, this is a good point. In response to this and to similar comments from reviewer 2, we have prepared 3 panels (above) which will be incorporated into Figure 2. This makes the step change clearer (Note we have also changed to a different landslide polygon due to changes requested by reviewer 2 – these time series are thus not comparable to those in Figure 2 of the original manuscript).

These panels also show as a grey line the convolution between the method and a step function to make it clearer how a step change results in a peak or trough in the convolution function.

**l. 164: "When combining methods, we found that using ..." Since the other two methods have not yet been described it might be better to move such statements towards the end of your methods section, when all three methods have been properly introduced?**
This line will be removed here and we will describe the combination of methods later in Section 3.1

**2.5 Step change identification: Up until here I was able to follow the text with rereading some parts several times again. Here I really had to pause and ponder upon backtracking multiple times to be able to understand what is described here. Please be more concise here on how all the aforementioned methods are combined exactly, how the step function is set up and why. Some sort of graphical depiction of the workflow would probably help a lot to foster overall understanding of the whole processing pipeline.**

This Section will be rewritten in response to this and to comments made by reviewer 2 in order to make it easier to follow. Additionally, the revision of Fig. 2, which now displays the time series and convolution with the step function, should help to clarify the text.

**l. 216: reporting a baseline as reference is a good idea for putting the achieved results in context.**
Thank you.

**Specificity is reported in table 2, F1-score is reported in Fig. 3. Providing confusion matrices of all results in the appendix might be interesting as a more detailed reference of results.**
This was our mistake – Table-2 does not show specificity and we will alter the table caption to better describe the results.

The confusion matrix in Table 1 of the original manuscript was designed to assess how the size of the peak in the convolution function can be used to predict whether or not an assigned date is correct and to allow the calculation of the F1-score.

After the thresholds for each method have been selected and we move to combining predictions from multiple methods and tracks, there is no way to divide the landslides into TP, FP, FN, TN because we have three categories: landslides that are assigned the correct timing, landslides that are assigned an incorrect timing and landslides that are not assigned any timing.

In response to this, and to comments made by reviewer 2, we will change to showing *correct / assigned* rather than *assigned (% of assigned that are correct)* for each method in Table 2. By providing the raw numbers, any statistic can be calculated by the reader if they require it.

**Overall, appropriate performance metrics and their interpretation is of key importance. In fact, when thinking about the implications of the method presented here, this is crucial. If no validation data are available (e.g. when this method is applied to a new data set), a vast majority of identified dates (more precise: time windows) will be incorrect. This needs to be discussed.**
The problem is not that a vast majority of landslides will be assigned an incorrect time window, but instead that the majority of landslides will not be assigned any time window at all. The subpart of the inventory that has timings assigned will be biased towards larger landslides and those in more vegetated areas. We will discuss this in the revised version of the manuscript. (See response to earlier comments.)

Of those landslides that are assigned a time interval, we expect that 80% of the time, this interval should be correct. Therefore if, in a new dataset, we observe a spatio-temporal cluster of landslides, we can assume the timing of this cluster is correct, since it is very unlikely that all the timed landslides in this cluster would be assigned the same incorrect date.

**Publishing the code (e.g. on GitLab/GitHub) would be welcome for the final manuscript, but also of interest from a reviewer's perspective. If there are concerns with respect to sharing code before the publication is accepted, there are surely opportunities for embargos.**

The GEE and python codes will be shared as a supplement to the next version of the manuscript.

**Technical comments**

**Please double check Equation (1), the dot and "area" are somewhat floating around there.**
Yes, we made a mistake with the equation here in LaTeX here, the dot is in the wrong place. This will be fixed in the next version of the manuscript

**Figure 2(a): green text on green background is hardly readable.**
We will adjust the colours on this figure to make it easier to read

**Figure 2(c): y-axis label is unreadable.**
We are replacing this panel and (d) with the panels shown in our response to your comment above on line 161. We will ensure the y-axis is legible in these new panels

**Table formatting in Table 2 is off (e.g. first line - alignment of "Total landslides"). The "Asc & Desc" columns are also aligned in a confusing way. Numbers should be rightjustified for better readability.**

Thank you for these suggestions, we will make these adjustments to the table to improve it's readability.

**Table 2: I suggest to split the information in the columns, and avoid combining multiple units (number and percentage, i.e. specificity) in one cell.**

In response to this and to comments made on this table by reviewer 2, we will change this so that each column contains *correct / assigned* rather than *assigned (% of assigned that are correct)*.

**Overall, I suggest to use a more consistent plotting style (including readable colorscales) throughout the manuscript.**

We will ensure this is the case in the revised manuscript.

---

## Author Response (AR1)

**Author's response to comments**

We thank the reviewers for taking the time to review our manuscript. The following is a complete set of responses to their comments, which have helped us to improve the quality and clarity of the manuscript.

**Reviewer 1**

**Comment 1** adding one/two sentences regarding the choice of using Google Earth Engine and its advantages compared to other options:

**Manuscript Change** New text: "*Google Earth Engine is a freely accessible, cloud-based platform that allows users to access Sentinel-1 data without the technical expertise and computational facilities otherwise required to process SAR data. It also provides access to other datasets used in this study such as Sentinel-2 and the shuttle radar topography mission (SRTM) digital elevation model (DEM)."* (Sect. 2.3, lines 135-137 of the revised manuscript)

**Comment 2** (maybe to add as a supplement material) a figure showing some landslides that have been successfully dated, how they appear in amplitude image before and after the failure and highlight the features of the three methods on them (the shadow area, the buffer around and or respective difference with the landslide body, etc.).

**Manuscript Change** We will attach as a supplement to the revised manuscript three examples of correctly timed landslides, which were selected due to their having a different SAR signal in order to demonstrate:

1) A landslide from the Hiroshima dataset successfully assigned a timing interval based on (i) a decrease in landslide vs. background amplitude, (ii) increased amplitude variability, and (iii) the emergence of shadow pixels. The emergence of bright pixels, which have been added as a 4th method in response to comments made by reviewers 2 and 3, are not applicable in this case since no bright pixels were identified within the landslide polygon.

2) A landslide from the Zimbabwe dataset successfully assigned a timing interval based on (i) an increase in landslide vs. background amplitude and (ii) increased amplitude variability. Since no shadow or bright pixels exist within the landslide polygon, these two techniques could not be applied in this case

3) A landslide from the Zimbabwe dataset successfully assigned a timing interval based on (i) increased amplitude variability, (ii) the emergence of shadow pixels and (iii) bright pixels. The difference in background versus landslide amplitude was not effective in this case.

A reference to these three examples is now given at line 161 of the revised manuscript.

**Reviewer 2**

Comment 1: However, the authors are able to come up with a time estimation only for 20% of landslides with an accuracy of 80%. Therefore, I doubt if this is successful research in the end. Frankly speaking, I am not sure and just hesitating to say that the results are promising. However, what I can say is the output of this research is not fulfilling what the authors are promising in the abstract/conclusions.

Response: First, we have made revisions throughout the manuscript to make it clearer that our methods assign timings to 20-30% of landslides in an inventory and that we do not promise a full "multi-temporal inventory" with timings for all landslides. This was not previously clear and we thank the reviewer (along with Reviewer 3) for drawing our attention to this.

However, while this is a drawback, we do not believe that it renders the work unsuccessful. First, landslide inventories often include thousands of landslides, meaning that by timing 20% of these, we obtain landslide timing information on a statistically useful number of landslides. Similarly to the large number of studies deriving rainfall threshold our methods would allow us to constrain the rainfall characteristic which have preceded the triggering of landslides (even with a time window of 6-12 days, at the scale of the monsoon it would be a substantial improvement), or to determine whether a subsample of monsoon-induced landslides display spatio-temporal clusters which could be interpreted in terms of triggers.

We have also made some adjustments to the methods, which somewhat improve the number of landslides for which we can assign a timing from ~20% to ~30%. This does not change the overall message of the manuscript, since we keep the same 80% accuracy and still do not assign a timing for the majority of the landslides in an inventory, but it represents an improvement to the method.

    (i)       A new method based on bright spots observed at the edges of landslide polygons

As described in Section 2.4.3 of our original manuscript, shadows are cast by trees at the edges of landslide scars due to the imaging geometry of the SAR sensor. (See figure below, altered from Figure 2b of the original manuscript). On the opposite side of the scar, we may observe a bright patch in the SAR amplitude images due to double bounce scattering between the exposed soil and trees on the far side of the landslide scar, and the focussing of the microwave energy into a small area (Villard and Borderies, 2007). Similarly to the shadow method, we compare the pre-event and post-event SAR time series and identify pixels which have experienced a strong increase in amplitude (we found a threshold of >5dB to perform best in this case). Incorporating this method means that, increases the final number of landslide that are assigned a timing by 3% in Hiroshima, 2% in Zimbabwe and 7% in Trishuli.

[Figure]

    (ii)      Increase in size of the landslide polygons

As described at lines 185-195 of our original manuscript, there may be a spatial mismatch between the optically-derived landslide polygons and the SAR imagery. Previously, we increased the size of landslide polygons by 10m (1 SAR pixel) for the geometric shadows method to try to account for this.

However, this effect may also affect the results from other methods, and decrease the number of landslides for which we are able to assign a time window.

Therefore, for landslides that are not assigned a date by at least 2/4 of our methods, we now increase the size of the landslide polygon using a 20 m buffer for all methods and repeat, with the aim of trying to assign landslide timings to some of the landslides that experience this spatial mismatch. This improves the number of landslides we are able to detect in each case study event by 4% in Hiroshima and 5% in Zimbabwe and Trishuli, and will be added to the next version of the manuscript.

**Manuscript Changes:**

i) " *our methods allow 20% of landslides to be timed with an accuracy of 80%. This will allow multi-temporal landslide inventories to be generated for long rainfall events such as the Indian summer monsoon, which triggers large numbers of landslides every year and has until now been limited to annual-scale analysis.*" at lines 10-12 of the original manuscript has been changed to "*Application of this method could provide an insight on landslide timings throughout events such as the Indian summer monsoon, which triggers large numbers of landslides every year and has until now been limited to annual-scale analysis*" at lines 10-12 of the revised manuscript

ii) We have rewritten the conclusions section to better reflect the limitations of our proposed method (See reviewer 3 comment 1 for new text)

iii) Change to the last line of the introduction to avoid overselling the method: previous text (lines 53-56) *"Here we present methods of accomplishing this using Sentinel-1 SAR times series in Google Earth Engine. This will greatly improve the temporal resolution of optically-derived landslide inventories and unlock new comparisons between measured or modelled hydrological time series and landslide occurrence."*
New text at lines 59-63 of the revised manuscript: "*Here we present landslide timing methods based on the Sentinel-1 SAR dataset in Google Earth Engine that represent a step towards this goal of improved landslide inventory temporal resolution, and could unlock new comparisons between measured or modelled hydrological time series and landslide occurrence*"

iv) The description of the first improvement we have made to the methods "A new method based on bright spots observed at the edges of landslide polygons" has been added as Section 2.4.4 of the revised manuscript *"Technique 4: Geometric bright spots"* with the following text (lines 214-222 of the revised manuscript): *As well as shadows, the new geometry created by a landslide scar may result in bright spots on the far side of the scar, which are due to double-bounce scattering of the microwave energy between the exposed soil and vertical objects such as tree trunks and focussing of the energy scattered from the 3D surface into a small area in the radar coordinate system (Villard and Borderies, 2007, Fig. 2b). Similarly to the Geometric shadows technique, we applied a 20 m buffer to the landslide polygon, identified pixels that had undergone a significant increase in mean $\gamma 0$ between the pre-event and post-event image stacks and assigned these as "Bright". Here we found that the optimum $\gamma 0$ increase threshold was 5 dB. The co-event time series of these bright pixels was then analysed and a step increase between median bright $\gamma 0$ compared to the median background $\gamma 0$ (Sect. 2.4.1) was used as an indicator of landslide timing."*

v) The description of the second improvement we have made to the methods "increase in the size of the polygons" has been included in the section which, in the revised manuscript, is entitled "2.5 Identification of landslide date pairs" (See response to comment 53 of this review for revised text)

vi) Following the adoption of these methods, the results sections, Table 2 and figures 2-7 have been updated. While, the method changes slightly improved the sensitivity of the methods from ~20% to ~30%, the points made in the original discussion section remained valid with these new results and so did not need to be updated.

**Comment 2:** This being said, one could consider this paper as a step towards developing better tools along this research direction and in this regard, could be still valuable. And yet, authors do not clearly present their work. Unfortunately, the manuscript is not well written. I had to read some parts more

than once to understand the authors' point. The figures are not well designed either. I have many comments that I hope the authors find useful to improve their work.

**Response:** We will take on board the individual comments throughout the manuscript and the figures and hope that by addressing these, the manuscript will be made clearer. In addition to the responses to these individual comments and those from the other reviewers, which we believe should make the manuscript clearer. We have made the following changes:

1. Whereas before, we used "methods" to sometimes refer to the four individual landslide timing detection methods and sometimes to refer to the final overall method, we now refer to four individual "techniques" and a single method in which these four techniques are used in combination over both ascending and descending SAR tracks.

2. Headings have been added to Table 2 of the revised manuscript to allow easier distinction between the results from individual techniques and the final method that combines them.

**Comment 3:** Last but not least, I would like to test the code/tool they developed but unfortunately, it is not available. This is a preprint with DOI, so I did not really get why it was not shared already.

**Response:** The code is provided as a supplement to the revised of the manuscript as requested.

**Comment 4:** Line 18: "to emergency response coordinators". I would say rainfall-induced landslide inventories are rarely used by emergency response coordinators as they are generated at least weeks or months after an event. But if you are referring to some kind of indirect usage of the dataset (for instance, as an input to develop a landslide early warning system or something) please be more specific.

**Manuscript change:** *"to emergency response coordinators"* at line 18 of the original manuscript has been changed to *"for hazard management"* at line 17 of the revised manuscript

**Comment 5:** Line 18: "physical and empirical" there are also statistically-based models exploiting the very same dataset

**Manuscript change:** "physical and empirical models" at line 18 of the original manuscript changed to "physical, empirical and statistical models" at lines 17 of the revised manuscript

**Comment 6:** Lines 18-19: Could you please cite relevant literature.

**Response:** We have added references to Jones et al., 2021, (estimate of eroded sediment volumes); Ozturk et al., 2021 (landslide polygons used to train and validate regression models) ; Wu et al. 2015 (physical model of rainfall-triggered landsliding tested on optically-derived landslide inventory)

**Manuscript change:** *"(Jones et al. 2021 ; Kirschbaum and Stanley, 2018; Ozturk et al. 2021; Wu et al. 2015)"* added at line 18-19 of the revised manuscript. The following have been added to the reference list:

*Jones, J.N., Boulton, S.J., Bennett, G.L., Stokes, M. and Whitworth, M.R., 2021. Temporal variations in landslide distributions following extreme events: Implications for landslide susceptibility modeling. Journal of Geophysical Research: Earth Surface, 126(7), p.e2021JF006067.*

*Kirschbaum, D. and Stanley, T., 2018. Satellite-based assessment of rainfall-triggered landslide hazard for situational awareness. Earth's Future, 6(3), pp.505-523.*

*Ozturk, U., Saito, H., Matsushi, Y., Crisologo, I. and Schwanghart, W., 2021. Can global rainfall estimates (satellite and reanalysis) aid landslide hindcasting?. Landslides, 18(9), pp.3119-3133.*

*Wu, Y.M., Lan, H.X., Gao, X., Li, L.P. and Yang, Z.H., 2015. A simplified physically based coupled rainfall threshold model for triggering landslides. Engineering geology, 195, pp.63-69.*

**Comment 7:** Line 20: "the size location" the size, location

**Manuscript change:** This typo has been corrected in the revised manuscript.

**Comment 8:** Line 22: "the size and location" the size, location and timing too. As you said occurrence dates of landslides could not be accurate in some cases via optical images but also, as you said, if we have cloud-free images it is doable.

**Response:** Optical images can *always* be used to get info. on the size and location of landslides, but the timing is not always constrained because of cloud cover, as we go on to describe in the following sentences. We feel it would be confusing to include timing in the list here because it would immediately be contradicted in the following sentence.

**Comment 9:** Line 24: "Williams et al., 2018; Robinson et al., 2019". These examples are from earthquake-triggered landslide events. But you focus on rainfall-triggered landslides. So, please replace them with some examples of rainfall-triggered landslide events.

**Response:** It is true these papers focus on earthquake-triggered landslides, but they both focus on areas that also experience frequent rainfall-triggered landsliding. Therefore observations on the usability of optical satellite images for earthquake triggered landslides are also relevant for rainfall-triggered landslides. Furthermore, while we focus on rainfall events, the methods we propose could be applied to any event where the timing of triggered landslides is unknown due to cloud cover. This also includes sequences of earthquakes and storms, for example Lombok, Indonesia (Ferrario, 2019), Papua New Guinea, (Tanyas et al. 2022) and Gorkha, Nepal (Martha et al. 2017).

**Manuscript change:** New text: *"Studies based on optical satellite images affected by cloud cover that attempt to map landslides triggered by sequences of earthquakes and/or rainfall events may also be unable to distinguish between different triggers (e.g. Ferrario, 2019; Martha et al. 2017; Tanyas et al. 2022)"* at lines 35-37 of the revised manuscript

Additions to the reference list:

Ferrario, M.F., 2019. Landslides triggered by multiple earthquakes: insights from the 2018 Lombok (Indonesia) events. *Natural Hazards*, *98*(2), pp.575-592.

Tanyaş, H., Hill, K., Mahoney, L., Fadel, I. and Lombardo, L., 2022. The world's second-largest, recorded landslide event: Lessons learnt from the landslides triggered during and after the 2018 Mw 7.5 Papua New Guinea earthquake. *Engineering Geology*, *297*, p.106504.

**Comment 10:** Lines 28-29: This line needs to be rewritten. Also, why did you prefer the term "landfall", why not "landslide"

**Response:** A typhoon 'making landfall' refers to the time when the storm moves over land after having developed over the ocean. We are not using 'landfall' as a synonym for 'landslide'. With this, the sentence no longer needs to be rewritten.

**Comment 11:** Lines 30-31: Is that the case? Landslides triggered by each of those typhoons were mapped separately or not? It is not clear from the line if you indicate what already happened or this is just a hypothetical remark.

**Response:** The remark was hypothetical, we have changed the text to make this clearer

**Manuscript change:** *"If no cloud-free optical satellite imagery is acquired between these trigger events"* at line 29-30 of the original manuscript changed to *"If no cloud-free optical satellite imagery is acquired between such trigger events"* at lines 29-30 of the revised manuscript

**Comment 12:** Lines 33-34: "This limits analysis of these landslides to the annual scale (e.g. Marc et al., 2019a; Jones et al., 2021)." But, for instance, Marc and others generated monsoon-induced landslide inventories and to do that you just need pre- and post- monsoon images. So you do not need cloud-free optical satellite images through the monsoon. Please remove this reference and also please be more specific about the limitations of generating seasonal landslide inventories.

**Response:** Since these multi-temporal inventories are compiled from images before and after each monsoon season, we can only associate them with a given monsoon season and not with any particular peak in rainfall within that monsoon season. In this case, annual and seasonal almost mean the same thing, since almost all the landslide occurring annually in Nepal occur during the monsoon season. We will make the following change to make this clearer:

**Manuscript change***: "This limits analysis of these landslides to the annual scale (e.g. Marc et al. 2019a; Jones et al. 2021)"* at lines 33-34 of the original manuscript changed to *"This limits analysis of these landslides to the seasonal scale and prevents association of individual landslides or spatio-temporal clusters of landslides to specific peaks in rainfall."* at lines 33-35 of the revised manuscript.

**Comment 13:** Line 35: "Current alternative methods of landslide timing are generally not widely applicable." Please rewrite this line, is not clear what you mean. What are those alternative methods? And why do you think they are not widely applicable (any reference for this?). You haven't said anything about any alternative methods yet. Please first describe them and then you can evaluate those methods based on the literature.

**Response:** The evidence for this statement came in the following sentences. We have restructured this paragraph so that the conclusion is given at the end. To address this and comments 14 and 15 from this reviewer, we have rewritten this paragraph.

**Manuscript change:** *"Current alternative methods of landsliding are generally not widely applicable. Landslides that occur close to inhabited areas or that damage important pieces of infrastructure may be described in news reports or social media (e.g. Kirschbaum et al. 2010) Information on the timing of such landslides can also be generated through interviews with local residents (Bell et al. 2021). Rainfall intensity-duration thresholds have previously been derived for landslides in this way (e.g. Dahal and Hasegawa, 2008) and for landslides whose timings and properties are known through monitoring and field surveys (e.g. Guzzetti et al. 2007 ; Ma et al., 2015). However, this is unlikely to be the case for the majority of landslides in an inventory, and will be biased towards populated areas. Seismic recordings of landslides can also provide highly precise information on their timings, but will mostly record large landslides and require multiple seismic stations to allow timing of an individual, localised landslide (e.g. Yamada et al. 2012; Hibert et al. 2019)"* at lines 35-43 of the original manuscript changed to

*"Beyond remote sensing, several approaches have been used to constrain landslide timing. Landslides that occur close to inhabited areas or that damage important pieces of infrastructure may be described in news reports or social media (e.g. Kirschbaum et al. 2010; Franceschini et al. 2021) Information on the timing of such landslides can also be obtained through interviews with local residents (Bell et al. 2021) and through citizen science initiatives (Sekajugo et al. 2022). Rainfall intensity-duration thresholds have previously been derived for landslides in this way (e.g. Dahal and Hasegawa, 2008) and for landslides whose timings and properties are known through monitoring and field surveys (e.g. Guzzetti et al. 2007 ; Ma et al., 2015). However, such information on landslide timing is unlikely to be available for the majority of landslides in an inventory, and is usually biased towards populated areas and areas accessible by road (Sekajugo et al. 2022). Seismic recordings of landslides can also provide highly precise information on their timings, but will mostly record large landslides and require multiple seismic stations to allow timing of an individual, localised landslide (e.g. Yamada et al. 2012; Hibert et al. 2019). Current methods of obtaining landslide timing information in the absence of cloud-free optical satellite images are therefore not widely applicable."* at lines 38-49 of the revised manuscript.

**Comment 14:** Line 36: But this is not the method Kirshbaum and others or if you take a look at more recent literature Franceschini and others (DOI 10.1007/s10346-021-01799-y) used, this is the source information for them. Please tell us the method they used.

**Response:** We have revised this paragraph, which now contains more recent references.

**Manuscript change:** See changes made in response to comment 13 of this review.

**Comment 15:** Line 43: "will" Why did you switch to the future tense

**Manuscript change:** *"will be biased"* at line 41 of the original manuscript changed to *"is usually biased"* at line 45 of the revised manuscript.

**Comment 16:** Line 49. Please put a full stop before giving the example.

**Response:** A full stop is not needed here as this is only one sentence. However, it was broken up by our placing references halfway through. These will be moved to the end of the sentence

Previous text (lines 47-49 of original manuscript) : *"Numerous studies have shown that SAR data can be used to detect the spatial distribution of landslides in the case where their timing is already known (e.g. Aimaiti et al. 2019; Burrows et al. 2019, 2020; Ge et al. 2019 ; Konishi and Suga 2019 ; Masato et al., 2020 ; Mondini et al. 2021 ; Yun et al. 2015) , for example in the case of earthquake-triggered landslides."*
New text (lines 53-56 of revised manuscript): *"Numerous studies have shown that SAR data can be used to detect the spatial distribution of landslides in the case where their timing is already known, for example in the case of earthquake-triggered landslides (e.g. Aimaiti et al. 2019; Burrows et al. 2019, 2020; Ge et al. 2019 ; Konishi and Suga 2019 ; Masato et al., 2020 ; Mondini et al. 2021 ; Yun et al. 2015)."*

**Comment 17:** Line 57: "timed landslide information" this is the first time that I have heard this term and it sounds weird, please rephrase it. And please do it not only here but through the manuscript.

**Manuscript Change:** *"timed landslide information"* at line 58 of the original manuscript changed to *"information on landslide timing"* at line 64 of the revised manuscript

**Comment 18:** Line 61: "three potential landslide timing methods" you haven't said anything about these methods yet, so it is not clear what these methods are

**Manuscript Change:** *"We use Sentinel-1 time series over inventories of landslides whose timings are already known to test three potential methods of landslide timing"* at lines 60-62 of the original manuscript changed to *"We use Sentinel-1 time series over inventories of landslides whose timings are already known to test potential landslide timing methods"* at line 67-68 of the revised manuscript.

**Comment 19:** Line 63 "Case study events" does not sound right. Please revise it. e.g., Case studies or Landslide inventories

**Manuscript Change:** "Case study events" at line 63 of the original manuscript changed to "Case studies" at line 69 of the revised manuscript.

**Comment 20:** Line 66: Why did you take 20 pixels as your threshold? Why not 10? It could be better to do it without any filtering first. And then, you can identify the threshold for the landslide size that your method
works well.

**Response:** Statistics we use, such as the standard deviation of pixels within the landslide, require the landslide to contain multiple SAR pixels. As shown in Figure 5d-f (original manuscript) larger landslides are both more likely to be assigned a time window, and more likely for this time window to be correct. Furthermore since landslide-area distributions obey a power law, decreasing the threshold to 1000m2 would greatly increase the number of landslides required to be processed.

However, we have changed the structure of this sentence slightly to make it clearer

**Manuscript Change:** Previous text (lines 65-66 of original manuscript: *"We filtered each inventory to remove landslides smaller than 2000m2, so that each landslide was expected to contain a minimum of 20 10 x 10 m SAR pixels."*

New text (lines 91-92 of the revised manuscript): *"We filtered each inventory to remove landslides smaller than 2000 m2. Since the Sentinel-1 GRD data set has a pixel size of 10 x 10 m, this should result in a minimum of 20 SAR pixels within each landslide."*

**Comment 21:** Line 68 "inventories of landslides" landslide inventories. btw, you do not really need to cite
Emberson et al. (2021) for the Hiroshima inventory because it was already available, right?

**Response:** **"**landslide inventories triggered by short rainfall events" does not sound correct since it is the landslides triggered by the rainfall, not the inventory. However, later in this paragraph, we will change "inventory of landslides" to simply "landslides"

We will remove the reference to Emberson et al. here for the Hiroshima dataset and elsewhere in the manuscript.

**Manuscript Change:**

previous text (line 68 of the original manuscript): *"we used two inventories of landslides from Emberson et al. (2021) triggered by short rainfall events"*

new text (line 73 of the revised manuscript): *"we used two inventories of landslides triggered by short rainfall events"*

previous text (line 69 of the original manuscript) *"First, an inventory of 543 landslides triggered in Hiroshima, Japan…"*

new text (line 74 of the revised manuscript) *"First, landslides triggered in Hiroshima, Japan"*

previous text (line 72 of the original manuscript *"Second, an inventory of 383 landslides triggered by Cyclone Idai…"*

new text (line 78 of the revised manuscript): *"Second, landslides triggered by Cyclone Idai…"*

**Comment 22:** Line 70: *"heavy rainfall event which took place from 28 June to 9 July 2018"* I guess these are the dates that they were able to acquire pre- and post- event images to map landslides, right? But was the study area also exposed to heavy precipitation during the entire period? It would be useful to see the amount of precipitation (as time series) that each of your study areas received during the periods under consideration.

**Response:** No, these are not the dates of the acquired images, they are the days between which there was heavy rain over Hiroshima as specified by Hashimoto et al. (2020). We will attach rainfall time series (NASA GPM product) for Hiroshima and Zimbabwe as a supplement so that the extents of the event are visible. We agree this would help with visualising the timelines of the events. These are referenced at line 74 of the revised manuscript.

**Comment 23:** Line 74: "Planetdove" Planet Scope?

**Response:** Yes, Planet Dove are the satellites, PlanetScope is the name of the constellation so it is better to use this. Thank you for correcting the mistake

**Manuscript change:** *"Planetdove"* at line 74 of the original manuscript changed to *"PlanetScope"*

**Comment 24:** Lines 74-75: "images acquired on 20 and 24 March....the majority of landsliding occurred between the 15-17 March" You mean, they did not examine pre and post images to identify landslides solely triggered by the rainfall event, is this correct?

**Response:** Apologies, this was not clear, we should have specified that the images acquired on the 20$^{th}$ and 24$^{th}$ March were post-event images

**Manuscript change:** *"using Planetdove optical satellite images acquired on 20 and 24 of March"* at line 74 of the original manuscript changed to *"using post-event PlanetScope optical satellite images acquired on 20 and 24 March"* at line 79 of the revised manuscript.

**Comment 25:** Line 77: I thought you focus on rainfall-triggered landslides as also you indicated in the title of your manuscript. Why do you use the co-seismic landslide inventory of Roback et al. (2018)? Ok, now I understand what you have done. You removed landslides triggered by the mainshock and work with the others. However, you do not know if these landslides were triggered by the aftershocks or rainfall events. Also, it is not clear when they were triggered. The confusing thing is you mentioned that you focus on "inventories of landslides whose timings are known a-priori to test and develop landslide". However, this does not one of those inventories.

**Response:** This is not correct, as stated at line 87 of the original manuscript *"We also removed all landslides whose trigger was specified by Roback et al. (2018) to be something other than the mainshock"* i.e. we removed all landslides triggered by aftershocks or by rainfall and keep only the landslides triggered by the mainshock. We have updated the text to make this clearer.

**Manuscript change:**

previous text (lines 85-88 of the original manuscript): *"The Mw 7.8 mainshock on 25 April was followed by other possible landslide triggers, including the Mw 7.3 Dolakha aftershock on 12 May as well as the annual monsoon, whose onset was around 9 June (Williams et al. 2018). Therefore for this inventory, as well as filtering by landslide area, we also removed all landslides whose trigger was specified by Roback et al. (2018) to be something other than the mainshock."*

new text (lines 93-97) *"the Mw 7.8 mainshock on 25 April was followed by other possible landslide triggers including the Mw 7.3 Dolakha aftershock on 12 May as well as the annual monsoon, whose onset was around 9 June (Williams et al. 2018). Therefore, we also removed all landslides specified by Roback et al. (2018) to have been triggered by an aftershock or by rainfall and used only those triggered by the mainshock in our analysis"*

**Comment 26:** Why do not you simply pick another rainfall-induced landslide event inventory?

**Response:** The reason we do not use a rainfall-triggered landslide inventory here is that Nepal is a country for which the methods we develop here will be especially useful due to its extensive annual monsoon-triggered landsliding and long periods of cloud cover. It is also particularly challenging for SAR applications due to its steep topography, which results in distortion of the SAR images. It is thus particularly important to test the methods in this environment. Unfortunately, rainfall-triggered landslide inventories of known timing are not available here, so we use an inventory of earthquake-triggered landslides instead.

To make our motivation for using the Gorkha earthquake as a case study, we will make the following change to the text:

**Manuscript change:** previous text (lines 78-81 of original manuscript) *"The Nepal Himalaya is an area which experiences long periods of cloud cover and large numbers of rainfall-triggered landslides annually due to the monsoon and the country's steep topography. It is therefore useful to test landslide timing methods in this area, and, since well-timed landslide information is not widely available, we use earthquake-triggered landslides…"*

new text (lines 83-87 of the revised manuscript): *"The Nepal Himalaya is an area which experiences long periods of cloud cover and large numbers of rainfall-triggered landslides annually due to the monsoon and the country's steep topography. The steep topography of Nepal also makes it particularly challenging for SAR applications as it leads to distortion of the SAR imagery. It is thus important to test landslide timing methods in this environment, but inventories of rainfall-triggered landslides of known timing are not available. Therefore we instead used earthquake-triggered landslides."*

**Comment 27:** Line 81: "we used earthquake-triggered landslides, which can be assumed to occur concurrently with the ground shaking" you already said it above, please remove this.

**Manuscript change:**
We have removed the second part of this sentence, which as you point out, had already been stated at line 50 of the original manuscript.

**Comment 28:** Lines 81-83: "since the inventory of Roback et al. (2018) covers a large area, with different areas having different Sentinel-1 coverage, we focused on triggered landslides within three large valleys"
This does not explain why you focus on these three rectangular areas (actually one of them has a weird shape). You can cover a larger area mapped by Roback et al. (2018) and one Sentinel 1 image

should be covering at least an area covering two of those rectangles. So how did you identify these rectangles really?

**Response:** The valleys were selected since Marc et al. 2019a have mapped landslides in these valleys during several monsoons following the earthquake and therefore will be of interest for future studies. In fact, the Sentinel-1 time series is different for each Valley, making it complicated to combine these (See Figure below). Although tracks 19 and 85 cover both Trishuli (Tr) and Buri Gandaki (BG), track 85 is much more complete over Tr, which is further South than BG, which is further north. The two valleys lie in the same track but within different scenes and several scenes are missing on track 85 for BG. This is probably because the satellite had not long been operational at the time of the earthquake. It is therefore not possible to combine them into a single time series. I attach a map below to illustrate this. Scenes on the descending track (19) are shown in green. In this case, BG and Tr are in the same scene (Bhote Kosi (BK) is outside and belongs to track 121 instead). On track 85 (shown in orange), BG and BK are in different scenes, while Tr lies in an area where the two overlap. Since data for Tr can be supplied from either the northern or southern scene, it has a more complete time series than either BG or BK. It is for this reason that data from track 85 is only used in Trishuli, while BK and BG, we are limited to the descending tracks (121 and 19 respectively)

[Figure]

Finally, the squares on Figure 1c do not show the exact extents of the inventories but show the locations instead, as in figures 1a and 1b, where the study areas are also not square. In order to be clearer, we will replace the squares, which are not particularly informative, with plots of landslide locations for each event. This should make the extents of each inventory clearer.

**Manuscript Change:** Figure 1 previously showed boxes delineating the approximate locations of the inventory. In the revised manuscript, it shows individual landslide locations, thus better showing the study areas.

**Comment 29:** Lines 82-83: "large area & large valleys" please be more specific; either tell it directly or not mention it at all.

**Manuscript Change:** *"large area"* at line 82 of the original manuscript changed to *"large area (28,000 km2)"* at line 87 of the revised manuscript.

*"large valleys"* at line 83 of the original manuscript changed to *"valleys"* at line 89 of the revised manuscript.

**Comment 30:** "valleys see large numbers of rainfall-triggered landslides" it does not sound correct, please fix the language"

**Manuscript Change** "*see*" at line 83 of the original manuscript changed to *"experience"* at line 88 of the revised manuscript.

**Comment 31:** Line 84 "the timing of which would be one of the key applications of our method" Please remove this line, you already indicated your motivation.

**Manuscript Change** *"the timing of which would be one of the key applications of our method"* at line 84 of the original manuscript removed at line 90 of the revised manuscript.

**Comment 32:** Line 89: You are using the inventory mapped by Roback and others but for some reason, you are citing Marc and others. Too much self-citation, remove Marc et al. (2019)

**Response:** Roback et al. do not draw any conclusions about landslides triggered by the 12 May aftershock, since most of the images they used to map the landslides were acquired between 2-8 May (i.e. between the mainshock and the aftershock). Therefore their inventory does not provide information landslides triggered by aftershocks and we cannot cite their paper here. However, landslides triggered by the aftershock have been observed in Marc et al. (2019a) and Martha et al. (2017). We have slightly rewritten this paragraph to make our meaning clearer.

**Manuscript change:** *"The Dolakha aftershock is known to have triggered further landsliding (see Marc et al. 2019a)."* at line 89 of the original manuscript extended to *"The Dolakha aftershock is known to have triggered further landsliding (see Marc et al. 2019a) and in Roback et al. (2018) noted that in some areas, no cloud-free optical satellite images were available between the mainshock and the aftershock, making it difficult to differentiate between these two triggers."* at lines 97-100 of the revised manuscript.

**Comment 33:** Line 90: "close enough" not clear what this means. What would be close enough? What were the PGA or PGV values at those valleys? Did Martha and others map no landslide at those valleys and say this based on their observations? Or is this just an interpretation?

**Response:** This is based on observations from Martha et al. We have rewritten this paragraph in response to this and to comment made by this reviewer to make it clearer.

**Manuscript change:** See response to reviewer 1, comment 37

**Comment 34:** Figure 1: If you have such a plot (i.e., panel d) then please indicate ascending and descending images in panel d. This will be specifically important to see the dataset you used in the Gorkha case where some of the ascending images should be missing. Please properly indicate what those abbreviations stand for (e.g., H in panel a and so on)

**Manuscript change:** Figure 1 has been redesigned in the revised manuscript to improve clarity. We have adopted the change in labelling suggested by reviewer 1, comment 41 (e.g. H083A to H_asc). With this labelling, it should be clear which tracks are ascending and which are descending.

**Comment 35:** It is not clear how did you define pre-, co- and post- event image acquisitions. Do you explain this later in the method section? But you already refer to the term "co-event pair" in line 91, so the reader needs to know what it means. For instance, you indicate that in the Hiroshima case the heavy rainfall event occurred between 28 June and 9 July. Therefore, landslides were triggered (or mapped, as I mentioned this is confusing anyway) during this 10-day period. And you also mention that landslides were most likely triggered between 6 and 7th of July. Then why do you have such a large time period for "co-event pair"

**Response:** I apologise, we have generated confusion by using co-event (no quotation marks) to refer to the true pair of images spanning the landslide timing (as in line 91) while using "co-event" (with quotation marks) to refer to a "co-event" period that we have defined for testing the methods (i.e. a series of images that contains the correct co-event pair). Evidently, these are too similar and we will change to an alternative e.g. "trigger pair" to refer to the pair of images that span the trigger event and "co-event series" to refer to the series of images we have designated as co-event for testing purposes.

**Manuscript change:** Throughout the manuscript, we have changed co-event pair and "co-event" to differentiate between a date pair that spans the landslide timing and the co-event period that we defined to test the methods.

**Comment 36:** Also, why do you represent the real event date as a single day? You mentioned above some time slots that landslides were most likely triggered. Why do not you indicate them also in panel d?

**Response:** Time series are aligned relative to the timing of the earthquake (in Nepal) or relative to the peak in rainfall (in Hiroshima and Zimbabwe)

**Manuscript change:** *"Real trigger event timing"* in the legend for Fig.1d in the original manuscript has been changed to "*Real trigger event timing / peak"* in the legend for Fig. 1d in the revised manuscript.

**Comment 37:** these two earthquakes can be considered as a single triggering event in Bhote Kosi" You can not consider them as a single event. However, if you cannot differentiate landslides possibly triggered by different factors for a given period of time, this needs to be indicated as a source of uncertainty in your analyses. I do not know what could be the consequences, but obviously, this needs to be discussed later on in the manuscript

**Response:** While it is true that in terms of the spatial distribution of landsliding, we cannot consider them as a single trigger, that is not the aim of this study. Here we are trying to identify the timing of a landslide whose location we already know. Both the mainshock on 25 April and the aftershock on 12 May lie between our SAR acquisitions on 24 April and 18 May, it does not matter if the landslide happened at the time of mainshock or the aftershock, an assigned timing of 24 April – 18 May is right in either case. The paragraph has been rewritten to make this clearer. Furthermore, although there are some areas in the inventory of Roback et al. where the lack of cloud-free images between the mainshock and the aftershock mean that they cannot tell which event landslides were triggered by, in Bhote Kosi, 97% of the landslides *were* identifiable in images acquired between the two earthquakes and can therefore be confidently associated with the mainshock and dated as the 25$^{th}$ of April. This will be added to this paragraph to strengthen our argument.

**Manuscript Change:**

Previous text (lines 89-92 of the original manuscript): *"The Dolakha aftershock is known to have triggered further landsliding (see Marc et al. 2019a). However, of the three valleys we consider here, only Bhote Kosi was close enough to the epicentre to be affected by that event (Martha et al. 2017). Since the co-event pair of SAR images for Bhote Kosi (24 April – 18 May 2015) spans both the Gorkha earthquake on 25 April and the Dolakha aftershock on 12 May, these two earthquakes can be considered as a single triggering event in Bhote Kosi."*

New text (lines 97-105 of the revised manuscript*): "The Dolakha aftershock is known to have triggered further landsliding (see Marc et al., 2019a) and Roback et al. (2018) noted that in some areas, no cloud-free optical satellite images were available between the mainshock and this aftershock. However of the three valleys we consider here, landslides associated with this aftershock have only been observed in Bhote Kosi, which was the closest to the epicentre (Martha et al., 2017). 97% of the co-seismic landslides in Bhote Kosi were recorded as identifiable in imagery acquired prior to the aftershock and can therefore be associated definitively with the mainshock (Roback et al., 2018). Furthermore, since the co-event pair of SAR images for Bhote Kosi (24 April - 18 May) spans both the Gorkha earthquake on 25 April and the Dolakha aftershock on 12 May, these two trigger events are blended into a single time window by our methods in Bhote Kosi."*

**Comment 38:** Could please explain how you defined these time windows (i.e, 6, 3 and 2 months)? What is the logic behind it?

6 months is the approximate length of the monsoon season in Nepal (e.g. May – October), and we expect that our methods will be useful applied to monsoon-triggered landslides. Altering the length of this window is later explored in Fig 7. / Sect. 4.3 of the original manuscript. The periods of 2 and 3 months were selected to strike a balance between having enough images to calculate reliable statistics and having too much time elapse over the course of the time series (which could lead to incorporating landslides from previous monsoon seasons etc.). Using more images also increases the computation cost.

**Manuscript Change:** Previous text (lines 131-138 of original manuscript): *"For each of our three events, we defined a "co-event" period of approximately six months. We also defined a three-month pre-event period and two-month post-event period immediately before and after the co-event window. These pre-event and post-event image stacks are required in some of the methods outlined in Sect. 2.*

*The dates that make up the pre-event, "co-event" and post-event time series for each case study are shown in Fig 1d. The length of the "co-event" period was defined as 6 months for the Hiroshima and Zimbabwe events. For the three Nepal inventories, this was reduced to 5 months in order to allow a sufficient number of pre-event images to be acquired following the satellite launch in 2014 and sufficient post-event images to be acquired before the end of July, since few Sentinel-1 images are available over Nepal in August, September and October 2015."*

New text (lines 147-154 of the revised manuscript) : "*For each of our three events, we defined "pre-event", "co-event" and "post-event" periods (shown for each event on Fig.1d). The length of the co-event period was defined as six months based on the intended application to the Nepal monsoon, in which landslides may occur between May and October. However, for the three Nepal inventories, this was reduced to five months in order to allow a sufficient number of pre-event images to be acquired following the satellite launch in 2014 and sufficient post-event images to be acquired before the end of July since few Sentinel-1 images are available over Nepal in August and September 2015. The lengths of the pre-event and post-event time series were selected to be long enough to calculate statistics such as the mean without requiring the processing of unnecessary images.  These pre-event and post-event image stacks are required in some of the techniques outlined in Sect. 2.*"

**Comment 39** Line 131: "approximately six months" Later on you are saying two cases you took it as 6 months and in another one like 5 months. No need to repeat the same things. Please remove "approximately six months"

**Response:** We have rewritten lines 131-138 of the original manuscript to reduce repetition and explain the choice of 6 months, 2 months etc. See response to Comment 38 for changes to manuscript.

**Comment 40**: Line 139: "In this figure" Which figure? Figure 1d? Then say it, please

**Manuscript change:** *"this figure"* at line 139 of original manuscript changed to *"Fig 1d"* at line 156 of the revised manuscript.

**Comment 41** line 141: "ascending track 72 over Zimbabwe will be referred to as Z072A" this is not a good idea. Why don't you refer to it, for instance, as Zimbabwe-asc or Z-acs. Or something like we can easily understand what you are referring to.

**Manuscript change:** We have adopted the format suggested here of Z_asc rather than Z072A. This has been changed here and throughout the manuscript for all SAR tracks.

**Comment 42:** Please make a kind of introduction and tell us that you will introduce three different methods for some reason. And please indicate that reason too. It is difficult to follow the text. You are explaining your method (which is ok, I do not have any complaints) but if we do not understand why you are providing this information, we cannot follow you

**Manuscript change:** Text added at line 159 of the revised manuscript*: "Here, we present four potential techniques for analysing Sentinel-1 GRD time series and identifying the image pair spanning the landslide date."*

**Comment 43:** Line 150: "pixels that are dissimilar to those within the landslide, for example pixels located on the opposite side of a ridge, in a river or with different surface cover" Could you be more specific? How do you define similar and dissimilar pixels? Based on what? Based on land cover? Or do you have some other criteria you take into account? I see, in the next lines you are explaining those variables. But please first tell us what we are talking about (i.e., what you mean by dissimilar pixels) and then you can mention that you removed them.

**Response:** this is described in the following lines. The current structure describes first what we did (removing dissimilar pixels) and then how we did it (how similarity was assessed)

**Manuscript change:**

previous text (lines 148-156 of the original manuscript): *"Then we filtered this buffer to remove any pixels that lie within other landslide polygons and pixels that are dissimilar to those within the landslide, for example pixels located on the opposite side of a ridge, in a river or with different surface cover. In order to assess pixel similarity we calculated three variables from pre-event satellite imagery. First, we calculated the greenest pixel composite of the normalised difference vegetation index (NDVI) from Sentinel-2 (or, where this was unavailable, Landsat-8) images acquired in the year prior to each event. Pixels with similar NDVI values are expected to have similar land-cover. Second, we used the mean amplitude $A_{mean,j}$ (Eq. 2) and third, the amplitude variability $\Delta A_{mean,j}$  (Eq. 3) for every pixel j through a stack of N pre-event images. Pre-event amplitude and amplitude variability have previously been used by Spaans and Hooper (2016) to identify statistically similar pixels in SAR images. This allows us to remove pixels that are unlikely to exhibit similar behaviour to those within the landslide, for example pixels located on the opposite side of a ridge, in a river or with different surface cover."*

New text (lines 166-176 of the revised manuscript): *"Then we filtered this buffer to remove any pixels that lie within other landslide polygons and pixels that are dissimilar to those within the landslide. In order to assess pixel similarity we calculated three variables from pre-event satellite imagery. First, we calculated the greenest pixel composite of the normalised difference vegetation index (NDVI) from Sentinel-2 (or, where this was unavailable, Landsat-8) images acquired in the year prior to each event. Pixels with similar NDVI values are expected to have similar land-cover. Second, we used the mean amplitude $A_{mean,j}$ (Eq. 2 and third, the amplitude variability $\Delta A_{mean,j}$ (Eq. 3) for every pixel j through a stack of N pre-event images. Pre-event amplitude and amplitude variability have previously been used by Spaans and Hooper (2016) to identify statistically similar pixels in SAR images. This allows us to remove pixels that are unlikely to exhibit similar behaviour to those within the landslide, for example pixels located on the opposite side of a ridge, in a river or with different surface cover."*

**Comment 44:** "line 151: three surfaces" three variables might be better

**Manuscript change:** *"surfaces"* at line 151 of the original manuscript changed to "*variables*" at line 169 of the revised manuscript

**Comment 45:** Line 154: "amplitude variability" is this the third one? You mentioned the first and second variables but which one is the third?

**Response:** yes, amplitude variability is the third one. See response to comment 43 for the revised text.

**Comment 46:** Figure 2: Please fix the label of the panel (c) and please also indicate the label (c). Remove label (b) from panel (a).
**Response:** Sorry for the oversight, the labels on this figure have been fixed in the revised version of the manuscript. 'b' has been removed from panel (a)

**Comment 47:** What do you mean by "vegetation removed"? Do you mean because of landsliding? If it is the case,no need to indicate this.
**Response:** We will remove this label from the figure.

**Comment 48:** Is the blue bar not centered for some reason? Did you do this on purpose? Or is this something you
need to fix? And please indicate the corresponding panels while referring to "blue bars".
**Response:** The blue bars represent the duration of the rainfall event, but the time series is centred on the peak in the rainfall, which is not necessarily halfway through the duration of the rainfall event.

**Manuscript change:** indications have been added -to the corresponding panels when describing the blue bars in the caption of figure 2

**Comment 49:** line 160: "this"?

**Manuscript change***: "this landslide"* at line 160 of original manuscript changed to has been changed to *"each landslide"* at line 179 of the revised manuscript.

**Comment 50:** Line 164: When combining methods, we found" This is still your method section and you haven't said anything about other methods yet. This is to say that I do not understand what you are referring to?

**Response:** The combining methods was done in Sect. 3.1 of the original manuscript, (Sect 2.5 of the revised manuscript) and this sentence refers to that. However, we agree it does not belong here.

**Manuscript change:** "When combining methods … potential indicator of landslide timing" at lines 163-166 of the original manuscript have been removed in the revised manuscript.

**Comment 51:** Lines 160-166: Based on what you explained here how we should interpret Figure 2c? "A step change in the difference between the median landslide amplitude and the median background amplitude is then used as an indicator of landslide timing." Based on your interpretation, could you point out the timing of the landslide in Figure 2c. Which one is a step increase or a step decrease? Other than the signal received from the shadow area, I do not see any significant change in overall fluctuations of amplitude values associated with rainfall events (indicated by the blue bar in fig2c).

**Manuscript change:** Figure 2c and 2d in the original manuscript have been replaced by four panels (Fig 2 c-f in the revised manuscript) each of which shows one of the four landslide timing detection

techniques proposed in Section 2. We have also underlain the convolution output to each of these time series on these panels so that the reader can see how the peak / trough corresponds to the step change.

We have also changed the example landslide used as an example, with the new one having a clearer step increase.

**Comment 52**: Lines 185-188: I can not see any connections between these two lines. Could you be more clear about what you mentioned about uncertainty in landslide mapping in the first line?

**Manuscript change:** We have added a line linking the two sentences in question at –lines 203-205 of the revised manuscript: *"Spatial mismatches between landslide polygon locations could lead to pixels on the edges of landslides being excluded from the analysis."*

**Comment 53:** Line 196: "Step change identification" As usual, please help the reader to follow you. You have just mentioned three methods to identify the timing of landslides. And I guess you are going to combine these three methods to get the best result out of all, right? This is also not clear and needs to be indicated. And here you keep going with another step of your methodology. I think it would be great if you make a flow chart explaining your methodology. You can briefly describe each and every step of your method at the beginning and then we would have an idea about what is going to be in the next step. I know what I am suggesting is a super smart thing, is quite a traditional way of presenting your method but it is also a good way of doing this.

**Response:** In response to this comment, comment 54 from this reviewer and comments made by reviewer 3, Section 2.5 has been rewritten.

We also now distinguish throughout the manuscript between the four techniques and our method as a whole, which involves combining these techniques over 2 SAR tracks.

*Manuscript change:* New version of Section 2.5 (lines 223-259 of the revised manuscript):

*"2.5 Identification of landslide date pairs*

*Here we detail how the four techniques described above are used to retrieve landslide timings both individually and in combination. The variable associated with each technique is calculated for each landslide for every SAR image during the co-event period (Fig. 2 c-f). For each technique, we expect that the landslide should cause a step change in the time series, allowing us to identify the date pair spanning the landslide timing. In order to identify this step change, we take the co-event time series and subtract from it its mean value to obtain a co-event time series centred on zero. Then we convolve this series with a step function composed of a series of -1s and 1s that is twice its length. The output of this convolution is a series that, after truncating to the same length as the original co-event time series, should contain a peak (in the case of a step increase) or trough (in the case of a step decrease) at the location of the strongest step change in the time series.*

*The size of the peak or trough depends on the magnitude of the increase or decrease, the level of noise elsewhere in the time series and the length of the co-event time series (ndates). A bigger peak or trough for a time series of the same length indicates a larger step change and less noise and is therefore a more reliable indicator of landslide timing. We therefore apply a peak size threshold to remove unreliable landslide timing estimates. To select this threshold for each technique, we use the F1-measure, a statistic that combines both precision and recall. This F1-measure was calculated for a range of peak thresholds using the confusion matrix defined in Table 1 (Fig. 3). Based on this, we require a peak of 0.4×ndates for the Landslide-Background Technique, 0.2×ndates for the Pixel Variability Technique, 0.75 ×ndates for the Geometric Shadows Technique and 1.25×ndates for the Geometric Bright Spots Technique. We also assessed whether the level of noise in the time series for each technique (estimated from the variability in pre-event and post-event time series), could be used to indicate whether a timing estimate was likely to be correct, but found this to be less reliable than the convolution peak size.*

*==After identifying landslide timings using each technique individually, we combined these, assigning a date pair to a landslide if it was selected by at least two of our four techniques.== As previously described, a 20 m buffer was applied to each landslide polygon for Techniques 3 and 4 in order to allow for some spatial mismatch between the landslide polygons and the SAR imagery. This was not done for Techniques 1 and 2 as including non-landslide pixels unnecessarily would have the effect of muting the step change in the time series for these techniques. However, for landslides that have not*

*been assigned a timing at this stage, we now repeat the above process using this 20 m buffer for Techniques 1 and 2 as well. This step increases the number of landslides assigned a timing by around 5%.*

*The final step is to combine the predictions from the ascending (satellite moving northwards and looking east) and descending (moving southwards and looking west) SAR tracks. By carrying out the process described above using both the ascending and descending track SAR time series, we can obtain two sets of timings for a given landslide inventory, which can then be combined. This has several advantages. First, landslides that are not assigned a date pair using data from one track may be better timed by 250 the second, increasing the number of landslides that can be assigned a date pair. In particular, landslides that are masked due to foreshortening or layover may be better imaged in the other track. Second, the acquisition dates of the two tracks are slightly offset so a landslide that is assigned a date pair by both tracks is timed more precisely. For example, a correctly timed landslide in our Zimbabwe inventory should be timestamped as 7-19 March 2019 by the descending track time series and 12-24 March 2019 by the ascending track time series. From both together, the landslide would be timed as 12-19 March 2019, improving the precision from 12 days to 7. This more precise date is also more likely to be correct since it is derived from two sets of independent observations of the landslide.*

Some of this information was previously described in Sections 3.1 and 3.2 of the results section (highlighted in the above text). These parts have therefore been removed from those sections to avoid repetition.

**Comment 54:** Line 197: "Sects. 2.4.2, 2.4.1 and 2.4.3" just say above

**Manuscript change:** "Sects. 2.4.2, 2.4.1 and 2.4.3" is no longer used in the revised manuscript (see response to Comment 53 for new text in this section.

**Comment 55:** Line 198: "The step function was made up of a series of -1s and 1s of twice the length of the co-event time series" Do twice the length of the co-event time-series means 12 months? And why?

**Response:** This does not mean 12 months, it merely means twice the number of points in the SAR times series. (e.g. for a series of 12 SAR images, we convolve with a series made up of 12 -1s followed by 12 1s i.e. a series of length 24). The output of this convolution contains peak at the location that best agrees with the step function. In this way, we can automatically detect the location of a step change in the SAR measure

**Manuscript change:** See new text in Sect. 2.5 in response to comment 53.

**Comment 56:** Can't you make a figure to explain what you have done at this step?

**Response:** We hope that by adding the convolution functions to the panels we will add to Figure 2 (see response to comment 51 from this review) along with changes made to Section 2.5 (See response to comment 53) it will be clearer how this process works and how the peak / trough in the convolution corresponds to the step increase / decrease in the landslide timing method

**Comment 57:** Lines 216-217: "the correct date by chance for a method with no skill" is not clear!

**Manuscript change:** *"A random baseline was calculated from 1/ndates: the percentage of landslides we would expect to be assigned the correct date by chance for a method with no skill."* at lines 216-217 of the original manuscript changed to *"When compared to a random baseline calculated from 1/ndates (the percentage of landslides we would expect to be assigned the correct date pair by a method with no skill assigning a random date pair)…"* at lines 264-265 of the revised manuscript.

**Comment 58:** Table 2: What do those percentages stand for? For instance, in Hiroshima (H083A), you have 540 landslides and based on Pixel Variability you correctly identified the occurrence dates of 181 landslides, right? This means you correctly identified 33% of them. Then where did 59% come from? Obviously, the percentages indicate something else but I did not get what it is. I am sorry maybe it is my fault that I could not get it but this is not clear for sure.

**Response:** This has been misunderstood. In fact, we have 540 landslides in Hiroshima, 181 of these landslides are assigned a timing by the Pixel variability method and 59% of these 181 are correct (i.e. 107 correct).

**Manuscript change:** In order to make this less confusing, we will remove the percentages and instead provide correct / assigned in each column of the table. The table caption will also be updated to reflect this and to make this less confusing. Numbers in table 2 are also different in the revised manuscript to reflect the improvements we have made to the methods described in the response to comment 1

**Comment 58:** Actually based on these numbers and what you present in Figure 5, you can make an estimation for only a small fraction of the examined landslide population, right

**Response:** Yes correct. See response to comment 57 for changes to the table that should make this clearer.

**Comment 59:** You mentioned about confusion matrix, then why don't you present your results based on that structure?

**Response:** The confusion matrix in Table 1 of the original manuscript was designed to assess how the size of the peak in the convolution function can be used to predict whether or not an assigned date is correct and to allow the calculation of the F1-score. After the thresholds for each method have been selected and we move to combining predictions from multiple methods and tracks, there is no way to divide the landslides into TP, FP, FN, TN because we have three categories: landslide assigned the correct timing, landslide assigned the incorrect timing and landslide not assigned any timing.

**Comment 60:** Lines 243-245: "Out of all the non-masked landslides in each inventory, 23% were assigned a date in Hiroshima, 21% in Zimbabwe and 14% in Trishuli and of these, 80% of the estimated dates in Hiroshima were correct, 73% in Zimbabwe and 81% in Trishuli (Table 2)." So as you also indicated in the abstract, these are the percentages of correctly predicted landslides:
Hiroshima ~18%
Zimbabwe~15%
Trishuli~11%
Then how about the rest? Then I do not think what you mentioned in your abstract is convincing: "This will allow multi-temporal landslide inventories to be generated for long rainfall events such as the Indian summer monsoon, which triggers large numbers of landslides every year and has until now been limited to annual-scale analysis."
Landslides could occur on different dates over a monsoon season in a given area of interest. And we would miss a great majority of them if we use this technique. Therefore, I do not think we can confidently argue that multi-temporal inventories can be generated based on this method (This method does not mean to generate multi-temporal inventories anyway). I am not saying this method is useless but is also clear that this could be just a small step towards what you are arguing in your abstract

**Response:** See response to comment 1 for changes to the text that we have made to better reflect the limitations of our method.

**Comment 61:** Line 250: "Factors affecting performance of each method" This does not sound like your results. You should move this section to the Discussion section

**Manuscript change:** This section, renamed "Factors affecting landslide detection ability", has been moved to the discussion section and is now section 4.1 of the revised manuscript, starting at line 295.

**Comment 62:** Line 383 : " Application to future events" Please merge this section with the conclusion section, no need to have this heading, the paper is already too long.

**Manuscript change:** This has been merged with the conclusions to form a new section titled "Conclusions and future perspectives"

**Comment 63:** Line 384: As you said you are just estimating the time window that landslide might have occurred. So you are not estimating the exact occurrence date of landslides. You should clarify this also in your title.

**Manuscript change:** New title to address this and comments 7,8 from Reviewer 3: *"Using Sentinel-1 radar amplitude time series to constrain the timing of individual landslides: a step towards understanding the controls on monsoon-triggered landsliding"*

Throughout the manuscript, we no longer refer to "dating" or "dated" landslides, which implies we have assigned a single date rather than a 12-day window to the landslides. Instead we use "date pair"/"timing"/time window or similar.

**Comment 64:** Line 406: "generate multi-temporal" You are not generating landslide inventories. You are just trying to label existing landslide inventory in terms of their time of occurrences

**Response:** Yes, we will change this phrase here and elsewhere, since it is misleading.

**Manuscript change:** This phrase is no longer used in the revised section 5 "Conclusions and future perspectives" which has been rewritten following this comment, comment 1 of this review and comments from reviewer 3.

**Response to Reviewer 3**

Comment 1: This manuscript presents a method for estimating the time-window of landslide occurrence based on Sentinel-1 GRD data. The method is tested against inventories of landslides with known timestamp. The underlying research question is definitely an interesting one and may also be of importance for e.g. finding timestamps to known polygons or for doublechecking timestamps reported in landslide inventories. I do commend the authors as clearly a lot of work was put into the analysis.
Having worked with Sentinel-1 data for similar endeavors, the reported goodness-of-fit of results is well in line with my previous experience and expectations. However, the fact that only 1/5 of all landslides can be detected at all with reasonable accuracy indicates one of the following two conclusions for me:

• the level of maturity of the analyses is still rather low (this is not limited to the study at hand, but rather a general statement);
• SAR data is only suitable to a limited extent for the task at hand, or rather for the detection of sudden gravitational natural hazard events in general.
Overall, I think that this is an interesting contribution that is worth publishing subject to major revision. Generally, I suggest to report the results more as a potential contribution towards using Sentinel-1 data for estimating time-windows of event occurrence. Some parts of the manuscript read as if a well-working method is presented that works generically for identifying timings of landslides. However, this is still very much work in progress. For instance, I don't think that "This will allow multi-temporal landslide inventories to be generated for long rainfall events such as the Indian summer monsoon" in a comprehensive manner.

**Response:**

We have made some changes to the methods that somewhat improve the number of landslides for which we are able to assign a timing, improving from 20% to around 30% in favourable cases (See response to reviewer 2 comment 1 for details)

We have also revised the manuscript to make it clearer throughout that our methods cannot establish timings for all the landslides in an inventory (and in fact, will only provide timings for ~30%). In particular, we have made changes to the abstract, discussion and conclusions of the manuscript.

**Manuscript change:**

(i)     Previous text: *"our methods allow 20% of landslides to be timed with an accuracy of 80%. This will allow multi-temporal landslide inventories to be generated for long rainfall events such as the Indian summer monsoon, which triggers large numbers of landslides every year and has until now been limited to annual-scale analysis."*
New text: *" our methods allow 20-30% of landslides to be timed with an accuracy of 80%. Application of our methods could provide an insight on landslide timings throughout events such as the Indian summer monsoon, which triggers large numbers of landslides every year and has until now been limited to annual-scale analysis"*

(ii)    In Section 4.2 (original manuscript), we explore why some landslides are not assigned a timing. This section has been rewritten and now includes a final paragraph exploring possible ways in which we could increase the number of landslides assigned a timing.
New text at lines 364-371 of the revised manuscript *"future refinement of the method may increase the number of landslides assigned a timing. Possible means of accomplishing this include finding robust and systematic links between landslide setting and optimal thresholds for the individual techniques and (as suggested by Fig 5, 6). This would allow metric thresholds to be adapted to the setting of each landslide polygon. To address the problem of noise within the time series that masks the landslide timing signal, future work may involve adding a first step to our algorithm in which pixels exhibiting high levels of temporal variability are excluded from the landslide and background areas. Finally, our method may be improved by the development of other metrics, for example based on VH polarised SAR data or InSAR coherence time series, which have previously been used to*

*detect landslides in forested and arid zones respectively (Cabre et al., 2020; Handwerger et al., 2022)."*

(iii)  The conclusions section has been rewritten and merged with the previous section "Application to future events" in response to this comment and comments 1, 62, 63 and 64, made by reviewer 2. It now reads as follows (lines 439-467 of the revised manuscript) and better reflects the limitations of the methods.

*"5 Conclusions and future perspectives*

*In the case of long or successive rainfall events, landslide inventories compiled from optical satellite imagery are often poorly constrained in time, making it difficult to associate them with specific triggering conditions. Here we present a method of using Sentinel-1 SAR amplitude time series in Google Earth Engine to identify the timing of triggered landslides to within a few days. We find that by combining multiple techniques and ascending and descending track SAR, it is possible to assign timings to up to 30% of landslides in an inventory with an accuracy of 80%. A small number of landslides (5-10%) can be timed with an accuracy of ⩾90%. Here we applied our method to optically-derived landslide inventories, but it could also applied to datasets from other sources, for example those based on LIDAR scans or high resolution optical images that allow landslide volumes to be estimated (Bernard et al., 2021). The precision of our method, which in most cases is 12 days, should be sufficient in the case of multiple successive storms or earthquakes to attribute landslides to a given event (Ferrario, 2019; Janapati et al., 2019; Tanya, s et al., 2022). For monsoon landslide timings, this precision is not sufficient for construction of intensity-duration or intensity-antecedent rainfall thresholds at the hourly scale typical in the literature (e.g. Bogaard and Greco, 2018). However, thresholds based on weekly rainfall would be achievable and of interest for understanding triggering conditions in the Himalayan region. Furthermore, it should allow us to establish whether landslides occur in temporal clusters that relate to specific peaks in rainfall or are distributed throughout the monsoon. These two end-members would have very different implications in terms of hydrological and slope stability modelling and thus on hazard evaluation. Application of our method to the Indian summer monsoon should also allow us to better constrain whether landslides systematically occur with a specific delay after the onset of the monsoon and/or simultaneously with reported flooding or bursts of intense rainfall (Gabet et al., 2004).*

*Our method assigns timings to only 30% of landslides in an inventory, thus timing information is not obtained for the majority of landslides. Therefore, while our method provides a valuable insight into landslide timings during long or successive rainfall events, further work could allow us to obtain a more comprehensive view. First, our method may be refined by future studies, for example through variable metric thresholds adapted to the setting of each landslide or by incorporating both amplitude and coherence time series. Second, remote sensing approaches such as we present here could be combined, where available, with other methods of establishing landslide timing, for example reports of individual landslides or seismic data (Bell et al., 2021; Hibert et al., 2019; Yamada et al., 2012). Finally, We also expect that both the precision and the number of landslides that can be assigned timings may increase in the future as more SAR data becomes available, for example from the planned NISAR constellation. Overall, our method represents a step towards improved temporal resolution for triggered landslide inventories. This could further our understanding of monsoon-induced landsliding in the Nepal Himalaya and elsewhere."*

(iv) We will also make sure it is clear throughout the manuscript that we are not assigning specific dates, but instead time windows of (in most cases) 12 days to each landslide (see response to reviewer 2, comment 63)

**Comment 2:** There will definitely be a bias in terms of identified slides

**Response**: The biases towards which slides can be assigned a timing is related to section 3.3 (original manuscript) "Factors effecting the performance of each method." (Section 4.1, "Factors affecting landslide timing detection ability. –For example, it is clear that larger landslides are more likely to be assigned a timing than smaller landslides using our methods (Fig. 5 d-f). In the manuscript, we considered how this effected where our methods could be applied (for example, they will not work well for inventories of small landslides or in arid environments), but we did not consider how well the timings of the 30% of landslides we assign a timing to using SAR methods will represent the full inventory. In fact, this 30% will be biased to contain a somewhat higher proportion of large

landslides and landslides in more heavily vegetated areas than the original inventory. We will add this point to the revised version manuscript

**Manuscript change:**

New text at line 252 of original manuscript (line 298 of revised manuscript): *"For future applications, this helps to determine the environments where our methods can be expected to work well. It also provides an insight on potential biases in terms of the subset of a landslide inventory that is assigned timings by our methods."*

**Comment 4:** a vast majority of slides will be missed or - worse - labelled incorrectly,

**Response:** Our methods should not incorrectly label a large percentage of slides. We expect that if we apply our methods to an inventory of rainfall-triggered landslides, we will obtain an inventory in which ~70% have no timing information, ~24% are correctly timed and ~6% are incorrectly timed. We will be careful to make this clearer in the abstract, results and conclusions sections of the revised manuscript

**Manuscript Change:** Section 4.2 has been rewritten to make this clearer. New text "*landslides that are not assigned a date pair are a direct result of the target criteria of our method: a significant step change in at least two of the techniques outlined in Sect. 2.4. We showed in Sect. 2.5 that imposing a threshold on the convolution function peak was essential to reach a usable specificity, but this will also have required some correct timings to be discarded. Thus time series with a high degree of noise or where the landslide results in only a small step change in the metric will not produce a date pair.*"

Table 2 has also been altered in response to this and to Reviewer 2 Comment 58 to show number of correct timings / number of landslides assigned a timing. This should make it clearer that the number of landslides assigned a timing is much less than the number of landslides in the inventory and that most of the landslides that are assigned a timing by multiple techniques are assigned the correct date pair.

**Comment 5:** and things might look dire when thinking beyond the scope of this study, e.g. if no polygons are available.

**Response:** If polygons are not available for an event, it will not be possible to apply our methods in their current form – they are not designed to be applied to events for which we do not have a pre-existing landslide inventory. We will ensure this is clear in the revised version of the manuscript

**Manuscript change:** The introduction to our data and methods section has been rewritten to make it clearer that polygons derived from optical or multi-spectral satellite images are required before our method can be applied.

New text at line 64 of the revised manuscript: *"In order to obtain information on event timings ‑for landslides triggered by sequences of earthquakes or rainfall or by long rainfall events, we propose a two-step process, whereby landslide locations are mapped as polygons using optical or multi-spectral satellite imagery, and the timings of individual landslides are then obtained from SAR time series."*

**Comment 6:** Rather, I suggest to present the status quo and clearly highlight the limitations and highlight needs for further research based on the findings of this study.

**Response:** We have made several changes to the abstract, discussion and conclusions of the manuscript to address this.

**Manuscript change:** See changes made in response to comment 1 of this review and changes (i-iii) made in response to ‑comment 1 of reviewer 2

**Comment 7:** Also, the title should be clarified to indicate that time-intervals are identified rather than exact time stamps in terms of exact dates.

**Response:** We have altered the title in response to this comment, comment 8 of this reviewer and reviewer 2 comment 63. It now better reflects the results in the paper.

**Manuscript change:** New title "*Using Sentinel-1 radar amplitude time series to constrain the timings of individual landslides: a step towards understanding the controls on monsoon-triggered landsliding*

**Comment 8:** On a sidenote, I was slightly confused when I saw that the special issue title concerns the "Himalayan region", and study areas in this manuscript include Hiroshima and Zimbabwe. Since the Tr, BG and BK case studies are located in Nepal I think that's fine. The authors might consider adding some indication of the case study areas in the title, as I think it is actually very nice consider inventories from three different locations.

**Response:** The manuscript was submitted to the special issue since, while only 1 of our 3 landslide inventories was located in the Himalayas, the methods we develop are expected to be particularly useful in Nepal due to high levels of landsliding and cloud cover during the monsoon season in that country. We will emphasise this in the revised introduction

**Manuscript change:** previous text (lines 31-34 of original manuscript): *"the Indian Summer Monsoon (June – September) triggers hundreds of landslides every year in the Nepal Himalaya and cloud-free optical satellite imagery is unlikely to be available throughout this period (Robinson et al. 2019). This limits analysis of these landslides to the annual scale (e.g. Marc et al. 2019a; Jones et al. 2021)."* new text (lines 31-35 of the revised manuscript: *"the Indian Summer Monsoon (June – September) triggers hundreds of landslides every year in the Nepal Himalaya and cloud-free optical satellite imagery is unlikely to be available throughout this period (Robinson et al. 2019). This limits analysis of these landslides to the seasonal scale and prevents association of individual landslides or landslide clusters with specific peaks in rainfall (e.g. Marc et al. 2019a; Jones et al. 2021)."*

See response to comment 7 of this review for changes to the manuscript title.

**Comment 9:** Abstract: suggest to remove "thousands of" as this is somewhat unspecific without a time unit and potentially misleading

**Manuscript:** "*thousands of*" has been removed at line 1 of the revised manuscript

**Comment 10:** "Landslide locations are typically mapped using optical satellite imagery". There are many more methods that are "typically" used for such purposes, including ALS and orthophotos. The authors even mention this in section 2.1 ("drone and aerial imagery"). This might not be the case for all regions around the world as this clearly depends on the country under consideration, but the regions of interest have not yet been specified up to this point. VHR optical satellite imagery is expensive, while the spatial resolution of free data (e.g. Sentinel-2) is often too coarse to detect small slides. Free VHR data might be available e.g. through Google Earth, but not at the temporal resolution required to pinpoint the time windows to periods of some days.

**Response:** We do not really consider small landslides in this study as these are difficult to detect with Sentinel-1 SAR, which has a relatively coarse resolution (once multi-looked to reduce noise) and our methods are shown in Figure 5 of the original manuscript to perform better on larger landslides. Also, optical satellite images are also subject to cloud cover so, while they allow smaller landslides to be mapped, they do not address the problem we are tackling in this manuscript.

**Manuscript change:** "*typically*" changed to "*commonly*" at line 2 of the revised manuscript.

**Comment 11:** Section 2.1: I think the structure of this section can be improved. For instance: l.64f: "We used three published polygon inventories of landslides whose timings are known a- priori to test and develop landslide timing methods." Since the authors continue "We filtered each inventory to remove ..." I was wondering whether there was a reference for these data sets? This point is re-established two lines later with a reference to Emberson et al. (2021), leading to some interruption of the flow from a reader's perspective. l.66: "10 × 10 m SAR pixels" are mentioned. So I assume at this point that S-1 GRD data was used. Yet, the data source is unclear at this point. Also, why 20? I suggest to keep methodological considerations (e.g. filtering slides < 2000m², minimum number of SAR pixels, etc) separate from the initial inventory description.

**Response:** We have restructured this section to first describe each inventory and then to describe how they were filtered to remove small landslides and, in the case of the Nepal earthquake, landslides triggered by rainfall / an aftershock. We have also now specify that it is the Sentinel-1 GRD dataset that is used with 10 x 10 m pixel size.

**Manuscript change:** All descriptions of how the inventories were filtered are now together at lines 91-97 of the revised manuscript *"All inventories were filtered to remove landslides smaller than 2000*

*m2. Since the Sentinel-1 GRD data set has a pixel size of 10 × 10 m, this should result in a minimum of 20 SAR pixels within each landslide. This resulted in inventories of 543 landslides for the Hiroshima event and 383 for Zimbabwe. In Nepal, an additional step was required; the Mw 7.8 mainshock on 25 April was followed by other possible landslide triggers including the Mw 7.3 Dolakha aftershock on 12 May as well as the annual monsoon, whose onset was around 9 June (Williams et al., 2018). Therefore, we also removed all landslides specified by Roback et al. (2018) 95 to have been triggered by an aftershock or by rainfall and used only those triggered by the mainshock in our analysis. This left 650 landslides in Trishuli, 1554 in Bhote Kosi and 922 in Buri Gandaki."*

**Comment 12:** line 74: Planetdove: Do you mean PlanetScope DOVEs?

**Response:** Yes, we should have used "PlanetScope" here, not Planetdove. Thank you for correcting this.

**Manuscript change** "*Planetdove optical satellite images*" at line 74 of the original manuscript changed to "*PlanetScope optical satellite images*" at line 79 of the revised manuscript.

**Comment 13:** Please double check figure references in the body text. Fig. 1 - specifically, only Fig.1(d) -– is referenced the first time on line 135. Fig. 4 is the first figure to be referenced in the text.

**Response:** This was a mistake. At line 119 of the previous version of the manuscript, we refer to "(e.g. Tozang landslide, Mondini et al, 2021, Fig 4)". "Fig 4" there was not intended to refer to Figure 4 of our paper, but to Figure 4 in the paper of Mondini et al. 2021. This was not clear, and in the next version of the manuscript, we will reduce this reference to not name a figure number. After checking this, it appears that figure 4 of Mondini (2021) is actually taken from Mondini (2017) and this reference would be more appropriate since the landslide and method are described in more detail there. Therefore, we change the reference from Mondini et al. (2021) to Mondini (2017) rather than trying to reference an individual figure.

**Manuscript change:** "*e.g. Tozang landslide, Mondini et al., 2021, Fig. 4*" at line 119 of the original manuscript changed to "*e.g. Mondini, 2017*" at line 131 of the revised manuscript.

new reference: "*Mondini, A.C., 2017. Measures of spatial autocorrelation changes in multitemporal SAR images for event landslides detection. Remote Sensing, 9(6), p.554.*"

**Comment 14:** it took me a while till I figured out the meaning of the terminology you used for the orbit IDs (e.g. "H083A"). Please specify more clearly that this is a combination of study area, orbit number and orbit direction.

**Response:** This was also raised by Reviewer 2 (comment 41) who suggested a change in terminology from e.g. "H083A" to "H_asc". We have made this change throughout the manuscript, which should also resolve the comment here.

**Comment 15:** "We tested both of these polarisations, but found VV to perform better than VH so present only the results for VV." This is an interesting finding. How was this evaluated?

**Response:** Yes it is interesting since other studies (e.g. Handwerger et al. 2022) use VH and find it to be the most successful option. We attach a version of table 2 in the original manuscript which contains the same results from VH in Zimbabwe and Hiroshima (VH data are not available for the Nepal case study, which occurred soon following the launch of Sentinel-1 before dual-pol data began to be consistently acquired – this would be a further disadvantage to using VH).

In general, it is expected that VH polarised SAR should be more sensitive to changes in volume scattering (here, scattering within the forest canopy). Therefore, since the removal of vegetation due to a landslide should reduce this component, we would expect VH to decrease for a landslide polygon. Therefore, of the methods analysed here, we might expect VH to perform better than VV for the landslide-background decrease method (ls-b dec in the above table), which is in fact the case for H083A, Z079D and Z072A (but not H090D). However, since VH performs worse than VV for the other methods, it is better to use VV.

In particular, the bright spots method which we have introduced in the revised manuscript would not be expected to perform well for VH, since double-bounce scattering is generally very weak in cross-polarised SAR images.

**Manuscript Change:** A table comparing the results in Table 2, which are calculated using VV, and the equivalent results that could be obtained using VH polarised SAR for Hiroshima and Zimbabwe will be added to the supplementary material of the paper and referenced at line 123 of the revised manuscript.

We raise the possibility of incorporating VH SAR at lines 364-366 of the revised manuscript: *"Finally, our method may be improved by the development of other metrics, for example based on VH polarised SAR data or InSAR coherence time series, which have previously been used to detect landslides in forested and arid zones respectively (Cabre et al. 2020; Handwerger et al. 2022)"*

**Comment 16:** The copernicus DEM would have been a more recent DEM version, also available globally at a resolution of 30 m

**Response:** We chose to use the SRTM DEM here since it is already available as a dataset in Google Earth Engine, making it easier to integrate into the slope correction module for future users

**Comment 17:** l. 123: "... geographic coordinates at a resolution of 20 x 22 m and a pixel size of 10 x 10 m". I suggest to specify this further, this statement might be confusing to an audience from the broader field of natural hazards research not familiar with (SAR) satellite data

**Manuscript change:** previous text (lines 121-123 of original manuscript): *"we used the Google Earth Engine Sentinel-1 ground range detected (GRD) data set. These data are preprocessed following the workflow of Filipponi et al. (2019) to obtain the sigma0 backscatter coefficient at a resolution of 20 x 22 m and a pixel size of 10 x 10 m."*

New text (lines 137-139 of revised manuscript): *"The Sentinel-1 GRD data are preprocessed following the workflow of Filipponi (2019) to obtain the backscatter coefficient σ0 at a resolution of 20 x 22 m in radar coordinates. The data are then resampled onto a 10 m grid in projected coordinates"*

**Comment 18:** . 161/Figure 2: "A step change in the difference between the median landslide amplitude and the median background amplitude is then used as an indicator of landslide timing." It might be beneficial to plot this difference.

**Response:** Yes, this is a good point. In response to this and to similar comments from reviewer 2 (comment 51), we have made changes to Figure 2 of the revised manuscript to show these. This makes the step change clearer (Note we have also changed to a different landslide polygon due to changes requested by reviewer 2 – these time series are thus not comparable to those in Figure 2 of the original manuscript).
These panels also show as a grey line the convolution between the method and a step function to make it clearer how a step change results in a peak or trough in the convolution function

**Manuscript change:** Update to Figure 2 in the revised manuscript to improve visibility of step change and corresponding convolution function.

**Comment 19:** l. 164: "When combining methods, we found that using ..." Since the other two methods have not yet been described it might be better to move such statements towards the end of your methods section, when all three methods have been properly introduced?

**Manuscript Change:** This sentence at line 164 of the original manuscript has been removed in the revised manuscript.

**Comment 20:** 2.5 Step change identification: Up until here I was able to follow the text with rereading some parts several times again. Here I really had to pause and ponder upon backtracking multiple times to be able to understand what is described here. Please be more concise here on how all the aforementioned methods are combined exactly, how the step function is set up and why. Some sort of graphical depiction of the workflow would probably help a lot to foster overall understanding of the whole processing pipeline.

**Response:** This Section has been rewritten in response to this and to comments made by reviewer 2 in order to make it easier to follow. Additionally, the revision of Fig. 2, which now displays the time series and convolution with the step function, should help to clarify the text.

**Manuscript:** See reviewer 2 comment 53

**Comment 21:** . 216: reporting a baseline as reference is a good idea for putting the achieved results in context

**Response:** Thank you

**Comment 22:** Specificity is reported in table 2, F1-score is reported in Fig. 3. Providing confusion matrices of all results in the appendix might be interesting as a more detailed reference of results

**Response:** This was our mistake – Table-2 does not show specificity and we will alter the table caption to better describe the results. The confusion matrix in Table 1 of the original manuscript was designed to assess how the size of the peak in the convolution function can be used to predict whether or not an assigned date is correct and to allow the calculation of the F1-score. After the thresholds for each method have been selected and we move to combining predictions from multiple methods and tracks, there is no way to divide the landslides into TP, FP, FN, TN because we have three categories: landslides that are assigned the correct timing, landslides that are assigned an incorrect timing and landslides that are not assigned any timing. In response to this, and to comments made by reviewer 2, we will change to showing correct / assigned rather than assigned (% of assigned that are correct) for each method in Table 2. By providing the raw numbers, any statistic can be calculated by the reader if they require it.

**Comment 23:** Overall, appropriate performance metrics and their interpretation is of key importance. In fact, when thinking about the implications of the method presented here, this is crucial. If no validation data are available (e.g. when this method is applied to a new data set), a vast majority of identified dates (more precise: time windows) will be incorrect. This needs to be discussed.

**Response:** The problem is not that a vast majority of landslides will be assigned an incorrect time window, but instead that the majority of landslides will not be assigned any time window at all. The subpart of the inventory that has timings assigned will be biased towards larger landslides and those in more vegetated areas. We will discuss this in the revised version of the manuscript. (See response to comment 4 of this review.)
Of those landslides that are assigned a time interval, we expect that 80% of the time, this interval should be correct. Therefore if, in a new dataset, we observe a spatio-temporal cluster of landslides, we can assume the timing of this cluster is correct, since it is very unlikely that all the timed landslides in this cluster would be assigned the same incorrect date.

**Comment 24:** Publishing the code (e.g. on GitLab/GitHub) would be welcome for the final manuscript, but also of interest from a reviewer's perspective. If there are concerns with respect to sharing code before the publication is accepted, there are surely opportunities for embargos.

**Manuscript change:** The code will be provided as a supplement to the revised manuscript.

**Comment 25:** Please double check Equation (1), the dot and "area" are somewhat floating around there

**Manuscript change:** The dot was in the wrong place, thank you for spotting this. The equation has now been fixed.

**Comment 26:** Figure 2(a): green text on green background is hardly readable.
**Manuscript change:** We have adjusted the colours on this figure to make it easier to read

**Comment 27:** Figure 2(c): y-axis label is unreadable.
**Response** We are replacing this panel and (d) with a new set of panels in response to comment 18 of this review. We will ensure the y-axis is legible in this revised figure.

**Comment 28:** Table formatting in Table 2 is off (e.g. first line - alignment of "Total landslides"). The "Asc & Desc" columns are also aligned in a confusing way. Numbers should be right justified for better readability.

**Response:** Thank you for these suggestions, we will make these adjustments to the table to improve its readability.

**Comment 29:** Table 2: I suggest to split the information in the columns, and avoid combining multiple units (number and percentage, i.e. specificity) in one cell.

**Response:** In response to this and other comments made on this table by this reviewer and reviewer 2, we will change this so that each column contains correct / assigned rather than assigned (% of assigned that are correct).

**Comment 30:** Overall I suggest to use a more consistent plotting style (including readable colorscales) throughout the manuscript.

**Response:** The figures have been altered and we hope that the plotting style is now more readable.